# Structure of the human KMN complex and implications for regulation of its assembly

**Soumitra Polley**[1,6], **Tobias Raisch** [2,6], **Sabrina Ghetti**[1], **Marie Körner** [3], **Melina Terbeck**[1], **Frauke Gräter** [3,4], **Stefan Raunser** [2], **Camilo Aponte-Santamaría** [3], **Ingrid R. Vetter**[1] **& Andrea Musacchio** [1,4,5] ✉

Biorientation of chromosomes during cell division is necessary for precise dispatching of a mother cell's chromosomes into its two daughters. Kinetochores, large layered structures built on specialized chromosome loci named centromeres, promote biorientation by binding and sensing spindle microtubules. One of the outer layer main components is a ten-subunit assembly comprising Knl1C, Mis12C and Ndc80C (KMN) subcomplexes. The KMN is highly elongated and docks on kinetochores and microtubules through interfaces at its opposite extremes. Here, we combine cryogenic electron microscopy reconstructions and AlphaFold2 predictions to generate a model of the human KMN that reveals all intra-KMN interfaces. We identify and functionally validate two interaction interfaces that link Mis12C to Ndc80C and Knl1C. Through targeted interference experiments, we demonstrate that this mutual organization strongly stabilizes the KMN assembly. Our work thus reports a comprehensive structural and functional analysis of this part of the kinetochore microtubule-binding machinery and elucidates the path of connections from the chromatin-bound components to the force-generating components.

With approximately 30 core subunits and myriads of regulatory and accessory subunits, kinetochores are among the largest and functionally most complex molecular machines in the cell[1]. Their fundamental function is to provide a platform for efficient capture of spindle microtubules during cell division. In most species, kinetochore composition and organization are broadly conserved[2,3]. In their simplest form, identified in *Saccharomyces cerevisiae* and other yeast species, kinetochores assemble as a single chromatin-anchored complex that captures a single microtubule[4]. In most other species, this 'point' configuration is implemented in a more complex 'regional' version that captures multiple microtubules and that is probably generated by convolving a 'point' structure with multiple adjacent docking sites on a specialized confined chromatin domain, the centromere[1,4].

At low resolution, kinetochores appear as layered structures, with a centromere proximal layer (the inner kinetochore) and a centromere distal layer (the outer kinetochore)[1]. The inner layer incorporates the subunits of the so-called constitutive centromere-associated network (CCAN, also known as the Ctf19 complex in *S. cerevisiae*). Parts of the ten-subunit KMN super-assembly, which connect inward to the CCAN, contribute to the outer layer, probably together with other components[5,6]. KMN network subunits are in turn distributed into three subcomplexes (Fig. 1a), named the Knl1 complex (Knl1C, consisting of KNL1 and ZWINT), the Mis12 complex (Mis12C, consisting of DSN1, MIS12, NSL1 and PMF1), and the Ndc80 complex (Ndc80C, consisting of NDC80[HEC1], NUF2, SPC24 and SPC25)[7]. During mitosis, the KMN becomes recruited to the CCAN through Mis12C, which interacts directly with CENP-C and CENP-T, two CCAN subunits[8–17]. Both

[1]Department of Mechanistic Cell Biology, Max Planck Institute of Molecular Physiology, Dortmund, Germany. [2]Department of Structural Biochemistry, Max Planck Institute of Molecular Physiology, Dortmund, Germany. [3]Heidelberg Institute for Theoretical Studies, Heidelberg, Germany. [4]Max Planck School Matter to Life, Heidelberg, Germany. [5]Centre for Medical Biotechnology, Faculty of Biology, University Duisburg-Essen, Essen, Germany. [6]These authors contributed equally: Soumitra Polley, Tobias Raisch. ✉e-mail: andrea.musacchio@mpi-dortmund.mpg.de

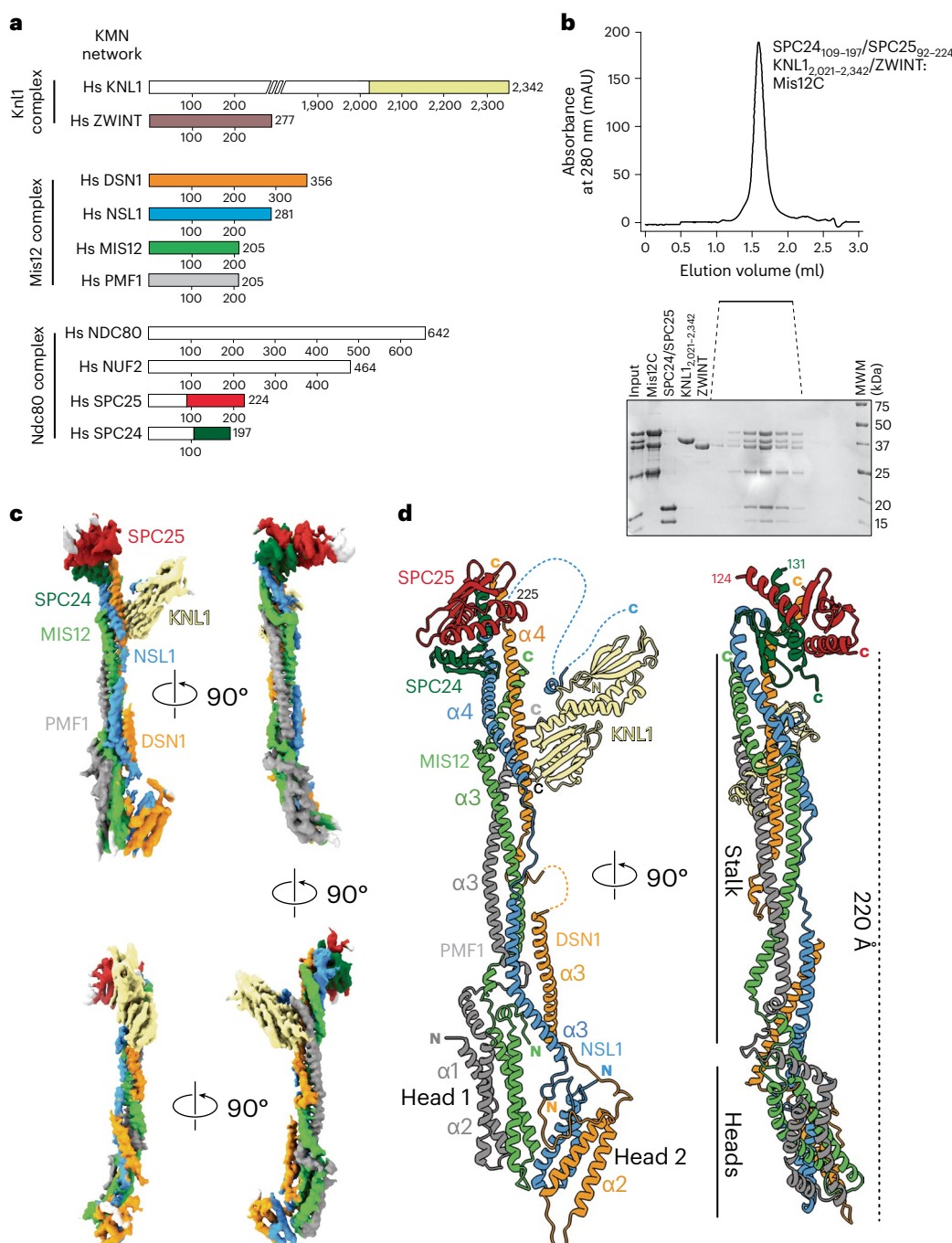

**Fig. 1 | Structural organization of the KMN network. a**, Schematic organization of the KMN subcomplexes. Fragments included in reconstitution for cryo-EM analysis are shown with different colors; missing fragments are in white. **b**, Size-exclusion chromatography of the reconstituted KMN assembly on a Superdex S200 5/150 column. Shown is a representative chromatogram from three repeats. **c**, Four rotated views of the experimental map. **d**, Cryo-EM model of the structured core of the KMN network in cartoon representation. N and C termini and other relevant structural elements are highlighted. Coloring corresponds to the cryo-EM map colors in **c**.

interactions are enhanced by phosphorylation of the Mis12C DSN1 subunits on conserved residues (Ser100 and Ser109 in humans)[13,18–23]. The prominent mitotic kinase Aurora B is responsible for these phosphorylation events that stabilize the outer kinetochore assembly during mitosis[18,23–25].

Within the KMN, Mis12C coordinates the association of Ndc80C and Knl1C[26,27]. Specifically, Mis12C recruits Ndc80C through interactions with the RWD (carboxy-terminal RING finger, WD repeat, DEAD-like helicases) domains of the SPC24/SPC25 subcomplex, and

Knl1C through interactions with the C-terminal RWD domain of KNL1 (refs. 11,12,14,28). High-resolution structures of the Mis12C, of the KNL1 tandem RWD domains that interact with Mis12C and of various segments of the Ndc80C, including in complex with microtubules, have been reported[11,14,21,28–36]. The Mis12C forms a moderately elongated (~22 nm) cylindrical structure with two head domains at the CCAN-binding end[21,32]. Each of the four subunits of Mis12C has long helical segments and runs in a parallel array along most of the long axis of the complex, with all amino termini at one end and all C termini

at the other. A low-resolution negative-stain electron microscopy analysis demonstrated that the KNL1 RWD domain docks on the side of the Mis12C cylinder[28].

Low-resolution rotary-shadowing analyses of Ndc80C revealed an even more elongated, ~65 nm structure—an overall shaft with globular domains at each end[12,37,38]. These consist of microtubule-binding calponin-homology domains at the microtubule-associated end, and—as already indicated—Mis12C-binding RWD domains at the centromere-targeting end. When combined, the Mis12C and Ndc80C add their length in series, generating 'antennas' with a combined contour length of almost 90 nm (ref. 12).

Despite this progress, several aspects of the organization of the KMN network remain unexplained. Specifically, the interfaces that allow Ndc80C and Knl1C to interact with Mis12C both engage RWD domains, but they appear to have no common structural theme[28]. Questions remain regarding how the Knl1–Mis12C interaction relates spatially to the Mis12C contact with SPC24/SPC25; whether cooperative, allosteric interactions favor establishing complete KMN assemblies, as suggested by the formation and stabilization of KMN complexes in assembly[18,39]; what the structure of ZWINT is and what interaction with KNL1 allows it to incorporate into the KMN complex; and how phosphorylation of Mis12C by Aurora B promotes binding to CENP-C and CENP-T[21,22]. Here, we shed light on these questions by reporting a cryogenic electron microscopy (cryo-EM) reconstruction of a KMN subcomplex, incorporating all interaction interfaces relevant to KMN assembly, which we validate with a combination of biophysical and cell biological experiments. Using model fitting and AlphaFold2 (AF2) modeling[40,41], we generate a molecular model of the entire KMN network and discuss its properties and implications for kinetochore assembly and function.

## Results and Discussion

### Structure determination of the human KMN network

We reconstituted an eight-subunit KMN network comprising the full-length Mis12C (Fig. 1a), a dimer encompassing residues 109–197 of SPC24 ($SPC24_{109-197}$) and residues 92–224 of SPC25 ($SPC25_{92-224}$), ZWINT and the C-terminal region of KNL1 (residues 2,021–2,342; numbering refers to isoform 1 of the *CASC5* gene that encodes KNL1). Elution from a size-exclusion chromatography column indicated a stoichiometric and monodisperse complex (Fig. 1b).

The resulting KMN sample was processed for cryo-EM data collection, and a reconstruction was calculated to a resolution of 4.5 Å as described in Extended Data Fig. 1a–f and Table 1. The map (Fig. 1c) includes density for the entire Mis12C, the globular RWD domains of the SPC24/SPC25 dimer and the tandem RWD domains ($RWD^N$ and $RWD^C$, marked in Fig. 2a) of KNL1, thus encompassing all interfaces that are relevant for the assembly of the KMN network. The region of KNL1 and ZWINT preceding the RWD domains are expected to interact in a heterologous helical coiled coil (see below) but are not clearly distinguishable in our maps. We generated a molecular model of the KMN network by fitting previously determined high-resolution structures of the individual modules as discussed in the Methods, complemented where necessary with AF2 predictions, and refining this model against the experimental cryo-EM map.

### Overall appearance of KMN network

Mis12C has a long axis of approximately 22 nm (refs. 21,32) (Fig. 1d). It has two globular heads (head 1 and head 2), each comprising a four-helix bundle, which encompass the N-terminal α1–α2 helices of PMF1 and MIS12, and of DSN1 and NSL1, respectively, and a stalk, which is a compact parallel helical bundle of the four subunits. The SPC24/SPC25 dimer, two tightly interacting RWD domains, caps the complex near the C-terminal tails of the Mis12C stalk, engaging in tight interactions with the C-terminal regions of DSN1 and NSL1. The $RWD^C$ of KNL1 docks against the lateral surface of the stalk, approximately 5 nm from

**Table 1 | Cryo-EM data collection, refinement and validation statistics**

| | KMNZ complex(EMD-18179), (PDB 8Q5H) |
|---|---|
| **Data collection and processing** | |
| Magnification | ×105,000 |
| Voltage (kV) | 300 |
| Electron exposure (e−/Å²) | 63.9 |
| Defocus range (μm) | −1.5 to −3.0 |
| Pixel size (Å) | 0.68 (physical) |
| Symmetry imposed | C1 |
| Initial particle images (no.) | 1,792,983 |
| Final particle images (no.) | 230,597 |
| Map resolution (Å) | 4.50 |
| FSC threshold | 0.143 |
| Map resolution range (Å) | |
| **Refinement** | |
| Initial model used (PDB code) | AF2 model |
| Model resolution (Å) | 5.5 |
| FSC threshold | 0.5 |
| Map sharpening *B* factor (Å²) | DeepEMhancer |
| **Model composition** | |
| Non-hydrogen atoms | 9,575 |
| Protein residues | 1,173 |
| Ligands | – |
| *B* factors (Å²) | |
| Protein | 101.0 |
| Ligand | – |
| R.m.s. deviations | |
| Bond lengths (Å) | 0.004 |
| Bond angles (°) | 0.815 |
| **Validation** | |
| MolProbity score | 1.92 |
| Clashscore | 15.6 |
| Poor rotamers (%) | 0.0 |
| Ramachandran plot | |
| Favored (%) | 96.5 |
| Allowed (%) | 3.4 |
| Disallowed (%) | 0.1 |

the upper end, engaging the DSN1 and PMF1 subunits. It also captures a C-terminal motif of NSL1 that is connected to the stalk through a flexible (and largely invisible) linker.

In the KMN structure, the two Mis12C heads are closely packed against each other (Extended Data Fig. 2a). Comparison with a previously determined crystal structure of the complex of the Mis12C with CENP-C indicates that this closely packed (closed) conformation of the heads is incompatible with robust CENP-C binding[21]. In the complex with CENP-C, head 2 sways out of position, and this conformational change resolves a predicted steric clash of CENP-C with NSL1 (see below and Extended Data Fig. 2a). Indeed, upon binding to Mis12C, CENP-C also pushes head 1 and the stalk away from each other, so that the closed conformation of the unbound complex is also comparatively more compact, as shown by a superposition (Extended Data Fig. 2a). Aurora

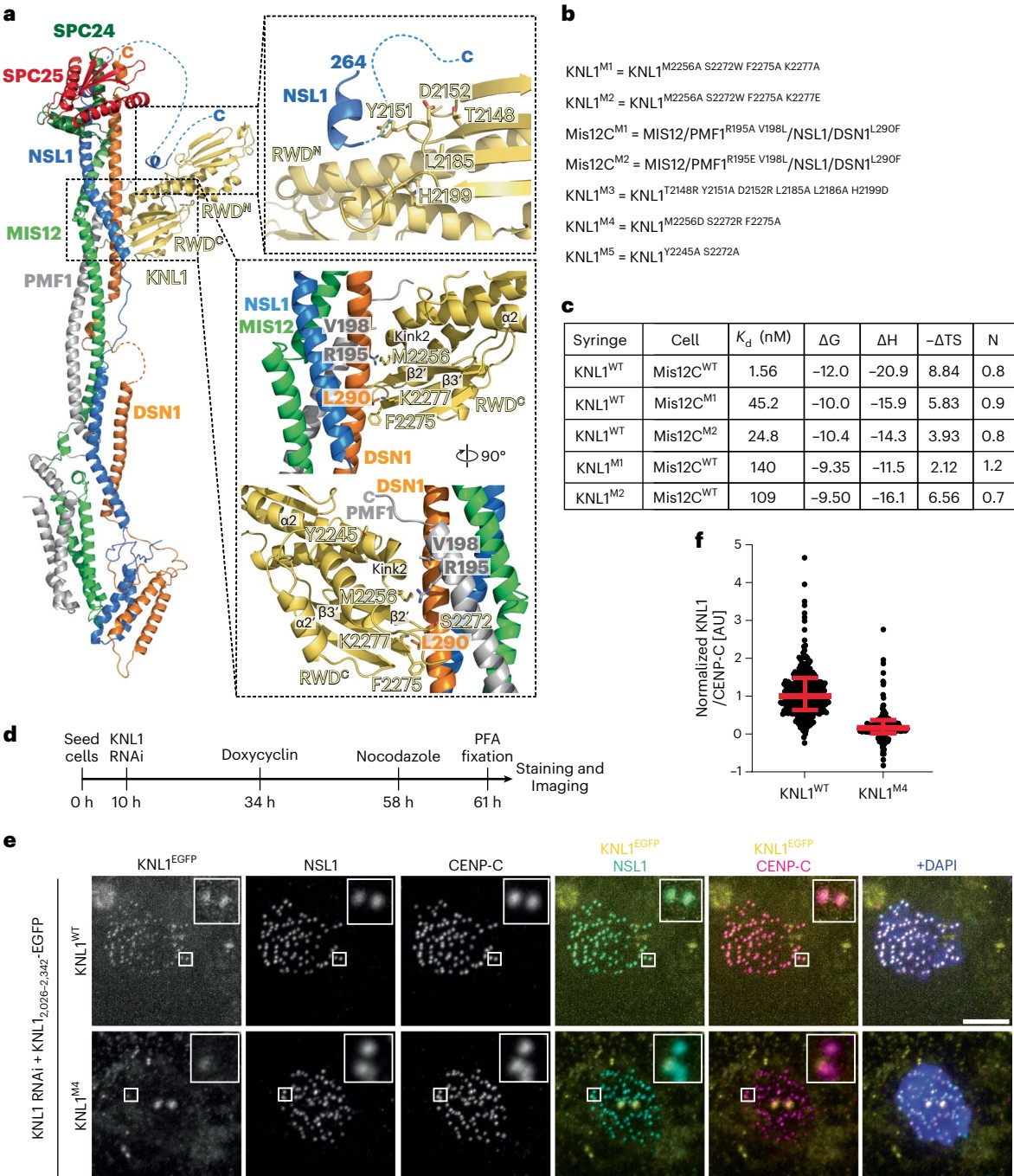

**Fig. 2 | The Mis12C/KNL1 interaction. a**, Cartoon model of the minimal KMN network and closeups describing the Mis12C–KNL1 interface. The upper inset is slightly rotated relative to the view on the left-hand side. The bottom two views are rotated approximately 90°. Dotted lines represent invisible segments of the model of the NSL1 C-terminal tail and are merely illustrative of the missing connections. In Extended Data Fig. 3b we include an AF2 model of the missing regions. **b**, List of seven mutants chosen for biochemical or biological validation. **c**, Table summarizing the results of the indicated ITC titration experiments.

**d**, Schematic representation of protocol for cell biology experiment shown in **e**. **e**, Representative images of cells depleted of endogenous KNL1 and expressing the indicated EGFP–KNL1$_{2,026-2,342}$ constructs, which were designed to reduce or prevent the interaction of KNL1$_{2,026-2,342}$ with endogenous Mis12C. Scale bar, 5 µm. **f**, Quantification of the experiment in **e**. Red bars represent median and interquartile range of normalized kinetochore intensity values for KNL1$^{WT}$ ($n = 1,003$) and KNL1$^{M4}$ ($n = 760$) from three independent experiments.

B kinase phosphorylates Ser100 and Ser109 (and possibly Ser95) on DSN1 to facilitate binding of CENP-C and CENP-T to Mis12C[18–23]. Deleting the region encompassing these residues (the 'phosphorylation loop') mimics the effect of Aurora B phosphorylation[19,21,22], suggesting that the non-phosphorylated state of the phosphorylation loop stabilizes the closed conformation of the heads, while phosphorylation

facilitates their release into an open conformation. Direct contacts between head 1 and head 2 in the closed conformation are scarce, with the most prominent involving the localized interactions of the side chains of Asp75 and Trp78 on NSL1, and of Ser56, Val58 and Lys6 on MIS12 (located in the region identified as a dotted circle in Extended Data Fig. 2a, bottom left). The phosphorylation loop may contribute

substantially to the stabilization of the closed position, but the experimental map is not sufficiently well-resolved in this region to pinpoint the exact position of the DSN1 phosphorylation loop. However, AF2 provides a robust prediction for this region that confirms the important role of the phosphorylation loop in the non-phosphorylated form, as a stabilizer of the closed conformation (Extended Data Fig. 2b). Specifically, the strongly positively charged DSN1$_{80-112}$ segment is predicted to bind at a negatively charged interface between head 1 and head 2, opposite to the CENP-C binding site (Extended Data Fig. 2b). Thus, the DSN1 phosphorylation loop does not have to compete directly with CENP-C binding, as we had reasoned based on local sequence similarity[21]. Instead, changes in the relative position of the head domains appears to be the dominant mechanism. Phosphorylation of the phosphorylation loop, like its deletion, may modify the charge balance in this region of DSN1, weakening the closed state of the heads and releasing a steric blockade to CENP-C binding as the heads open.

## The Mis12C–KNL1 interaction

The Mis12C stalk engages residues 282–304 of DSN1 and residues 188–204 of PMF1 (near the C-terminal ends of these subunits) to interact with the KNL1 RWD$^C$. The strongly kinked end of the α1′ helix and the β2′–β3′ loop of RWD$^C$ insert into a guide generated by the intertwined PMF1 and DSN1 helices (Fig. 2a; apostrophes or lack thereof indicate secondary structure elements in RWD$^C$ and RWD$^N$, respectively. See Extended Data Fig. 3a). A second crucial contact, which mainly engages RWD$^N$, captures the C-terminal segment of NSL1 (Fig. 2a and Extended Data Fig. 3b). This part of the structure is closely related to a previously determined crystal structure of the KNL1 C-terminal region with an NSL1 peptide (PDB 4NF9; ref. 28). In addition to the interaction with the KNL1 C-terminal region, AF2 modeling also suggests that the NSL1 tail is further stabilized by the terminal part of the ZWINT–KNL1 coiled coil, as explained more thoroughly below. Thus, RWD$^N$ and RWD$^C$, respectively specialize in the interaction with the C-terminal region of NSL1 and with the DSN1 and PMF1 helices in the stalk.

The extensive interaction includes 2,000 Å$^2$ of protein surface. Using isothermal titration calorimetry (ITC), we determined the dissociation constant ($K_d$) of the Mis12C–KNL1$_{2,026-2,342}$ interaction to be ~1.6 nM (Fig. 2c and Extended Data Fig. 4a), in line with previous measurements[21,27]. Mutations in the Mis12C interface implicated in KNL1 binding (Mis12C$^{M1}$ and Mis12C$^{M2}$; mutated residues are listed in Fig. 2b and hydrodynamic profiles are shown in Extended Data Fig. 3c–f, where hydrodynamic profiles of other mutants discussed in the text are also displayed unless specified) increased the $K_d$ of the Mis12C–KNL1 interaction from ~15-fold to ~29-fold (Fig. 2c and Extended Data Fig. 4b,c). Similar binding measurements with two mutants predicted to affect the KNL1 interface involved in Mis12C binding (KNL1$^{M1}$ and KNL1$^{M2}$) also weakened the interaction, increasing the $K_d$ from ~12-fold to ~100-fold (Fig. 2c and Extended Data Fig. 4d,e). Even though not tested in vitro for Mis12C binding, three additional KNL1 mutants designed to impair binding to Mis12C (KNL1$^{M3}$, KNL1$^{M4}$ and KNL1$^{M5}$; Fig. 2b) were tested for kinetochore localization in vivo. For this procedure, we applied the pipeline schematized in Fig. 2d to deplete endogenous KNL1 by RNAi (Extended Data Fig. 4f) and replace it with stable inducible transgenes expressing KNL1$_{2,026-2,342}$ fused to EGFP to monitor kinetochore localization (Extended Data Fig. 4g). The wild-type KNL1$_{2,026-2,342}$ fragment localized robustly to kinetochores. KNL1$^{M4}$ failed to localize to kinetochores even if expressed at the same levels of the control wild-type construct (Fig. 2e,f and Extended Data Fig. 4g). KNL1$^{M3}$ and KNL1$^{M5}$ also failed to localize to kinetochores but were also expressed at lower levels, possibly also indicative of structural destabilization (Extended Data Fig. 4g–i).

## The Mis12C–Ndc80C interaction

The C-terminal regions of DSN1 and NSL1 mediate interactions with the SPC24/SPC25 dimer required to recruit the Ndc80C onto the kinetochore[27]. As they emerge from the end of the stalk with their α4 helices, both the DSN1 and NSL1 chains make a sharp turn to interact with SPC24/SPC25 (Fig. 3a). The interaction is also extensive in this case, collectively burying ~2,500 Å$^2$. The interaction of SPC24/SPC25 with DSN1 closely resembles those it has with CENP-T in chicken[14] (PDB 3VZA) and with Dsn1 in yeast[11] (PDB 5T6J). Specifically, the position and binding mode of the DSN1 α5 helix is very similar to that of an equivalent helix of CENP-T in complex with SPC24/SPC25 (Extended Data Fig. 5a). Indeed, the two helices in DSN1 and CENP-T have related sequences that alternate hydrophobic and positively charged residues[21].

Measured by ITC, the $K_d$ of the interaction of full-length SPC24/SPC25 with Mis12C was ~46 nM (Fig. 3c and Extended Data Fig. 5c). Also using ITC, we previously measured a smaller $K_d$ (4 nM) for the interaction with the entire Ndc80C[27], suggesting a slight decrease in binding affinity when using SPC24/SPC25. Mutants affecting hydrophobic residues in the short DSN1 α5 helix, including Mis12C$^{M3}$ and Mis12C$^{M4}$ (Fig. 3b), reduced the binding affinity for the SPC24/SPC25 heterodimer from ~10-fold to ~20-fold (Fig. 3c and Extended Data Fig. 5c,d). When reintroduced in HeLa cells depleted of endogenous Mis12C (following the protocol schematized in Fig. 3d), the same mutants caused an ~50% reduction in the kinetochore levels of endogenous Ndc80C (Fig. 3e,f). In addition to engaging the α5 helix, NSL1 also interacts with SPC24/SPC25 by engaging the PVIHL motif (residues 209–213) that caps the end of its α4 helix (Extended Data Fig. 5a), previously shown to be essential for the interaction with Ndc80C[27]. Within this motif, the side chain of Ile211 is entirely buried at the interface with SPC24 (Fig. 3a and Extended Data Fig. 5a). Mutations of a single hydrophobic residue on the NSL1 α5 helix (I216A in mutant Mis12C$^{M5}$; Fig. 3b) at the interface with SPC24/SPC25 reduced the binding affinity of Mis12C and SPC24/SPC25 by a factor of ~15-fold (Fig. 3c and Extended Data Fig. 5e); the Mis12C$^{M5}$ mutant had a similar disruptive effect on Ndc80C kinetochore recruitment as that described above for the Mis12C$^{M3}$ and Mis12C$^{M4}$ mutants (Fig. 3e,f).

We also tested two constructs with mutations affecting SPC24/SPC25 residues at the interface with DSN1 (SPC24$^{Y138A}$/SPC25 and SPC24/SPC25$^{L154A-L176A}$, indicated as SPC$^{M1}$ and SPC$^{M2}$ in Fig. 3b, respectively). The mutants had ~5-fold and ~27-fold reduced binding affinity for the wild-type Mis12C, respectively (Fig. 3c and Extended Data Fig. 5f,g). We introduced the SPC$^{M1}$ and SPC$^{M2}$ mutations in the complete recombinant Ndc80C (further modified by addition of a fluorophore; see Methods) and electroporated the resulting mutant complexes in

---

**Fig. 3 | The Mis12C–Ndc80C interaction. a**, Cartoon model of the minimal KMN network and closeups describing the Mis12C–Ndc80C interface. **b**, List of five mutants chosen for biochemical or biological validation. **c**, Table summarizing the results of the indicated ITC titration experiments. **d**, Schematic representation of protocol for cell biology experiment shown in **e**. **e**, Representative images of cells depleted of endogenous Mis12C and electroporated with the indicated Mis12C constructs. The levels of Ndc80C$^{Hec1}$ were monitored. Scale bar, 5 μm. **f**, Quantification of the experiment in **e**. Red bars represent median and interquartile range of normalized kinetochore intensity values for NDC80$^{Hec1}$ in presence of the indicated Mis12C constructs ($n$ = 355 kinetochores for Mis12C$^{WT}$, $n$ = 365 for Mis12C$^{M3}$, $n$ = 298 for Mis12C$^{M4}$, $n$ = 358 for Mis12C$^{M5}$). The levels of electroporated Mis12 complex quantified in the panel were evaluated in a distinct control sample ($n$ = 429 kinetochores for Mis12C$^{WT}$, $n$ = 393 for Mis12C$^{M3}$, $n$ = 331 for Mis12C$^{M4}$, $n$ = 361 for Mis12C$^{M5}$). Data were collected in three independent experiments. **g**, Fluorescence and immunofluorescence analysis on cells processed for Ndc80C RNAi and electroporated with the indicated FAM-labeled Ndc80C. This mount is duplicated from a larger experiment displayed in Extended Data Fig. 6a, where sample size and number of repeats are also reported. Scale bar, 5 μm.

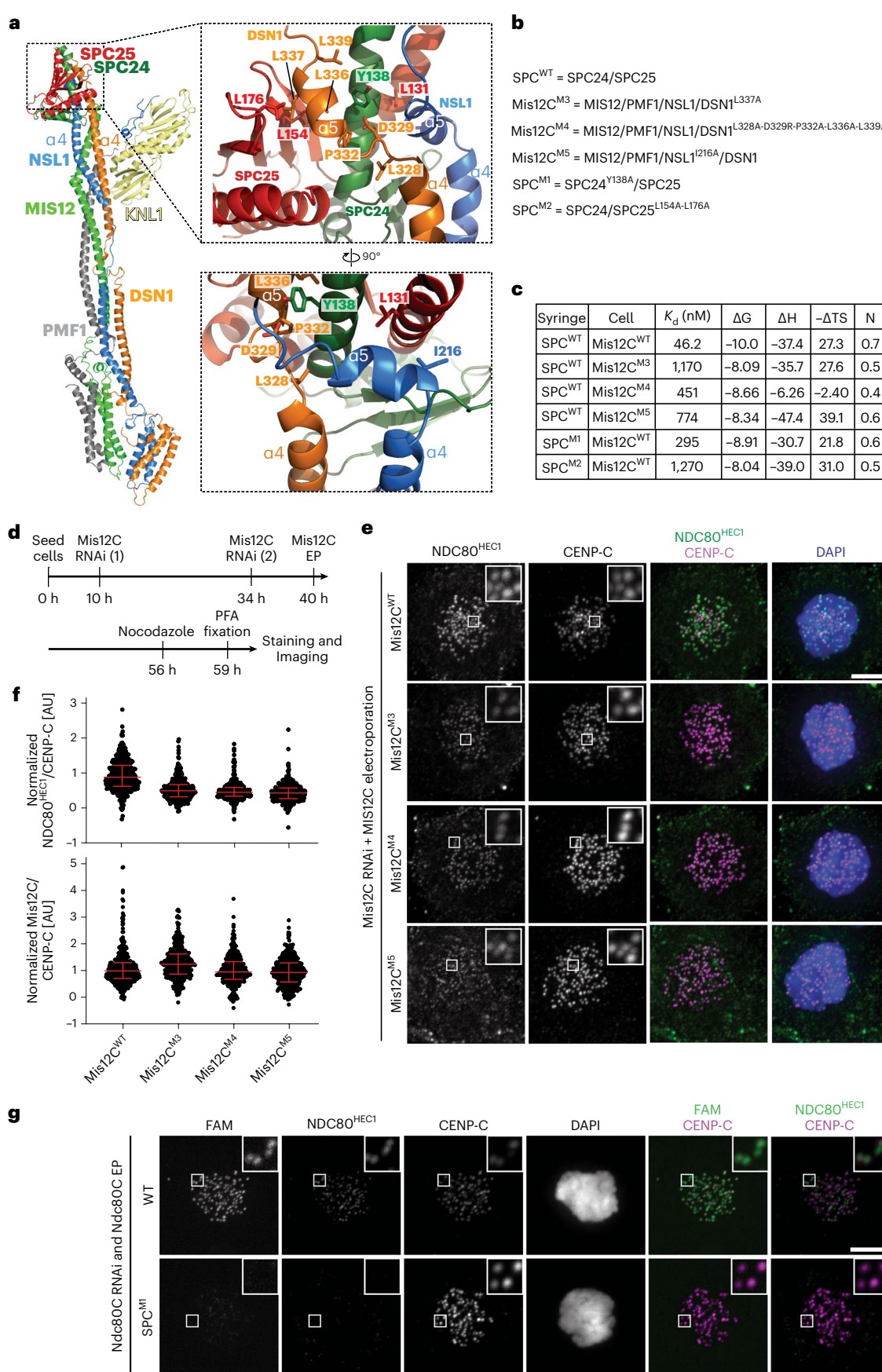

cells previously depleted of endogenous Ndc80C. Both mutants prevented kinetochore recruitment of Ndc80C, contrary to the wild-type complex, which was recruited robustly (Fig. 3g and Extended Data Fig. 6a–c). Thus, collectively, the mutational analysis was consistent with the interactions of the Mis12C with KNL1 and with the SPC24/SPC25 moiety of the Ndc80C revealed by our structural work.

## Mutual influence of KMN subcomplexes on localization

The binding sites on Mis12C for KNL1 and for Ndc80C are clearly demarcated, and KNL1 and Ndc80C do not appear to interact directly. In previous work, however, we observed that individual depletions of KNL1 or Ndc80C had reciprocal effects on kinetochore localization[42]. Indeed, depletion of KNL1 negatively affected the levels of kinetochore NDC80[HEC1] (found to settle at ~70% of control levels). Conversely, depletion of the Ndc80C (quantified in Fig. 4b) caused an even more significant reduction in the levels of kinetochore KNL1 to ~60% of control levels (Fig. 4a; quantified in Fig. 4c). In addition, we found that the levels of the Mis12C (measured through the NSL1 subunit) were also affected by individual depletions of KNL1 or Ndc80C, an effect reinforced by the depletion of both (Fig. 4a,d). Thus, KNL1, Ndc80C and Mis12C appear to influence each other's abundance or stability at the kinetochore. A reduction in the levels of Mis12C upon depletion of Ndc80C or KNL1 has been reported previously[20]. Stabilization by Ndc80C may reflect enhanced accessibility of Mis12C to phosphorylated CENP-T[12,20]. How KNL1 stabilizes Mis12C, on the other hand, is unclear.

To corroborate this conclusion with biochemical experiments, we used bio-layer interferometry (BLI) to measure the binding affinity of the Mis12C for Ndc80C in the presence or absence of KNL1$_{2,026-2,042}$. Mis12C bound Ndc80C with a $K_d$ of ~6 nM (Fig. 4e and Extended Data Fig. 7a–c), which is almost tenfold lower (indicative of tighter binding) than for the interaction of Mis12C with the SPC24/SPC25 pair (Fig. 3c). This result is consistent with the idea that the complete Ndc80C binds Mis12C with somewhat higher affinity than the SPC24/SPC25 subcomplex, as also in agreement with our previous measurements (as discussed above). However, we note that the two $K_d$ measurements were performed with BLI and ITC, respectively, and therefore the differences in binding affinity may also reflect technical differences. We continued to use BLI to measure the strength of the MIS12–Ndc80C interaction in the presence of KNL1$_{2,026-2,342}$. The measured $K_d$ of ~1 nM (Fig. 4e and Extended Data Fig. 7a–c) is consistent with the idea that KNL1 increases the binding affinity of Ndc80C for the Mis12C. The $K_d$ for the Mis12C–KNL1 interaction was ~2 nM, almost identical to the value obtained by ITC (see Fig. 2c). For technical reasons, we could not address whether the addition of Ndc80C increased the binding affinity of the Mis12C–KNL1, an expectation raised by the biological experiments in Fig. 4a–d.

To shed light on the potential mechanism of reciprocal stabilization of KNL1 and Ndc80C through Mis12C, we conducted molecular dynamics simulations (see Methods). For computational efficiency, we considered a reduced fragment ('KMN$^{hub}$') containing a truncated Mis12C helical bundle and the SPC24/SPC25 and KNL1 binding headgroups (Fig. 4f). In three additional sets of simulations, we removed SPC24/SPC25 (ΔNdc80C), KNL1 (ΔKNL1) or both (Mis12C) (Fig. 4f). Each system was simulated five times for 500 ns for a total cumulative sampling of 2.5 µs per system. Despite the truncation, the Mis12C helical bundle of KMN$^{hub}$ was stable and SPC24/SPC25 and KNL1 remained stably bound to it (Supplementary Video 1), with a root mean square deviation (r.m.s.d.) from the initial conformation leveling off at values smaller than 1 nm (Extended Data Fig. 7d). Omission of SPC24/SPC25, KNL1 or both increased the flexibility of the DSN1 and NSL1 segments without significantly impacting MIS12 and PMF1 (Fig. 4f and Extended Data Fig. 7e). Structural fluctuations were maximal in the regions implicated in Ndc80C and KNL1 binding, which were more rigid in association with the cognate ligands and highly flexible upon their removal, providing a tentative molecular explanation for the reciprocal dependence of KNL1 and Ndc80C on localization and binding demonstrated in Fig. 4a–e. Further details and implications of the molecular dynamics analysis are discussed in the Methods.

## Limited plasticity of the Mis12C–Ndc80C connection

Our analysis so far suggests that the extensions of DSN1 and NSL1 implicated in Ndc80C binding do not act as simple docking motifs by and large independent of the rest of the structure. Rather, these regions seem more closely integrated within the Mis12C and may contribute to its stabilization. To further probe this idea, we asked whether these C-terminal regions could be freely grafted onto other Mis12C subunits. To this end, we generated 14 variants of the Mis12C in which the NSL1 and DSN1 C-terminal segments implicated in Ndc80C binding were grafted on MIS12 and PMF1, singly or as a pair (listed in Extended Data Fig. 8a). In size-exclusion chromatography experiments, all 14 Mis12C constructs appear to co-elute with KNL1$_{2,026-2,342}$ (Extended Data Fig. 9). On the other hand, none of the constructs retained the ability to interact with Ndc80C (Extended Data Fig. 9), with the exception of Mis12C$^{M18}$ and Mis12C$^{M19}$ (Fig. 5a–c). In these constructs, grafting residues 227–281 of NSL1 (the NSL1 C-terminal segment downstream from the Ndc80C binding sites) onto either MIS12 or PMF1 was compatible with KNL1 binding. In no case were we able to swap the C-terminal segment of DSN1 or NSL1 encompassing Ndc80C binding regions onto another subunit of the Mis12C and retain binding to Ndc80C (Extended Data Fig. 9).

When electroporated into HeLa cells depleted of endogenous Mis12C (ref. 43), Mis12C$^{WT}$, Mis12C$^{M18}$ and Mis12C$^{M19}$ localized to kinetochores essentially indistinguishably (Fig. 5d and Extended Data Fig. 8b) as expected on the basis of the experiments in vitro showing normal KMN assembly. Mis12C$^{WT}$, Mis12C$^{M18}$ and Mis12C$^{M19}$ were all capable of restoring correct chromosome alignment and timing of mitotic exit, as revealed by their ability to suppress the mitotic arrest

**Fig. 4 | Dynamic binding interdependency in the Mis12–Knl1–Ndc80 complex. a**, Representative immunofluorescence analysis of KNL1 and NSL1 upon RNAi-mediated depletion of Ndc80C subunits or KNL1 (see Methods). Scale bar, 5 µm. This experiment, together with an equivalent experiment carried out to assess the localization levels of Ndc80C and KNL1 under the same RNAi conditions, are quantified in **b**–**d**. **b**–**d**, Quantification of experiments displayed in **a**. Red bars represent median, indicated above the plots, and interquartile range of normalized single kinetochores (n) intensity values. Statistical analysis was performed with non-parametric Mann–Whitney test comparing two unpaired groups. In **b**, reporting normalized NDC80[HEC1] levels, the following number of kinetochores from three independent experiments were analyzed: $n = 3,437$ for Ndc80C RNAi + KNL1 RNAi, $n = 3,637$ for Ndc80C RNAi, $n = 2,520$ for KNL1 RNAi. In **c**, reporting normalized KNL1 levels, the following number of kinetochores from three independent experiments were analyzed: $n = 3,466$ for Ndc80C RNAi + KNL1 RNAi, $n = 2,992$ for Ndc80C RNAi, $n = 2,273$ for KNL1 RNAi.

In **d**, reporting NSL1 levels, the same number of kinetochores was analyzed as in **c**. **e**, Summary of BLI experiments shown in Extended Data Fig. 7a–c. **f**, Minimal fragment (KMN$^{hub}$, black box) containing key interacting partners considered for MD simulations. Mis12C was truncated approximately at the middle of the helical bundle. In addition, the SPC24/SPC25 and KNL1 RWD headgroups of Ndc80C and KNL1, respectively, were considered. Summary of the four systems simulated (shown as spheres). **g**, Change in per-residue root mean square fluctuations (r.m.s.f.) of the systems lacking at least one of the interaction partners relative to the system with both interaction partners bound (Δr.m.s.f. = (r.m.s.f.(X)/r.m.s.f.(KMN$^{hub}$)) − 1) with X indicated at the top of each panel. The change is mapped on the surface of the Mis12C fragment according to the color scale (top left). Residues that displayed an increased (reduced) r.m.s.f. in the absence of a binding partner are colored in red (cyan). The transparent regions with black outlines indicate the binding positions of the SPC24/SPC25 (ΔNdc80C) and KNL1 domains (ΔKLN1), or both (Mis12C).

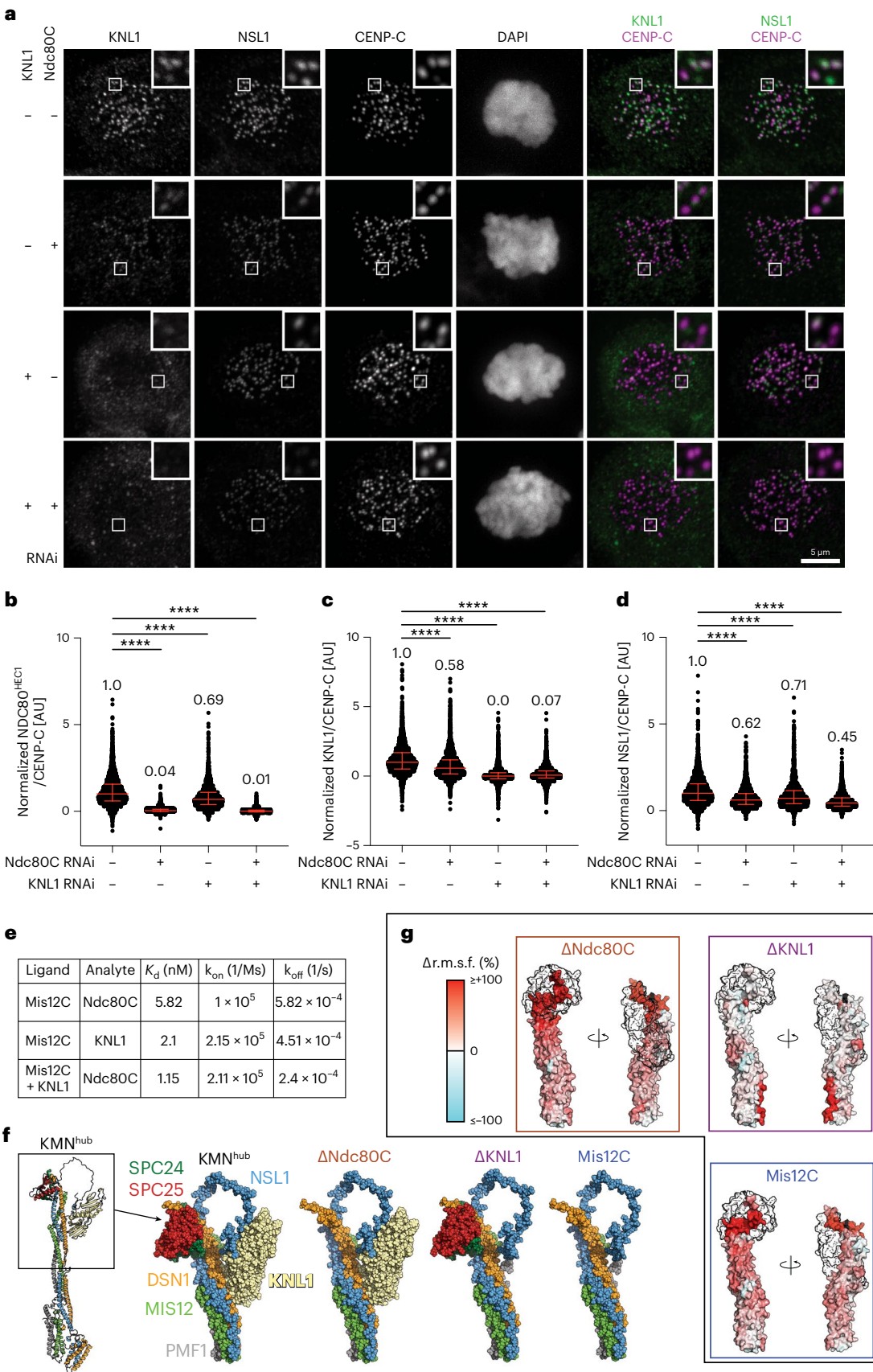

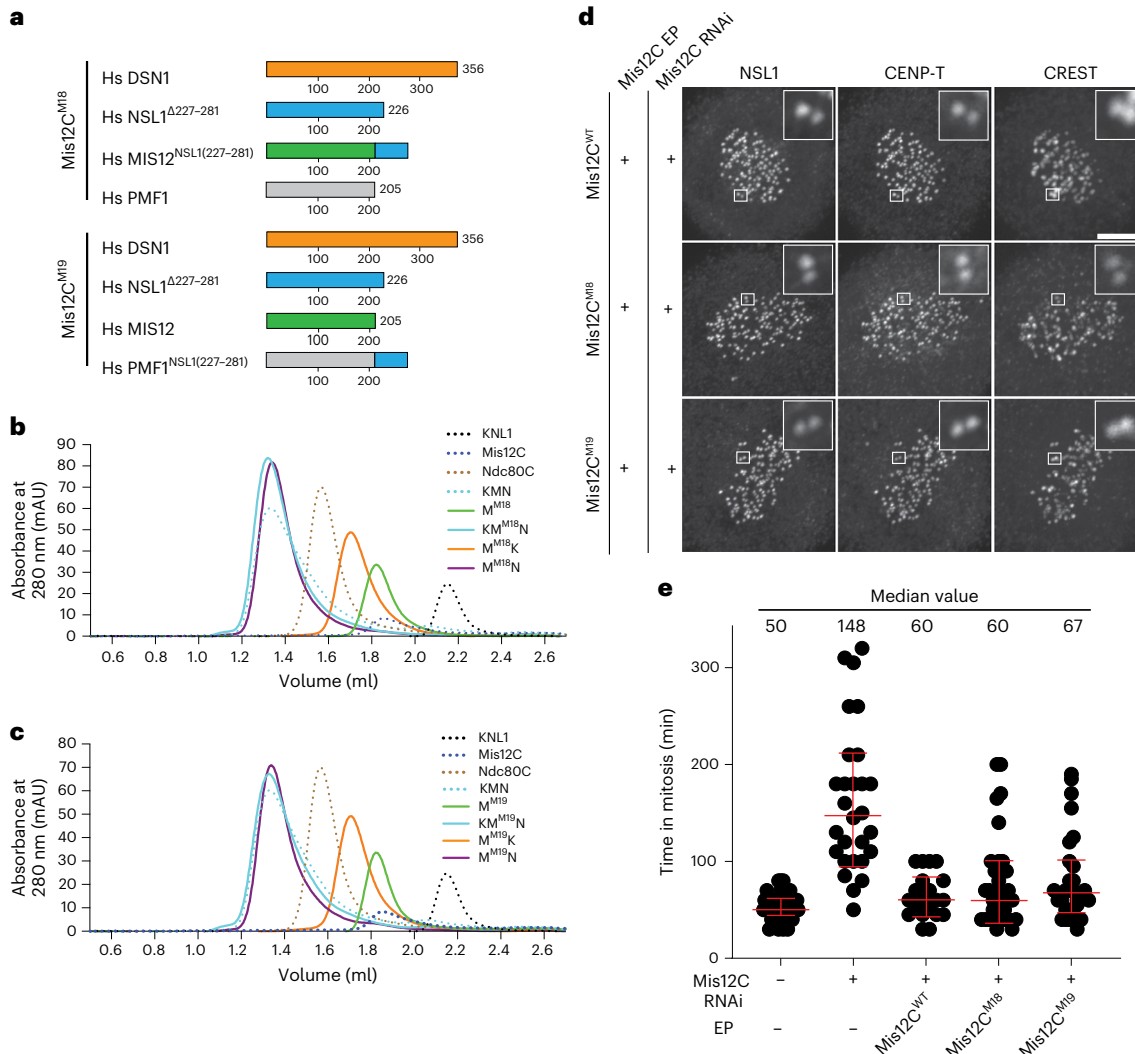

**Fig. 5 | Swapping of Mis12C C-terminal tails. a**, Schematic representation of the Mis12C$^{M18}$ and Mis12C$^{M19}$ constructs. The complete list of Mis12C constructs is presented in Extended Data Fig. 8a and size-exclusion chromatography profiles in isolation or with potential binding species are presented in Extended Data Fig. 9a–m. **b,c**, Size-exclusion chromatography of the indicated biochemical species on a Superdex S200 5/150 column for Mis12C$^{M18}$ (**b**) and Mis12C$^{M19}$ (**c**). **d**, Representative images of cells depleted of endogenous Mis12C and electroporated with the indicated recombinant Mis12C species from three independent experiments. Scale bar, 5 μm. **e**, Time from nuclear envelope breakdown to chromosome decondensation in H2B-mCherry HeLa cells. Live-cell movies were performed for 16 h. Each black dot represents a single cell. Median values with interquartile range of mitotic exit time are reported from $n = 45$ cells for control, $n = 28$ cells for Mis12C RNAi, $n = 20$ cells for Mis12C$^{WT}$, $n = 33$ cells for Mis12C$^{M18}$ and $n = 30$ cells for Mis12C$^{M19}$ from three independent experiments.

observed in cells depleted of endogenous Mis12C (Fig. 5e). Thus, the segment of the NSL1 C-terminal tail downstream from the SPC24/SPC25 binding site (that is, residues 227–281) can be grafted onto other Mis12C subunits without significant disruptions, an observation that suggests that communication between the binding sites for Ndc80C and KNL1 does not involve the NSL1 C-terminal region but rather the helical stalk, as also suggested by the molecular dynamics simulations.

## Structure and role of ZWINT

Density for ZWINT and for the KNL1 region preceding the RWD$^{N–C}$ domains was not visible in our maps, even at low map contouring levels. We therefore opted to model the ZWINT–KNL1 complex using AF2 Multimer[41]. The resulting model contains all ZWINT residues and residues 1,880–2,342 of KNL1 (KNL1$_{1,880–2,342}$) (Fig. 6a and Extended Data Fig. 10) and shows high reliability scores (Supplemental Fig. 1a,e). The model consists of tightly intertwined ZWINT and KNL1 helices (Fig. 6a). The model predicts ZWINT$^{76–220}$ and KNL1$_{1,960–2,133}$ to engage in a bent

parallel coiled coil that terminates into the KNL1 C-terminal globular domain. This coiled coil is propped up by additional interactions with N-terminal fragments of both proteins so that the coiled coil's initial segment is incorporated into a reinforced four-helix bundle with short additional connecting helices. We used AF2 to carry out additional predictions of the Knl1C, including amphibian, avian and yeast complexes (Extended Data Fig. 10a–d). Amphibian and avian Knl1 complexes are predicted to be closely related to the human complex. In both complexes, however, ZWINT is shown to have tandem RWD domains similar to those of KNL1, which appear to have been lost in humans. These observations strongly suggest that ZWINT and KNL1 are paralogs that have undergone significant functional and structural diversification after duplication from a common singleton. Yeast ZWINT (Kre28) is predicted to have a single RWD domain, and the overall appearance of its complex with KNL1 (Spc105) is much more compact relative to the other three modeled complexes (see also ref. 44) (Extended Data Fig. 10d).

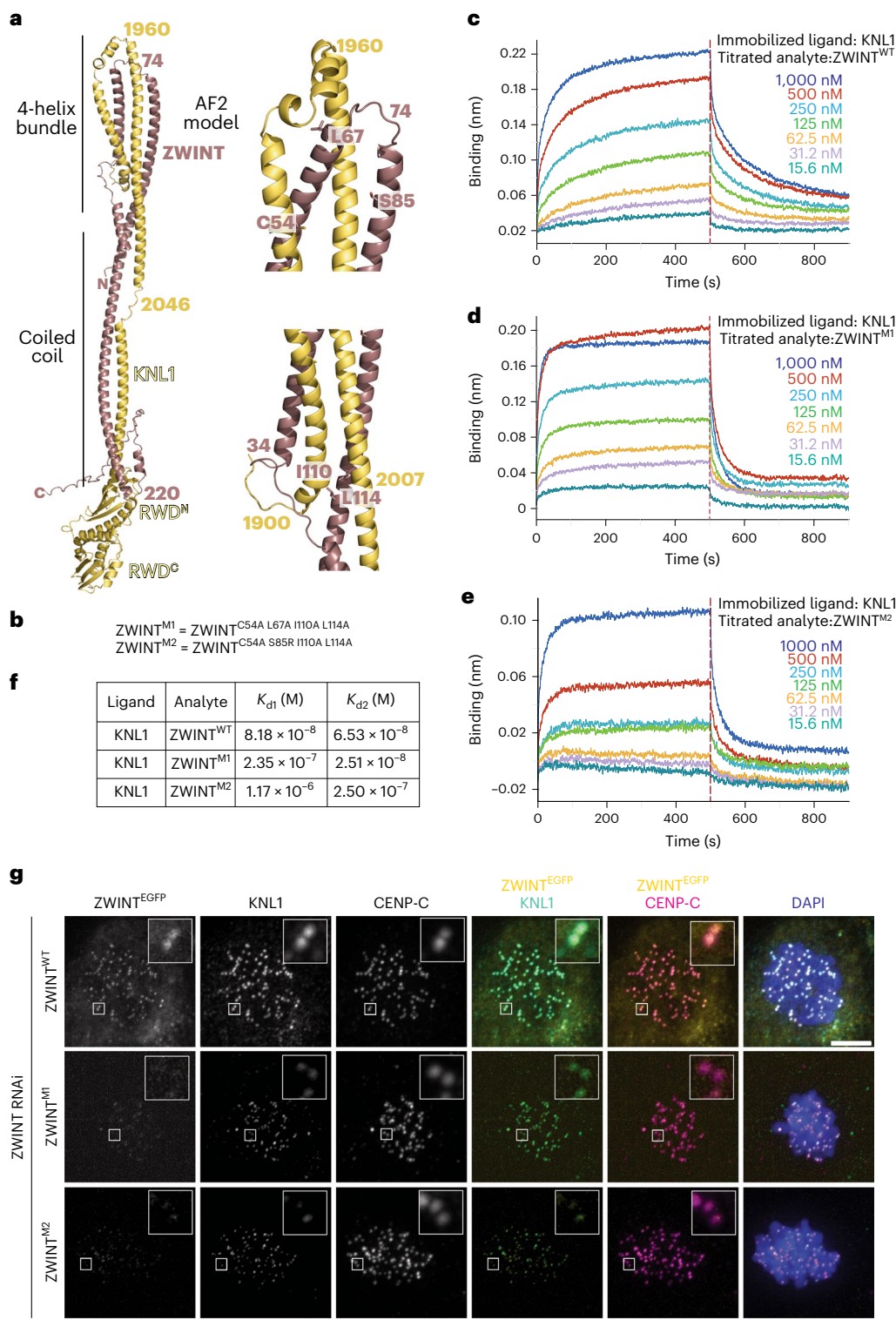

**Fig. 6 | Model of the KNL1–ZWINT interaction and its validation. a**, AF2 model of the complex of KNL1[1,880–2,342] and ZWINT with closeups showing position of complex-stabilizing residues. **b**, Table describing two ZWINT mutants at interface residues. **c–e**, BLI traces for titration experiments with the indicated species for ZWINT[WT] (**c**), ZWINT[M1] (**d**) and ZWINT[M2] (**e**). **f**, Summary of binding titrations. **g**, Representative images of HeLa cells depleted of endogenous ZWINT and expressing GFP fusions of the indicated ZWINT constructs from a stably integrated transgene. Quantifications of these experiments are displayed in Extended Data Fig. 10i–j. Scale bar, 5 μm.

Based on the AF2 model of human Knl1C, we generated two ZWINT mutants (ZWINT[M1] and ZWINT[M2]) harboring point mutations predicted to destabilize the interface with KNL1 (Fig. 6b). BLI measurements were best fitted by a two-site binding model and demonstrated that both mutants affected KNL1 binding (Fig. 6c–f). Depletion of Ndc80C caused similar reductions in the kinetochore levels of ZWINT and KNL1 (Extended Data Fig. 10e,f). Similarly, depletion of ZWINT (Extended Data Fig. 10f) caused a significant reduction in the kinetochore levels

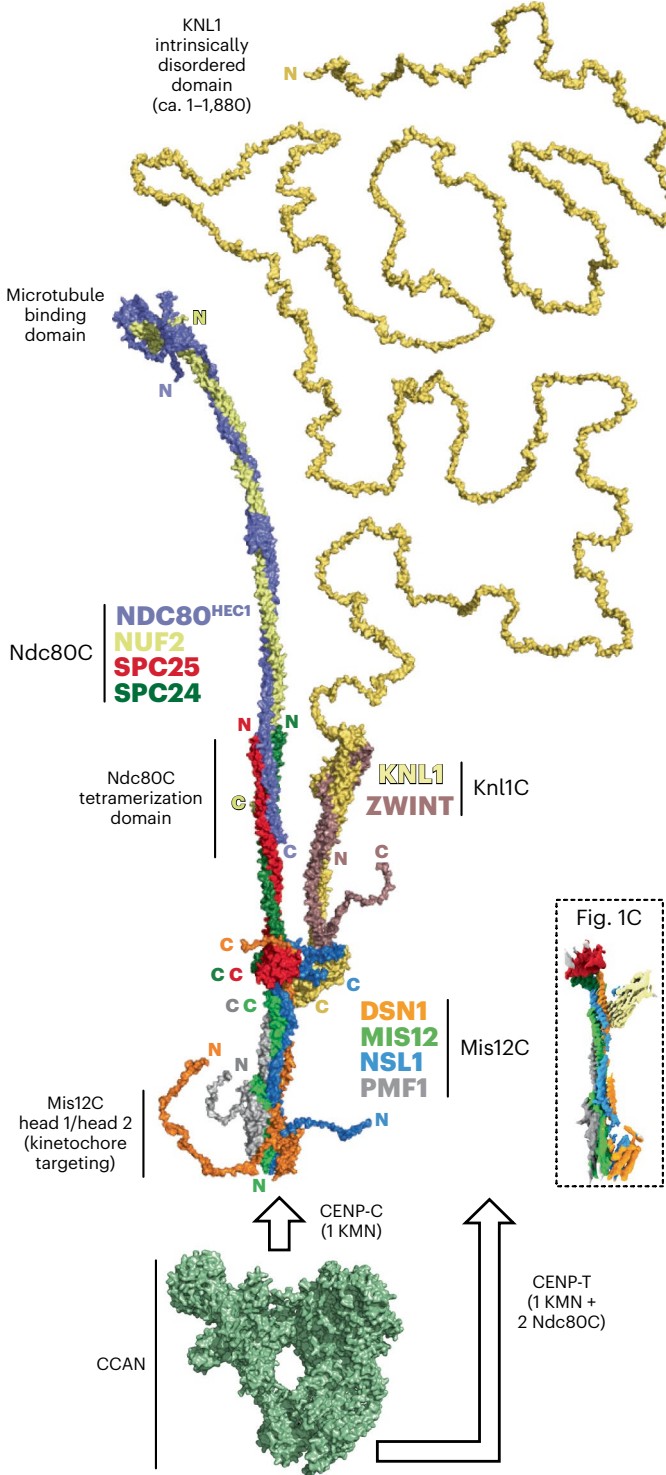

**Fig. 7 | Model of the complete human KMN network.** Surface representation of the human KMN hybrid model discussed in this paper (see Methods for description of approaches behind construction of the model and Supplemental Fig. 1 for AF2 quality scores). Each subunit is labeled with its color, and for each chain, the position of the N terminus and C terminus is indicated. The intrinsically disordered segment of KNL1 is included for reference. Shorter disordered tails are also present at the N termini of NDC80[HEC1], PMF1, DSN1 and NSL1 as well as the C termini of DSN1 and ZWINT. A surface representation of the 16-subunit CCAN complex (PDB 7QOO, displayed in green)[50] is also shown for reference and roughly in scale. In humans, CCAN provides a point of attachment to the head 1 of Mis12C through CENP-C (Extended Data Fig. 2). One CCAN recruits an entire KMN complex and also provides a second point of attachment to an entire KMN through CENP-T. Human CENP-T recruits two additional Ndc80C complexes through direct binding sites. See main text for details.

of KNL1 (Extended Data Fig. 10a), suggesting KNL1 destabilization in the absence of ZWINT. The levels of Ndc80C were also affected (Extended Data Fig. 10g), as expected based on the results in Fig. 4a,b. When expressed from a stable transgene in cells depleted of endogenous ZWINT (Extended Data Fig. 10h), an ectopically expressed fusion of ZWINT[WT] with EGFP localized robustly to kinetochores, whereas similar constructs expressing ZWINT[M1] and ZWINT[M2] demonstrated very significantly reduced localization (Fig. 6g; quantified in Extended Data Fig. 10i,j). These reduced levels of ZWINT also correlated with lower levels of endogenous KNL1, suggesting that KNL1 and ZWINT are interdependent for kinetochore localization. These results validate the AF2 model and indicate that robust binding to KNL1 is necessary for ZWINT localization.

### A model of the KMN network

Starting from the minimal core structure of the KMN network, we generated a model of the KMN network by incorporating ZWINT, the entire KNL1 subunit and all Ndc80C subunits[45], with an overall molecular mass of 593 kDa (Fig. 7). In the absence of its known partners, KNL1 is predicted to be disordered from its N terminus up to the point of entry into the helical domain (at residue 1,880). The contour length of the disordered segment is predicted to be ~700 nm (ref. 46), several times longer than the predicted length of Ndc80C of ~65 nm. A striking feature of the model is the prediction that the helical moiety of the Knl1C branch and the SPC24/SPC25 coiled coil emerge from the Mis12C essentially in parallel. Despite the absence of corroborating experimental information in our study, this conclusion seems reliable, as AF2 predictions of amphibian and avian Knl1C converge on similar models, in which the angle of departure of the Knl1C coiled coil from the RWD domains is largely fixed (although predictions of the avian complex also show bent conformations) (Extended Data Fig. 10a–d). This conclusion is also supported by experimental evidence in an independent accompanying study of the human outer kinetochore[47]. Our structure may have failed to capture this feature because of the use of Knl1$_{2,021-2,342}$, a KNL1 fragment that binds ZWINT robustly but may fail to fully stabilize the KNL1–ZWINT complex (refer to Fig. 6a). The KMN network model suggests that the ends of the SPC24/SPC25 dimer and the beginning of the folded domain of the Knl1C may, with some likelihood, face each other at a relatively short distance. The functional implications of this configuration will have to be carefully analyzed in future work. We speculate that it may be related to spindle assembly checkpoint signaling, given the importance of Ndc80C and KNL1 in the recruitment of proteins involved in this pathway.

As already briefly mentioned, AF2 modeling predicts that the terminal segment of the ZWINT–KNL1 coiled coil may contribute to the stabilization of the NSL1 C-terminal tail poised to interact with the KNL1 RWD[N] domain (Extended Data Fig. 10k). Thr242 in this region of NSL1 has been recently shown to be a phosphorylation target of cyclin-dependent kinase, which might destabilize the interaction of Knl1C with the kinetochore[48].

### Conclusions

We report a comprehensive model of the human KMN network, the microtubule-binding apparatus of the kinetochore and a critical platform for mitotic checkpoint and error-correction activities. Our conclusions substantially agree with, and are complementary to, those of an independent co-submitted study[47]. The model sheds light on several fundamental aspects of KMN structure and function. First, it suggests a structural mechanism for the stabilization of an autoinhibited state of Mis12C by DSN1 and for switching to an open conformation by phosphorylation, thereby facilitating CENP-C binding. Second, the model reveals the molecular basis of the interactions of Knl1C and Ndc80C with Mis12C, which we further validated experimentally. Our experiments and simulations further indicated that each of the interaction

interfaces between KMN subcomplexes leads to a collective, cooperative stabilization of KMN. Third, the model indicates that ZWINT and KNL1 are paralogs. An AF2 prediction indicates that the helical domain of KNL1 begins around residue 1,880, and therefore earlier than the first residue of the KNL1 construct included in our experimental electron microscopy reconstruction. The KNL1 fragment that we used for experimental structure determination is thermodynamically stable, monodisperse and interacts robustly with ZWINT. However, the model predicts that the use of this fragment might have prevented the assembly of the four-helix bundle (Fig. 6a), possibly rendering this region of the structure more flexible and disordered. The intertwined helices of KNL1 and ZWINT form an elongated domain that AF2 predicts to emerge in parallel to that of SPC24/SPC25. AF2 also predicts that the helical domain of Knl1C contributes to the stabilization of the interaction with NSL1. The predicted proximity of the SPC24 and SPC25 and the KNL1 and ZWINT coiled coils also supports the speculation that Knl1C and Ndc80C may cooperate in providing points of contact for checkpoint proteins and other microtubule binders. Our future biochemical investigations will try to investigate this possibility.

In humans, CENP-C and CENP-T each have the ability to recruit one KMN complex (through interactions with Mis12C) to the kinetochore. Furthermore, the N-terminal region of human CENP-T provides binding sites for two additional isolated Ndc80C complexes[12]. Thus, each CCAN complex is collectively predicted to recruit two KMN assemblies (each carrying Ndc80C and Knl1C) and two additional Ndc80Cs (that is, four Ndc80C per CCAN). Microtubule binding appears to promote the assembly of Ndc80Cs into small clusters with enhanced potential to capture forces generated by depolymerizing microtubules[45,49]. Our model of the KMN network provides a framework to understand the high-order organization of the kinetochore in future studies.

## Online content

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

## Methods

### Cloning, expression, and purification of KNL1, MIS12, SPC24/SPC25 and ZWINT

All mutants in this study were generated by PCR-based site-directed mutagenesis. KNL1 constructs were expressed in *E. coli* BL21(DE3)-Codon-plus-RIPL in Terrific Broth supplemented with chloramphenicol, ampicillin, and 0.2 mM IPTG at 18 °C for 16 h. Cells were pelleted down, washed once in PBS and stored at −80 °C for future use. All purifications were carried out on ice or at 4 °C unless stated. For purification, frozen cells were removed from −80 °C storage, thawed and resuspended in GST lysis buffer (50 mM HEPES pH 8.0, 150 mM NaCl, 2 mM TCEP, 10% glycerol, 0.5 mM PMSF, DNase I, protease inhibitor mix HP plus (Serva)). Cells were lysed by sonication and centrifuged at 108,000×*g* for 1 h. The cleared lysate was mixed with 5 ml glutathione-agarose beads (5 ml of resin for 2 l of expression) and incubated on a roller overnight. The next day, beads were washed with 30 column volumes of wash buffer (lysis buffer without DNase and protease inhibitors). Beads were resuspended in wash buffer containing tobacco etch virus (TEV) protease and incubated overnight. Flow-through was passed through the Ni-beads to trap TEV protease and was dialyzed in GST SEC buffer (50 mM CHES pH 9.0, 50 mM NaCl, 2 mM TCEP, 5% glycerol). The dialyzed protein was concentrated and injected into a HiLoad S75 16/600 column (GE Healthcare) pre-equilibrated with GST SEC buffer. Elution fractions were analyzed through Tricine-SDS–PAGE, and relevant fractions were pooled, concentrated, flash-frozen in liquid nitrogen and stored at −80 °C. All mutants of ZWINT were purified using the same buffers, conditions and columns of KNL1 purification, except that 3C protease was added to remove the GST tag instead of TEV protease. The pBIG1A vector of the Mis12C was generated by combining expression cassettes of MIS12, PMF1, NSL1 and DSN1[SORT-HIS]. Bacmids from this vector were transfected in Sf9 insect cells (Gibco, cat. no. 11496015) to generate baculoviruses. Protein expression was carried out in Tnao38 insect cells. Cells were collected after 72 h of transfection and stored at −80 °C. For protein purification, cells were resuspended in Talon lysis buffer (50 mM HEPES pH 8.0, 400 mM NaCl, 20 mM imidazole pH 8.0, 2 mM TCEP, 10% glycerol, 0.5 mM PMSF, DNase I, protease inhibitor mix HP plus (Serva)), lysed by sonication and centrifuged at 100,800×*g* for 1 h to clear the lysate. The lysate was filtered and added to 15 ml of Talon beads pre-equilibrated with Talon lysis buffer and then incubated overnight. The beads were washed with 15 column volumes of Talon wash buffer I (lysis buffer without DNase and protein inhibitors) and then three column volumes of Talon wash buffer II (50 mM HEPES pH 8.0, 400 mM NaCl, 30 mM Imidazole, pH 8.0, 2 mM TCEP, 10% glycerol). Then, 20 ml of Talon elution buffer (Talon wash buffer I + 300 mM imidazole) was added to beads and incubated for 2 h. Flow-through was collected, concentrated and injected into HiLoad 16/600 Superose 6 column (GE Healthcare) pre-equilibrated with Mis12C SEC buffer (50 mM HEPES pH 8.0, 250 mM NaCl, 2 mM TCEP, 5% glycerol). Elution fractions were analyzed by Tricine-SDS–PAGE and relevant fractions were pooled together, concentrated, flash-frozen in liquid nitrogen and stored at −80 °C. SPC24/SPC25 and its mutants were expressed in BL21(DE3)-codon-plus-RIPL cells by induction of 0.4 mM IPTG at 18 °C for 16 h. Cells were collected, washed once in PBS and stored at −80 °C. Frozen pellets were resuspended in SPC lysis buffer (50 mM HEPES pH 8.0, 500 mM NaCl, 2 mM TCEP, 10% glycerol, 1 mM EDTA, 0.5 mM PMSF, protease inhibitor mix HP plus (Serva)), lysed by sonication and the lysate cleared by centrifugation. The cleared lysate was added to glutathione-agarose beads pre-equilibrated with SPC wash buffer (SPC lysis buffer without DNase I and protease inhibitor) and incubated overnight. Beads were extensively washed with wash buffer and were incubated in SPC wash buffer containing 3C protease to cleave off the GST. Flow-through was collected, concentrated and loaded into HiLoad S75 16/600 column (GE Healthcare) pre-equilibrated in SPC SEC buffer (50 mM HEPES pH 8.0, 250 mM NaCl, 2 mM TCEP, 5% glycerol). Relevant elution fractions were pooled, concentrated, flash-frozen in liquid nitrogen and stored at −80 °C.

### Biotin and fluorescent labeling of proteins

C-terminal peptide conjugations in all the proteins were carried out by using calcium-independent sortase 7M enzyme. The synthetic peptide used for biotinylation was GGGG[Biotin] (Thermo Fisher Scientific) and for FAM labeling was GGGGK[FAM] (Genscript). Reactions were performed at 4 °C overnight with sortase:protein:peptide in an approximately 1:10:100 ratio in SEC buffer of relevant proteins. Excess peptide and sortase were removed from the labeled protein by running the whole mixture on a size-exclusion chromatography column.

### Cell culture, siRNA transfection and immunoblotting

HeLa cells expressing mCherry-H2B[51] and DLD-1 Flp-In-T-REx[52] cell lines were cultured in DMEM media (PAN Biotech) supplemented with 10% FBS (Clontech) and 2 mM L-glutamine (PAN Biotech) (DMEM complete media) in a humidifier chamber maintained at 37 °C in the presence of 5% CO$_2$. Cell lines were regularly tested for mycoplasma contamination. The following siRNAs were used in this study: siMis12C (Dharmacon siMIS12 5′-GACGUUGACUUUCUUUGAU-3′; Sigma siDSN1 5′-GUCUAUCAGUGUCGAUUUA-3′, siNSL1 5′-CAUGAGCUCUUUCUGUUUA-3′); siKNL1 (Invitrogen HSS 183683 5′-CACCCAGTGTCATACAGCCAATATT-3′, HSS125942 5′-TCT ACTGTGGTGGAGTTCTTGATAA-3′, HSS125943 5′-CCCTCTGGA GGAATGGTCTAATAAT-3′); siZWINT (Sigma, 5′-GCACGUAGA GGCCAUCAAA-3′); siNdc80C (Sigma siNDC80[HEC1] 5′-GAG TAGAACTAGAATGTGA-3′, siSPC24 5′-GGACACGACAGTCACAATC-3′, siSPC25 5′-CTACAAGGATTCCATCAAA-3′). RNAiMAX (Invitrogen) was used as a transfection reagent, and transfections of siRNAs were carried out according to the manufacturer's guidelines. For immunoblotting, cells were thawed on ice and resuspended in blot lysis buffer (75 mM HEPES pH 7.5, 150 mM KCl, 1.5 mM EGTA, 1.5 mM MgCl$_2$, 10% glycerol, 0.075% NP-40 and protease inhibitor cocktail (Serva)). Cells were lysed by vigorously pipetting and were centrifuged at 16,000×*g* for 30 min at 4 °C to clear lysate. Lysates were run in NuPAGE 4–12% Bis-Tris gradient gels and then proteins were transferred to PVDF membranes for further analysis. The following antibodies were used in this study for immunoblotting: anti-GFP (rabbit, custom-made EIO antibody facility, Milan (Italy), 1:1000), anti-Vinculin (mouse monoclonal, clone hVIN-1, Sigma-Aldrich, 1:10,000), anti-rabbit-HRP (Amersham NA934, 1:10,000) and anti-mouse-HRP (Amersham NXA931-1ML, 1:10,000).

### Mammalian cell electroporation

All electroporation experiments in this study were performed by using a Neon Transfection Kit (Thermo Fisher Scientific). For electroporation, cells were collected after being trypsinized. Cells were then washed with PBS, counted and split between two and three million. Cells were washed once in PBS and resuspended in 90 μl of proprietary buffer R. The Mis12C complex and its mutants were taken at a concentration of 40 μM for electroporation. Proteins were tawed and centrifuged at 16,000×g at 4 °C for 10 min. Proteins were diluted with buffer R in a 1:1 ratio. Cells and proteins were mixed together and loaded into a 100 μl Neon Pipette Tip. The Pipette Tip was mounted to the electroporation adaptor chamber and cells were electroporated at 1,000 V with two consecutive pulses of 35 ms. Cells were washed once in PBS, trypsinized to remove excess cell surface-bound proteins and resuspended in DMEM complete media. Cells were then plated in six-well plates with coverslips inside them for immunofluorescence analysis or in 24-well plates (Ibidi) for live-cell imaging.

### Generation of stable mammalian cell lines

Flp-In-T-REx DLD-1 osTIR1 cell lines were used to generate stable cell lines. Sequences encoding KNL1$_{2,026–2,342}$ and ZWINT wild-type and mutants were cloned into a pCDNA5/FRT/TO-EGFP-IRES plasmid and

co-transfected with pOG44 (Invitrogen), encoding the Flp recombinase, into freshly seeded cells using X-tremeGENE (Roche) transfection reagent. To obtain positive clones, cells were selected in DMEM complete media supplemented with hygromycin B (250 µg ml$^{-1}$; Thermo Fisher Scientific) and blasticidin (4 µg ml$^{-1}$; Thermo Fisher Scientific) for 2 weeks. Positive clones were pooled, expanded and expressions of transgenes were verified by immunofluorescence and immunoblot analyses. Gene expression was induced by the addition of 0.3 µg ml$^{-1}$ of doxycycline (Sigma-Aldrich) for 24 h.

### Cell treatment, microscopy, immunofluorescence detection and live-cell imaging

For immunofluorescence analysis, cells were treated with Nocodazole (3.3 µM) or Nocodazole and MG132 (10 µM) for 3 h before fixing them for staining. Cells were permeabilized with 0.5% Triton-X-100 in PHEM buffer and fixed on coverslips with 4% paraformaldehyde. Cells were then blocked with 5% boiled donkey serum and stained with primary antibodies in a humidified chamber at room temperature (22–25 °C) for 2 h. Cells were washed with 0.1% Triton-X-100 in PHEM buffer to remove excess primary antibodies and stained with secondary antibodies at room temperature for 1 h. DNA was stained with DAPI and coverslips were mounted on slides with Mowiol. The following primary and secondary antibodies were used in this study: anti-NSL1 (mouse monoclonal, QL24-1 (25), in-house, 1:800), anti-CENP-C (guinea pig polyclonal, MBL-PD030, MBL, 1:1,000), anti-HEC1 (human NDC80, mouse monoclonal, ab3613, Abcam, 1:3,000), anti-CENP-T (rabbit polyclonal, SI0822, in-house, 1:1,000), anti-KNL1 (rabbit polyclonal, SI0788, 1:500), goat anti-guinea pig Alexa Fluor 647 (Invitrogen A-21450), goat anti-mouse Alexa Fluor 488 (Invitrogen A-11001, 1:200), goat anti-mouse Rhodamine Red (Jackson Immuno Research 115-295003, 1:200), donkey anti-rabbit Alexa Fluor 488 (Invitrogen A-21206, 1:200), donkey anti-rabbit Rhodamine Red (Jackson Immuno Research 711-295-152, 1:200) and goat anti-human Alexa Fluor 647 (Jackson Immuno Research 109-603-003, 1:200). For immunofluorescence imaging, cells were imaged with a customized 3i Marianas system (Intelligent Imaging Innovations) equipped with an Axio Observer Z1 microscope chassis (Zeiss), a CSU-X1 confocal scanner unit (Yokogawa Electric Corporation), Plan-Apochromat ×100/1.4 NA objective (Zeiss) and an Orca Flash 4.0 sCMOS camera (Hamamatsu). Images were taken as a Z-section, one channel at a time, using Slidebook software. For live-cell imaging, cells were plated on 24-well imaging plates (Ibidi), and 1 h before starting movies, complete DMEM media was replaced with CO$_2$-independent L-15 imaging media containing 10% FBS, with or without drugs. Images were acquired using a DeltaVision Elite System (GE Healthcare) equipped with an IX-71 inverted microscope (Olympus), a UPlanFLN 40×1.3 NA objective (Olympus) and a pco.edge sCMOS camera (PCO-TECH) at 37 °C. Live-cell images were taken as a Z-scan starting from the bottom of the plate, one channel at a time per Z at 10 min intervals for 16 h. The softWoRx software was used to set up movies and further analysis. All images were visualized and processed using Fiji (v. 2.0.0)[53]. Quantifications of intensities of protein signals were calculated using a custom-written script, using the DAPI channel as a masking boundary. All protein signals were normalized to either CENP-C or CREST. To quantitate time in mitosis, 16 h movies were analyzed manually. Cells entering the mitosis were taken as entry points and cells exiting mitosis were taken as end points. The time differences between entry and exit points are represented as time in mitosis in the analysis in Fig. 5e.

### ITC

All ITC experiments were carried out using a PEAQ-ITC microcalorimeter (Malvern Preanalytical). All proteins were dialyzed overnight at 4 °C in ITC buffer (50 mM HEPES pH 8.0, 150 mM NaCl, 1 mM TCEP) to correct buffer mismatches. A typical ITC experiment consists of 19 injections of protein from the syringe, with the first injection of 0.4 µl

followed by 18 injections of 2 µl, with a spacing of 150 s to allow the heat change curve to hit the baseline again. Binding isotherms were integrated and corrected for offset, and the data were fitted in a one-site binding equation in Microcal PEAQ-ITC analysis software.

### BLI

All BLI experiments were carried out on a FORTÉBIO BLI instrument. All proteins were dialyzed in BLI dialysis buffer (50 mM HEPES pH 8.0, 150 mM NaCl, 2 mM TCEP, 0.05% Tween-20) to reduce buffer mismatch. Experiments were performed at 25 °C by using Octet SA Biosensors. Biotinylated proteins (ligands) were loaded on sensors and different concentrations of analytes were used for binding to the bound ligand. The resulting curves were fitted either in a 1:1 or 2:1 binding model to obtain association constants.

### KMN reconstitution

KNL1$_{2,021–2,342}$, ZWINT, Mis12C and SPC24$_{109–197}$/SPC25$_{92–224}$ were mixed together in equimolar concentration in KMN reconstitution buffer (50 mM HEPES pH 8.0, 150 mM NaCl, 2 mM TCEP pH 8.0) on ice for at least 1 h. The sample was injected into an S200 5/150 column (GE Healthcare) pre-equilibrated with KMNZ reconstitution buffer, and the peak fraction was taken for structural analysis.

### Analytical size-exclusion chromatography

Proteins were mixed in equimolar concentrations, incubated on ice for at least 2 h, centrifuged at 16,000×g for 20 min at 4 °C and then injected into a Superose 6 5/150 increase column (GE Healthcare) pre-equilibrated with analytical size-exclusion chromatography buffer (50 mM HEPES pH 8.0, 150 mM NaCl, 2 mM TCEP). Typically, fractions of 80 µl were collected at a manufacturer's recommended flow rate and analyzed by running in SDS–PAGE.

### Grid preparation and cryo-EM data acquisition

Grids were prepared using a Vitrobot Mark IV (Thermo Fisher Scientific) at 13 °C and 100% humidity. A total of 4 µl of KMN complex at 0.3 mg ml$^{-1}$ supplemented with 0.003% Triton was applied to glow-discharged UltrAuFoil R2/2 grids (Quantifoil). Excess liquid was removed by blotting (3.5 s at blot force −3), followed by vitrification in liquid ethane. Cryo-EM data were recorded using the EPU software on a Cs-corrected Titan Krios G2 microscope equipped with a field emission gun. In total, 9,896 micrographs were acquired on a K3 camera (Gatan) in super-resolution mode at a nominal magnification of ×105,000, corresponding to a physical pixel size of 0.68 Å per pixel and super-resolution pixel size of 0.34 Å per pixel, respectively (Extended Data Fig. 1a). A total of 63.9 e$^-$ Å$^{-2}$ was distributed over 60 frames. The Bioquantum energy filter (Gatan) zero-loss slit width was set to 15 eV and the defocus range from −1.5 µm to −3.0 µm. Details on data acquisition and processing can be found in Table 1.

### Processing of cryo-EM data

Data were pre-processed on the fly in TranSPHIRE (v.1.4)[54], including motion correction and dose weighting using MotionCorr2 (v.1.3.0) (ref. 55), estimation of CTF parameters using CTFFIND4 in movie mode[56] and particle picking in crYOLO (v.1.7) using a neuronal network trained on 20 representative micrographs of the dataset[57]. A total of 2,925,200 particles were picked by crYOLO, and 1,792,983 twofold binned particles that had a crYOLO confidence value of 0.2 or higher were extracted in SPHIRE[58] with a box size of 256 pixels (Extended Data Fig. 1b). Two-dimensional (2D) classification was performed in CryoSPARC 4.2.1 (ref. 59), and 582,795 particles associated with high-quality 2D classes were selected for 3D refinement in RELION 3.1.2 (refs. 60,61) using a reference volume created from an independent dataset that was acquired using similar imaging settings but with a volta phase plate to increase contrast. This initial 3D refinement reached 7.1 Å resolution. Particle polishing was performed in RELION and all further steps

were carried out in CryoSPARC. Polished particles were subjected to a second round of 2D classification. The 430,553 particles assigned to high-quality 2D classes were subjected to heterogeneous refinement using one good and three junk models; 230,597 particles were sorted into the class associated with the good model. Those were refined to a nominal resolution of 4.5 Å (Extended Data Fig. 1b,c), although the reconstruction does not extend to this resolution in all directions owing to heavily preferred orientations that could not be alleviated by 2D class rebalancing or similar approaches (Extended Data Fig. 1d). DeepEMhancer[62] was used to create the final map that is displayed in the figures (Extended Data Fig. 1b).

## Molecular modeling

AF2 was used for all molecular modeling in AF2 Multimer 3.2.1 (ref. [41]). Given that AF2 Multimer tends to artificially bend extended coiled-coil models even more than the original AF2 (ref. [40]), the models comprising the Ndc80C were predicted in segments. For the KMN model, three subcomplexes were predicted—model 1, Ndc80C N-terminal region with $NDC80_{1-530}$ and $NUF2_{1-368}$; model 2, Ndc80C with $NDC80_{312-C}$, $NUF2_{476-C}$, SPC24 and SPC25; and model 3, Mis12c complex with ZWINT$_{FL}$, $KNL1_{1,880-C}$ (isoform 1 numbering) and SPC24/SPC25. These three models were stitched together using $NDC80_{1-498}$ and $NUF2_{1-337}$ of model 1; $NDC80_{499-642}$, $NUF2_{338-464}$, $SPC24_{1-97}$ and $SPC25_{1-91}$ of model 2; and $SPC24_{98-197}$, $SPC25_{92-224}$ and all remaining chains from model 3. Thus, all relevant interfaces are predicted within a single AF2 model. The fragments were superimposed with PyMOL Molecular Graphics System (version 2.5.4; Schrödinger), and the stitching points within the alpha-helices were relaxed using alpha-helical geometry restraints within COOT (v.0.9.8.2)[63]. In the final model of the full-length KMNZ (excluding the disordered segments), 90% of the residues had very reasonable pLDDT scores above 70; 77% had scores above 80; that is, were predicted at least as 'confident'; and 34% of the residues scored above 90; that is, with very high confidence. Corroborating the high quality of the model, the predicted alignment error plots consistently indicated well-defined relative positions of the domains to each other, including the tetramerization domain of Ndc80c that is composed of SPC24, SPC25 and the C termini of NDC80 and NUF2. This model was used to create the cryo-EM model by fitting into the experimental map, deleting the parts that are either not present in the constructs used for cryo-EM experiments or invisible owing to flexibility, and optimized iteratively by manual building on COOT[63] and real-space refinement in PHENIX (v.1.20.1)[64]. Panels showing structures were prepared using UCSF ChimeraX (v.1.6)[65] or PyMOL (v.2.5.4). For the $KNL1^{1,880-2,342}$–ZWINT complexes, complete models were predicted with AF2 Multimer 3.2.1. The sequence of chicken ZWINT, which was not available from the databases, was kindly provided by Tatsuo Fukagawa. For the human model (accession IDs Q8NG31 ($KNL1^{1,880-2,342}$) and O95229), 82% of residues have a pLDDT score over 70. In the other species, the scores are 89% over 70 for *Xenopus laevis* (accession IDs A0A8J1LL48 ($KNL1_{2330-C}$) and A0A8J1LF42), 80% over 70 for *Gallus gallus* (A0A8V0ZUL4 ($KNL1_{1500-C}$)) and 65% over 70 for the *S. cerevisiae* model ($SPC105_{421-C}$ P53148, Kre28 Q04431).

## Molecular dynamics simulations

For computational efficiency, we considered a reduced fragment ($KMN^{hub}$) containing a truncated Mis12C helical bundle and the SPC24/SPC25 and KNL1 binding headgroups (Fig. 4f). More specifically, the simulated system (called $KMN^{hub}$) consisted of a truncated C-terminal part of the helical bundle of the Mis12C (residues 130–205 of MIS12, 157–205 of PMF1, 262–343 of DSN1 and 154–281 of NSL1) and the C-terminal headgroups of the SPC24 (residues 129–197), SPC25 (residues 127–224) and KNL1 domains (residues 2,131–2,342). Initial positions were taken from the cryo-EM structure determined in this study. The termini of the truncated fragments were capped with neutral groups. The protonation state of histidines was determined with PROPKA3 (ref. [66]), through

the APBS web server[67]. The CHARMM36m force field[68] was used for the protein, the CHARMM-modified TIP3P model for the water molecules[69] and default CHARMM parameters for the ions. In three additional sets of simulations, we removed SPC24/SPC25 (ΔNdc80C), KNL1 (ΔKNL1) or both (Mis12C) (Fig. 4f). In all four cases, the protein complex was placed in a dodecahedron box, such that the distance between it and the periodic boundaries was at least 2 nm. The protein was surrounded by approximately 141,000 water molecules. Sodium and chloride ions were added at a concentration of 0.15 M and additional ions were added to neutralize the net charge of the protein. The solvated $KMN^{hub}$, ΔNdc80C, ΔKNL1 and Mis12C systems consisted of 434,355, 434,290, 434,243 and 434,071 atoms in total, respectively. The potential energy of the solvated systems was energy-minimized by using the steep descent algorithm. Subsequently, the systems were thermalized by maintaining both the volume and the temperature constant during 500 ps (NVT conditions). A solvent relaxation step of 1 ns followed, with both the pressure and temperature constant (NPT conditions). During these two equilibration steps, the positions of the heavy atoms of the protein complex were harmonically restrained (elastic constant of the harmonic restraints, 1,000 kJ mol$^{-1}$ nm$^{-2}$). After release of the position restraints, simulations of 500 ns were performed. Five independent simulation runs (after the energy-minimization step) were carried out for each system for a cumulative simulation time in the production run phase of 2.5 μs per system. The first 100 ns of each production run was accounted as equilibration time and was therefore discarded from the analysis.

The temperature was kept constant at a reference value of 310 K using the Berendsen thermostat[70] during the NVT and NPT steps and the V-rescale thermostat[71] during production runs (with a coupling time constant of 1 ps). Pressure was maintained constant at 1 bar using the Berendsen[70] (NPT) and C-rescale[72] barostats (coupling time constant of 5 ps and reference compressibility of $4.5 \times 10^{-5}$ bar$^{-1}$). Both temperature and pressure were updated every 100 integration steps. Neighbors were treated according to the Verlet buffer scheme[73] with an update frequency of 20 steps and a buffer tolerance of 0.005 kJ mol$^{-1}$ ps$^{-1}$. Electrostatic interactions were calculated using the particle mesh Ewald algorithm[74,75], computing the Coulomb potential in the direct space for distances smaller than 1.2 nm and in the reciprocal space for larger distances. Short-range interactions were modeled by a Lennard Potential truncated at a distance of 1.2 nm. Bonds with hydrogen atoms in the protein complex were constrained via LINCS[76]. In addition, both bond and angular degrees of freedom of water were constrained using Settle[77]. Equations of motion were integrated using the Leap Frog algorithm at discrete time steps of 2 fs. Simulations were carried out using the GROMACS molecular dynamics package[78] (2023 version).

Rigid-body translation and rotation of the center of mass of the protein was removed before the simulation analyses by least-squares superposition of the conformations at different simulation times with the initial conformation. The backbone atoms of the rigid part of the Mis12 complex, excluding the flexible loop of the NSL1 domain, and in the case of the r.m.s.d. calculation the SPC24/SPC25 and KNL1 domains, were considered for the least-squares superposition. The following observables were extracted from the molecular dynamics simulations. Unless stated otherwise, normalized histograms of the quantities of interest were calculated, after concatenating the five independent simulation replicas of each system.

The r.m.s.d. of the coordinates of the backbone atoms of all protein domains (except the highly flexible C terminus of the NSL1 domain) was monitored relative to the initial coordinates. The r.m.s.d. was computed using the GROMACS gmx rms tool. The root mean square fluctuation (r.m.s.f.) of each residue of the Mis12C was computed after concatenation of the five simulation runs of each respective system. The r.m.s.f. was computed with the GROMACS gmx rmsf utility. The relative change in r.m.s.f. was defined as Δr.m.s.f. = (r.m.s.f.($X$)/r.m.s.f.($KMN^{hub}$)) − 1, with $X$ corresponding to the ΔNdc80C, ΔKNL1 and the Mis12C systems.

Accordingly, Δr.m.s.f. takes positive values if the r.m.s.f. of the *X* system (that is, a system lacking at least one of the binding partners) is larger than that of the KMN[hub] system. Reciprocally, Δr.m.s.f. takes negative values if the r.m.s.f. of the *X* system is smaller. Principal component analysis, consisting of the calculation and diagonalization of the covariance matrix of the atomic positions[79], was carried out for the backbone atoms of the rigid part of the Mis12C (that is, discarding the flexible NSL1 C terminus). To highlight the conformational changes owing to the presence (or absence) of the binding partners, the molecular dynamics trajectories of the four simulated systems were concatenated. The first principal component eigenvector explained ~33% of the variation of the positions and was considered for further analysis. Molecular dynamics trajectories were projected along this first principal component. This analysis was performed by means of the gmx covar and gmx anaeig tools. The solvent-accessible surface area of the Ndc80C (that is, the SPC24/SPC25 domains) and the KNL1 (that is, the KNL1 domain) binding sites on the Mis12C were computed. The area was calculated for the systems for which the binding partners were not present and therefore their binding sites were exposed, that is, the ΔNdc80C, ΔKNL1 and Mis12C systems. The area was obtained by using the double cubi lattice method[80] probing, as solvent, a sphere of radius 0.14 nm. The GROMACS gmx sasa tool was used for this calculation. Mis12C residues at the binding interface were assumed to be those that were at a distance of 5 nm or less from the respective binding protein in the initial conformation of the KMN[hub] system, and they were identified with PyMOL (http://www.pymol.org/pymol). Accordingly, the KNL1 binding site was composed of amino acid segments 185–204 of PMF1, 283–305 of DSN1 and 255–273 of NSL1, and the SPC24/SPC25 binding site of segments 313–342 of DSN1 and 197–224 of NSL1. The number of atomic contacts between heavy atoms of Mis12 and heavy atoms of the Ndc80C (SPC24/SPC25) and Knl1C (KNL1) binding domains was obtained. Two heavy atoms were assumed to be in contact if they were at a distance equal to or smaller than 0.35 nm. To avoid redundancy, contacts corresponding to the same atom in Mis12C were grouped and counted as one contact. The number of contacts was computed with the GROMACS gmx mindist option. Simulation analyses were carried out with our own in-house Bash and Python scripts. Protein conformations and movies were rendered with PyMOL.

Despite the truncation, the Mis12C helical bundle of KMN[hub] was stable and SPC24/SPC25 and KNL1 remained stably bound to it (Supplementary Video 1), with an r.m.s.d. from the initial conformation leveling off at values smaller than 1 nm (Extended Data Fig. 7d). Omission of SPC24/SPC25, KNL1 or both increased the flexibility of the DSN1 and NSL1 segments without significantly impacting MIS12 and PMF1 (Fig. 4g and Extended Data Fig. 7e). Structural fluctuations were maximal in the regions implicated in Ndc80C and KNL1 binding, which were more rigid in association with the cognate ligands and highly flexible upon their removal. We inspected more closely the dependence between binding partners. (Un)binding of the SPC24/SPC25 domains did not affect the r.m.s.f. of the KNL1 binding site and, reciprocally, (un)binding of KNL1 did not alter the r.m.s.f. of the SPC binding site (Fig. 4g). Principal component analysis revealed that the DSN1 C-terminal region adopted a straighter orientation when the SPC24/SPC25 RWD domains were bound to it, irrespective of KNL1 binding (see distributions along the first principal component for KMN[hub] and ΔKNL1 systems in Extended Data Fig. 7f). On the contrary, this segment of DSN1 adopted a broader range of conformations, including bent ones, when SPC24/SPC25 was not bound (ΔNdc80C and Mis12C in Extended Data Fig. 7f). In these systems where Ndc80C was omitted, KNL1 caused NSL1 to sample straight conformations reminiscent of those observed when NSL1 is bound to Ndc80C (shown by minor but significant peaks corresponding to straight orientations in the ΔNdc80C projection in Extended Data Fig. 7f). This tendency to stabilization was also evident near the Mis12C–Ndc80C interface, whose residues were significantly more solvent-exposed in the presence of KNL1 (ΔNdc80C system) than in its absence (Mis12C system) (Extended Data Fig. 7fh, left). Thus, our simulations suggest that binding of KNL1 shifts Mis12C into an unshielded straight conformation that prepares it for binding to SPC24/SPC25. The effect of Ndc80C on the KNL1 binding site in the simulations was less pronounced. The flexibility of the PMF1 and DSN1 helices at the KNL1 binding site remained largely unaffected by the removal of any of the binding partners (Fig. 4f and Extended Data Fig. 7e,f). Binding of SPC24/SPC25 did not alter the exposure of the KNL1 binding site (Extended Data Fig. 7g, right). Nevertheless, the number of atomic contacts at the Mis12C–KNL1 interface displayed a broader distribution in the presence of SPC24/SPC25 (Extended Data Fig. 7h, right), possibly indicating how SPC24/SPC25 predisposes Mis12C to KNL1 binding.

## Reporting summary

Further information on research design is available in the Nature Portfolio Reporting Summary linked to this article.

## Data availability

The cryo-EM map and atomic coordinates have been deposited in the Electron Microscopy Data Bank (EMDB) under accession code EMD-18179 and the Protein Data Bank (PDB) under accession code PDB 8Q5H, respectively. Other structural models used in this study can be retrieved from the PDB under accession codes PDB 4NF9 (KNL1/NSL1), PDB 3VZA (chicken CENP-T), PDB 5T6J (yeast Dsn1), PDB 5LSK (Mis12C/CENP-C) and PDB 7QOO (CCAN complex). Validation reports and processed files were provided with the manuscript for peer review. Source data are provided with this paper.

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

## Acknowledgements

We are grateful to S. Maffini and N. Schmidt for help with microscopy experiments and data analysis, O. Hofnagel and D. Prumbaum for help in electron microscopy data collection, R. Gasper-Schönenbrücher for help with biophysical experiments, T. Fukagawa for sharing unpublished sequences and S. Yatskevich and D. Barford for sharing unpublished results. A.M. gratefully acknowledges funding from the Max Planck Society, the European Research Council (ERC) Synergy Grant 951430 (BIOMECANET), the Marie-Curie Training Network DivIDE (project number 675737), the DGF's Collaborative Research Centre 1430 'Molecular Mechanisms of Cell State Transitions' and the CANTAR network under the Netzwerke-NRW program. M.K., C.A.S. and F.G. are grateful for financial support by the Klaus Tschira Foundation. The funders had no role in study design, data collection and analysis, decision to publish or preparation of the manuscript.

## Author contributions

A.M. conceived the project. S.G., T.R., M.K., C.A.S. and I.R.V. conducted formal analysis. S.R., F.G. and A.M. acquired funding. S.G., S.P., T.R., M.K., C.A.S. and I.R.V. conducted the investigation. C.A.S, F.G., A.M., S.R. and I.R.V. administered the project. M.T. acquired resources. C.A.S., F.G., A.M., S.R. and I.R.V. supervised the project. C.A.S., A.M., S.R. and I.R.V. validated the data. C.A.S., A.M., S.P., T.R. and I.R.V. visualized the results. A.M. wrote the original draft; C.A.S., F.G., A.M., S.P., T.R., S.R. I.R.V. reviewed and edited the final version of the manuscript.

## Funding

## Competing interests

The authors declare no competing interests

## Additional information

**Extended data** is available for this paper at https://doi.org/10.1038/s41594-024-01230-9.

**Correspondence and requests for materials** should be addressed to Andrea Musacchio.

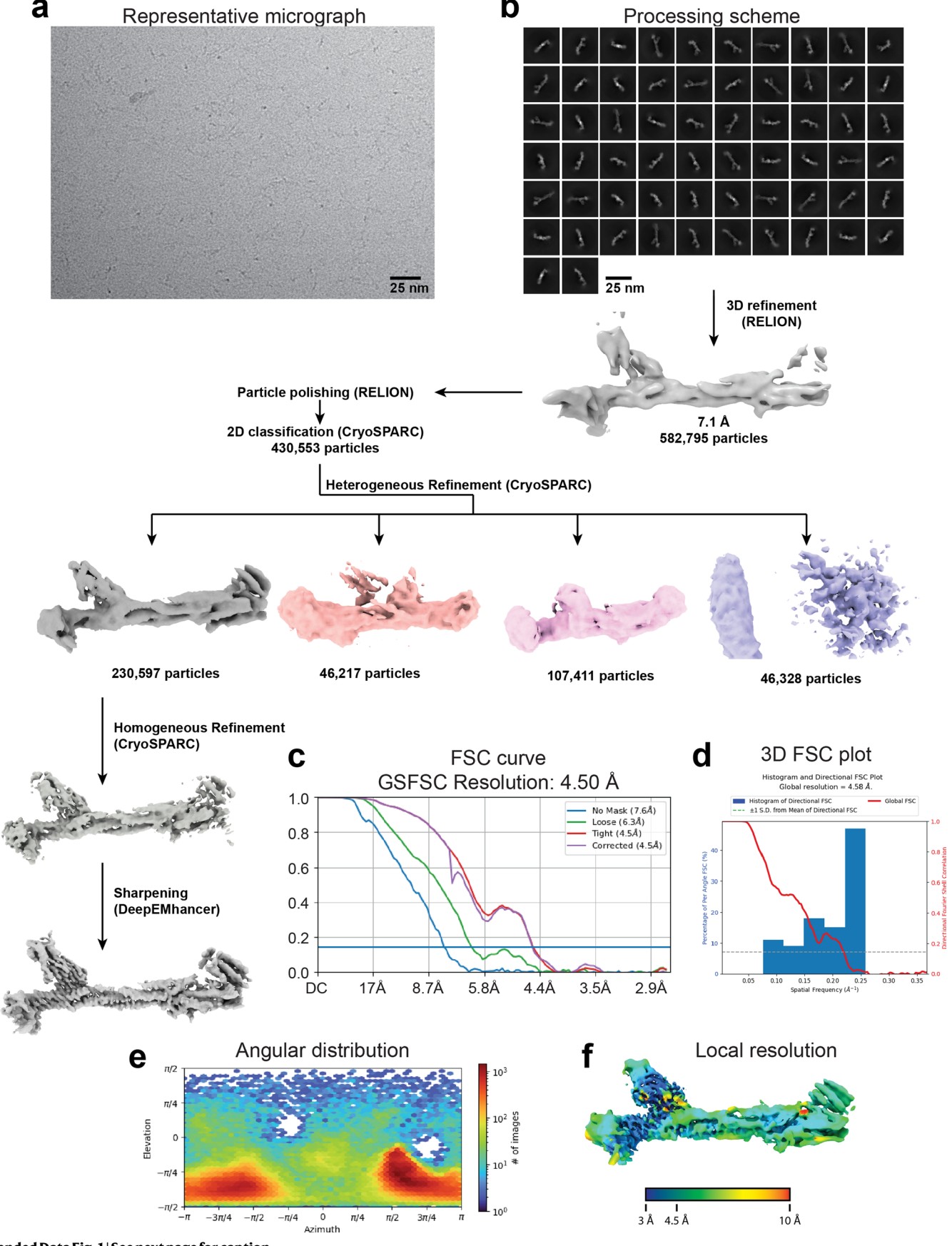

**Extended Data Fig. 1 | See next page for caption.**

**Extended Data Fig. 1 | Cryo-EM data acquisition and processing strategy.**
(**a**) Representative micrograph of the KMN sample. (**b**) Outline of data processing procedure (see Methods). (**c**) Fourier shell correlation (FSC) plots between two independent half-maps without masking (blue) or masking with a loose (green) or tight (red) mask around the particle that have been automatically created by cryoSPARC. The purple curve corresponds to masking with the tight mask with correction by noise substitution. The horizontal blue line marks the 0.143 gold-standard FSC criterion. (**d**) Three-dimensional Fourier shell correlation plot showing the distribution of directional resolution to judge the effect of preferred orientations on the quality of the reconstruction. (**e**) Angular distribution of the particles. (**f**) Local resolution of the final reconstruction plotted on a locally filtered map. Colors according to the scale below range from blue (3 Å) over green, yellow and orange to red (10 Å).

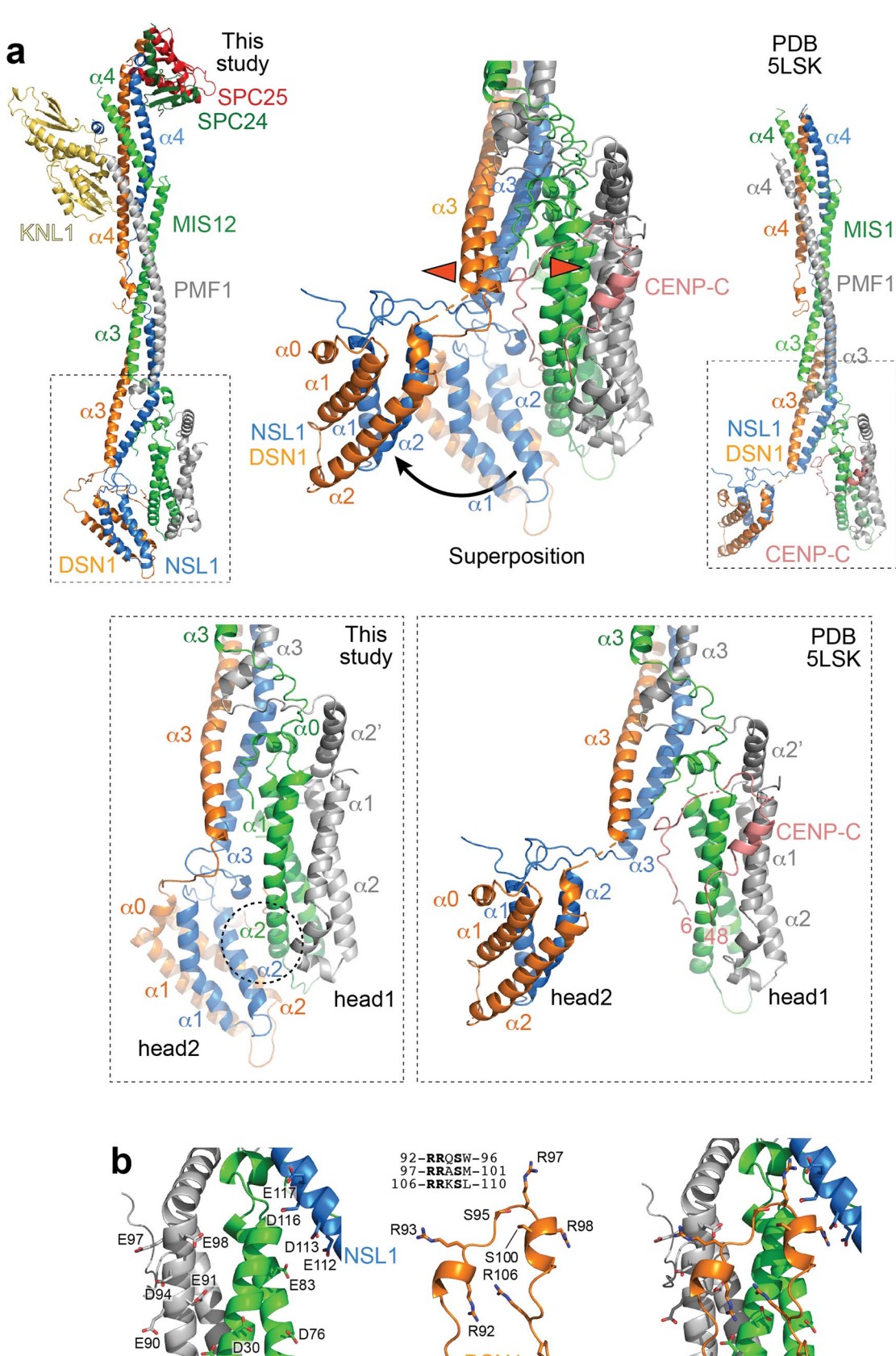

**Extended Data Fig. 2 | See next page for caption.**

**Extended Data Fig. 2 | Analysis of the closed conformation of the Mis12C heads. (a)** Comparison of the closed conformation of the heads observed in this study and the open conformation of the heads observed in a previous crystal structure of the complex of Mis12C with CENP-C (PDB 5LSK; ref. 1). In the central superposition, the movement that sways the head2 out of position is marked by a black arrow. The red arrowheads indicate an outward movement of the helices lining the cleft that separates the DSN1 and MIS12 helices when CENP-C binds. **(b)** AF2 model of the interface between a positively-charged regulatory loop of DSN1 and the negatively charged head1 domain (with contributions from NSL1 and DSN1 in head2).

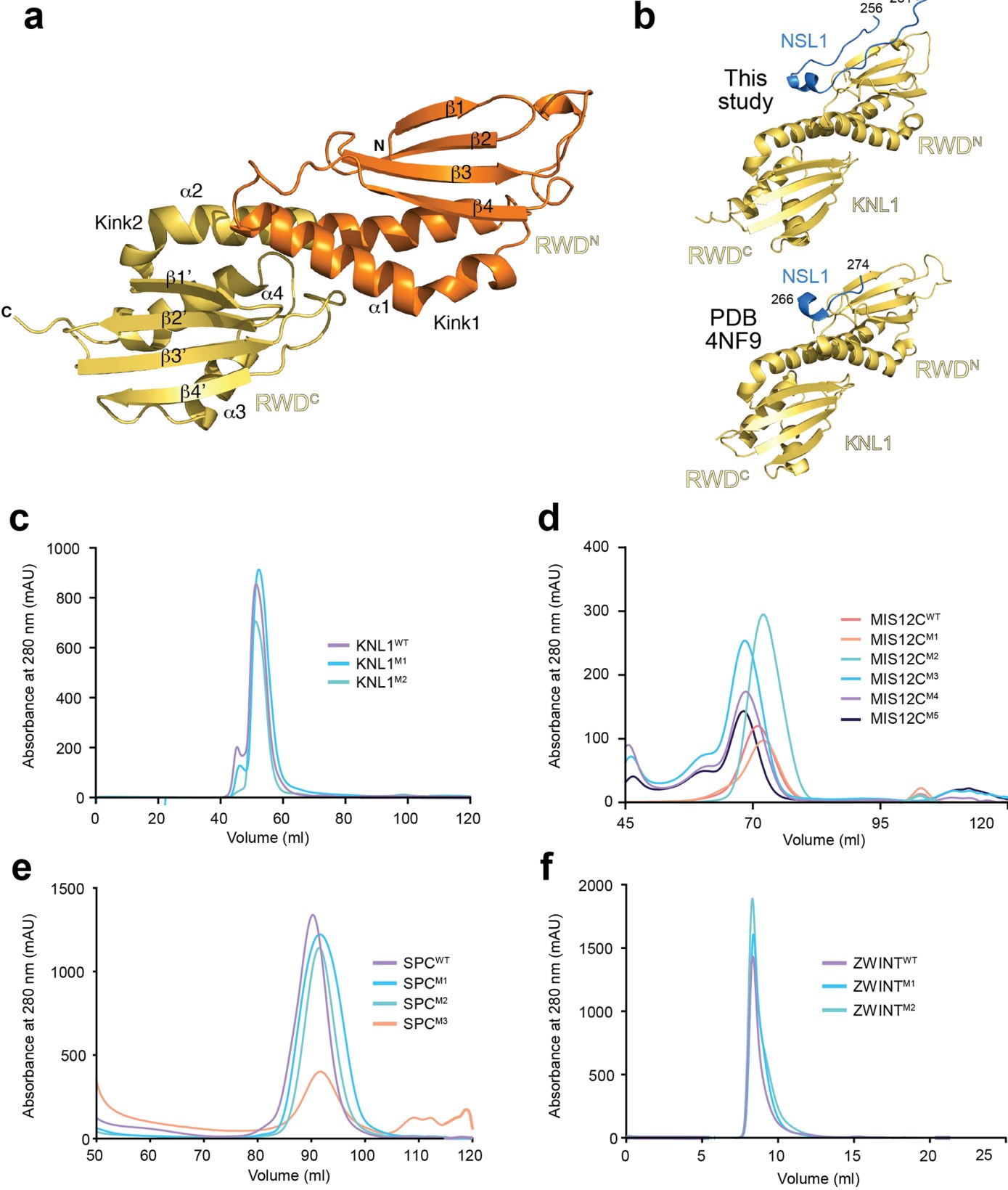

**Extended Data Fig. 3 | KNL1 RWD domains and hydrodynamic analyses of mutants.** (**a**) Definition of secondary structure of the two RWD domains of the KNL1 C-terminal region, as originally defined in reference[2]. (**b**) Comparison of the interaction of the NSL1 C-terminal tail with KNL1 as emerging from our current and previous work[2]. The model of the NSL1 chain in this representation includes extensions, built by AlphaFold2, to the helical segment shown in Fig. 2a. Details

in Methods and main text. (**c-f**) Size-exclusion chromatography elution profiles for the indicated wild type and mutant species. (**c**) KNL1 was analyzed on a HiLoad 16/600 Superdex 75 column, (**d**) Mis12C and (**e**) SPC24/SPC25 species were analyzed on a HiLoad 16/600 Superdex 200 column, (**f**) ZWINT was analyzed on a Superdex 75 10/300 column.

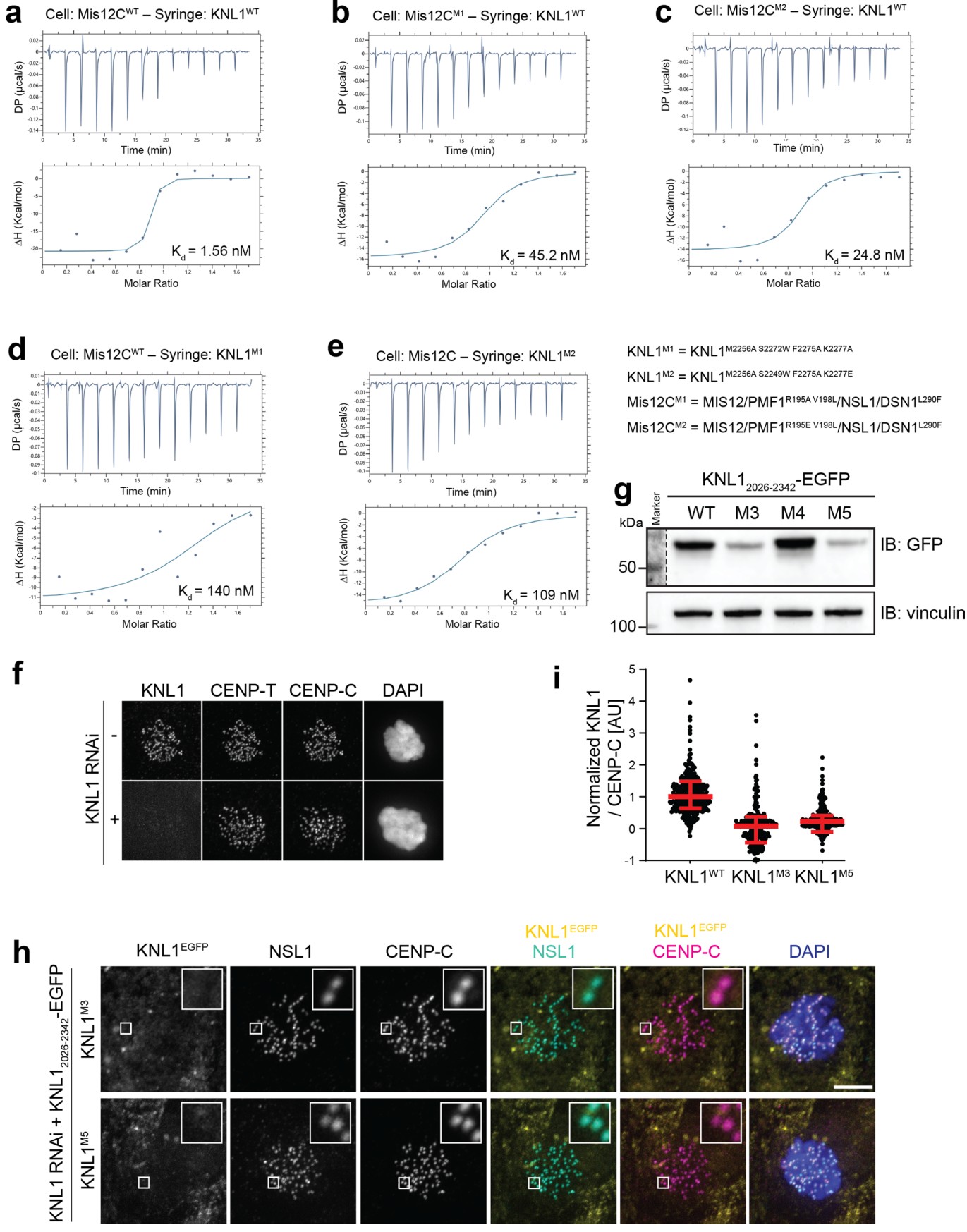

**Extended Data Fig. 4 | See next page for caption.**

**Extended Data Fig. 4 | Further analyses of the KNL1/Mis12C interaction.**
(**a-e**) ITC thermograms for the experiments tabulated in Fig. 2c. (**f**) Assessment by immunofluorescence of efficiency of protocol of RNAi-based depletion of KNL1. This is a representative image from at three independent experiments. Scale bar 5 μm. (**g**) Expression levels of wild-type and mutant KNL1$^{2026-2342}$ constructs used for the experiments in panel f. The shown marker lane is from an intensity-enhanced version of the same blot. The vertical line is included solely to indicate this fact. The blots at normal and enhanced intensities are shown in Supplementary Fig. 2. (**h**) Immunofluorescence images of the KNL1$^{M3}$ (n = 586 kinetochores) and KNL1$^{M5}$ (n = 566 kinetochores) mutants, and complementing the experiment displayed in Fig. 2e-f, where the experiment's positive control is displayed. Scale bar 5 μm. (**i**) Quantification of experiments in panel **h**. Red bars represent median and interquartile range of normalized single kinetochores.

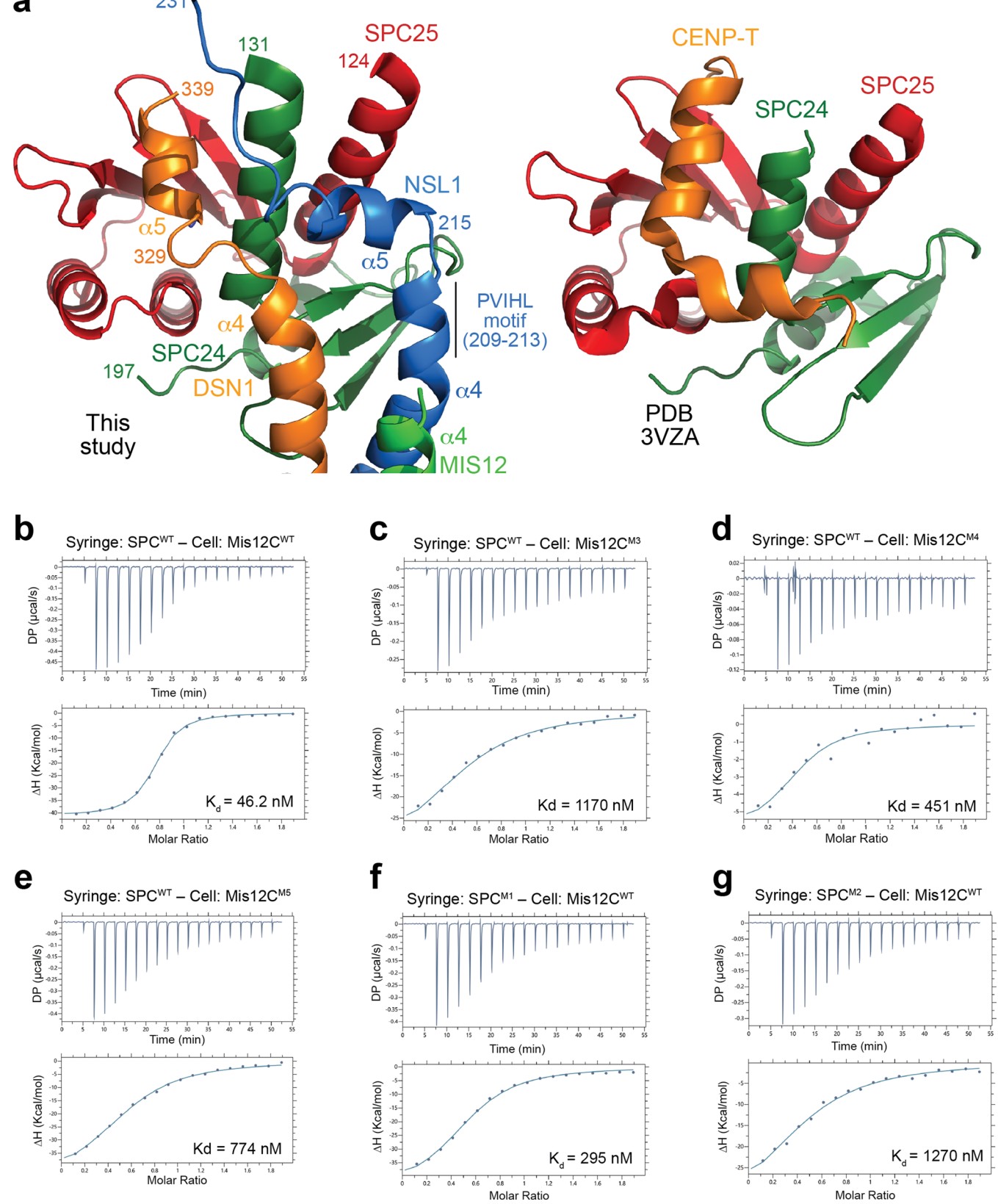

**Extended Data Fig. 5 | Further analyses of the Ndc80C/Mis12C interaction.** (**a**) Cartoon models of the interaction of the human (left, this study) and avian (PDB ID 3VZA) SPC24/SPC25 complexes, displayed in the same orientation. CENP-T is shown in orange to emphasize the structural similarity with DSN1. (**b-g**) Thermograms for the experiments tabulated in Fig. 3c.

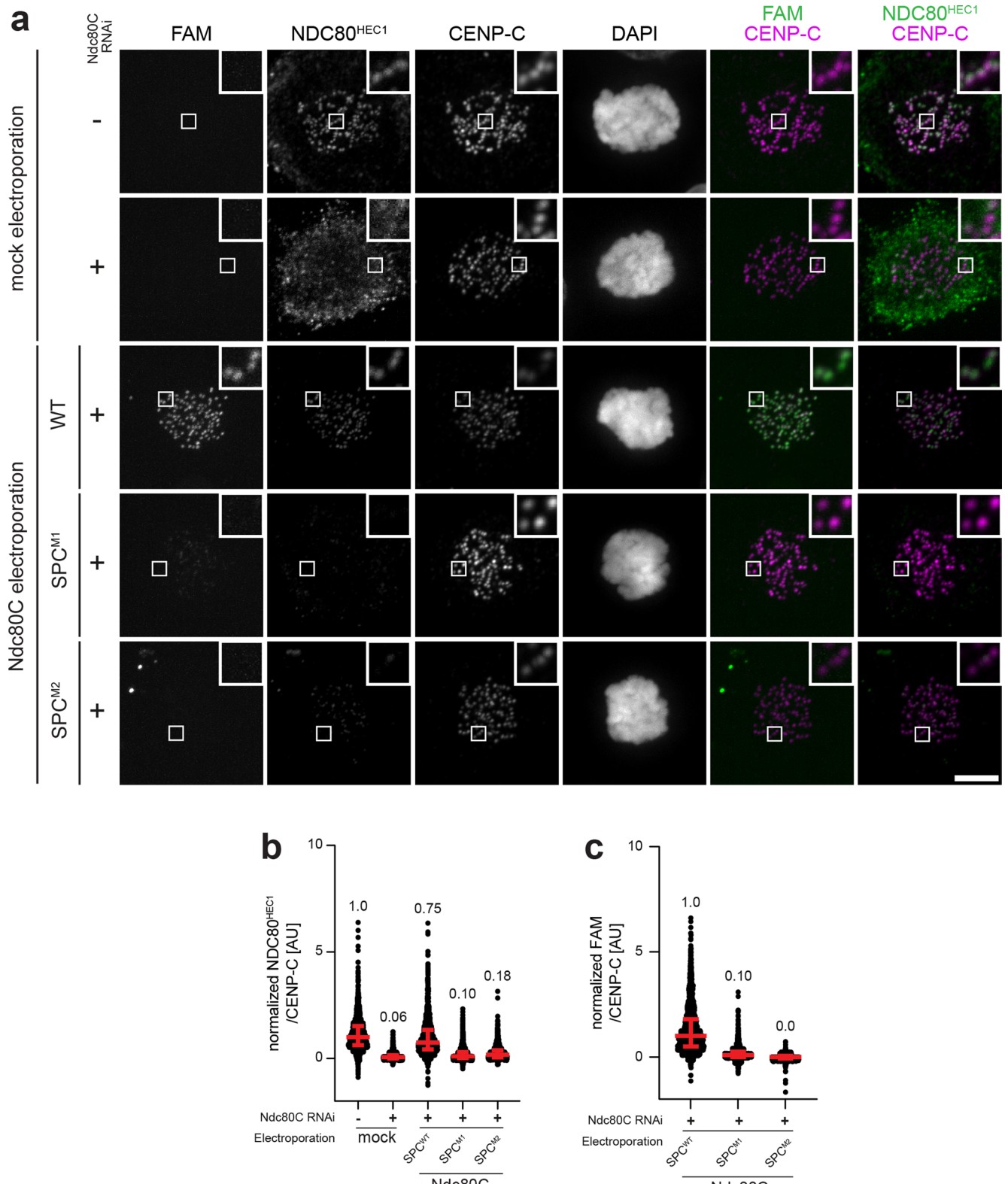

**Extended Data Fig. 6 | Complete experiment for Fig. 3g. (a)** Fluorescence and immunofluorescence analysis on cells treated for Ndc80C RNAi as indicated, and electroporated with Ndc80C-labelled FAM. Samples showing electroporation of wild type Ndc80C and the SPC^M1 mutant (Ndc80C reconstituted with the mutation) were already shown in Fig. 3g. Scale bar = 5 µm. **(b-c)** Quantification of experiments displayed in **a**. Red bars represent median, indicated above the plots, and interquartile range of normalized single kinetochores (n) intensity values. In **b**. for mock electroporation Ndc80C RNAi - (n = 1464), Ndc80C RNAi + (n = 1539). In **b,c**. for SPC^WT (n = 1873), SPC^M1 (n = 1784), SPC^M2 (n = 954) from two independent experiments.

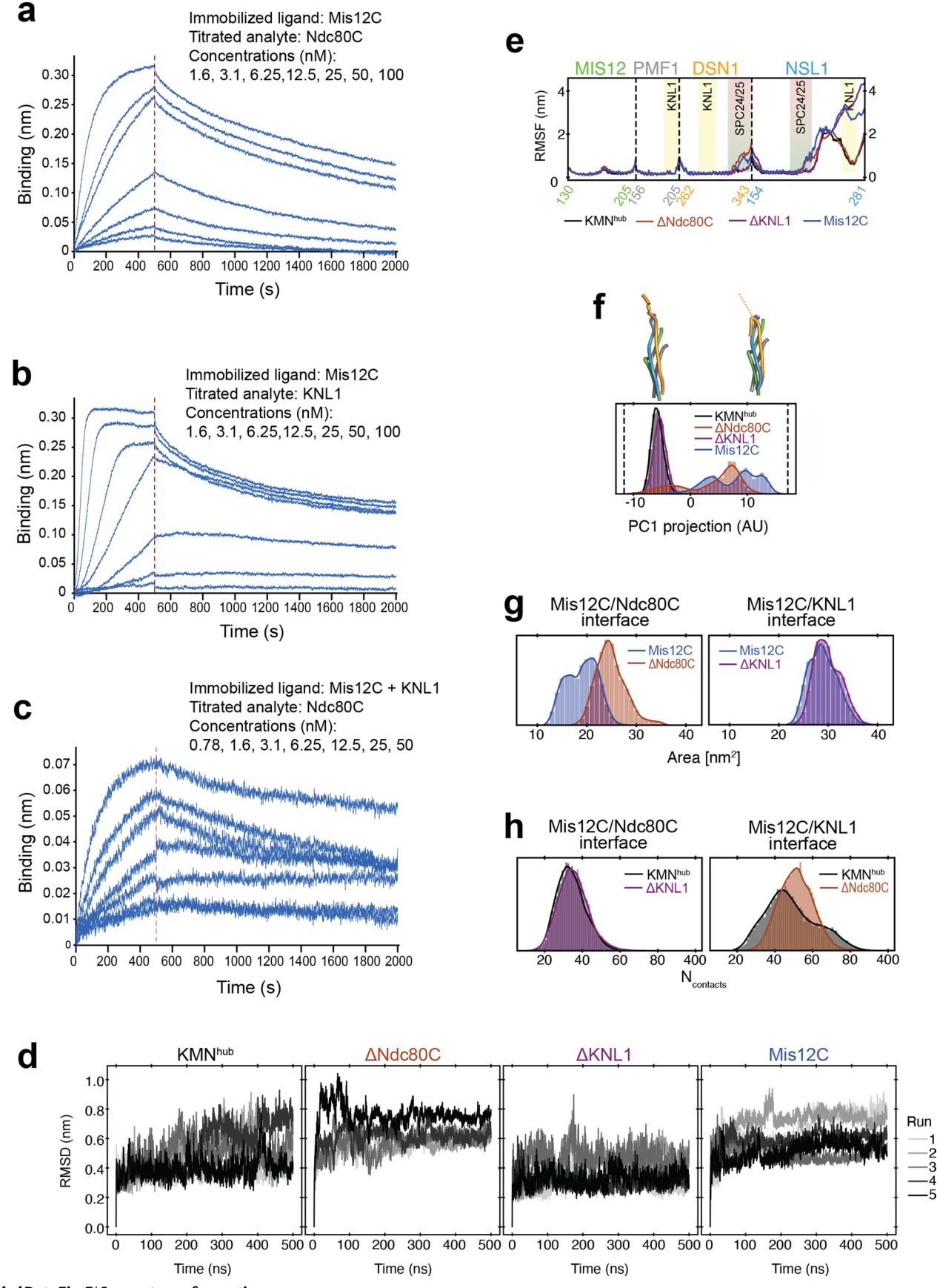

**Extended Data Fig. 7 | See next page for caption.**

**Extended Data Fig. 7 | Additional data on KMN cooperativity and MD simulations.** (**a**-**c**) BLI experiments on the indicated species and that generated binding data reported in Fig. 4e, (**a**) for Mis12C-Ndc80C, (**b**) for Mis12C-KNL1, (**c**) for Mis12C + KNL1-Ndc80C. (**d**) Root mean square deviation (RMSD) from the initial conformation recovered from molecular dynamics simulations of the four indicated systems. Each line represents one independent simulation replica (n = 5). (**e**) Per-residue root mean square fluctuations (RMSFs) of the Mis12C segments for the four systems indicated at the bottom. The four helical elements are separated by the dashed lines and their sequence coverage is shown on the X-axis. The colored rectangles indicate the binding site of the SPC24/SPC25 domains (red/green) and KNL1 (yellow). (**f**) Principal component analysis retrieved a main principal component (PC1) related to the bending

motion of the C-terminus of the DSN1 domain (see cartoon representations). This component explained 33% of the positional fluctuations of the rigid part of the studied Mis12C fragment. Projection of the MD trajectories onto this PC component indicates whether the DSN1 unit adopted either a straight or a bent conformation. The normalized distribution of such projection is shown for the four studied systems (arbitrary units and same color as in Fig. 4f-g). (**g**) Normalized distribution of the exposed surface area of the Mis12C/Ndc80C (*left*) or Mis12C/KNL1 (*right*) binding interfaces for the indicated systems. (**h**) Normalized distribution of the number of atomic contacts ($N_{contacts}$), between the Mis12C/Ndc80C (*left*) or Mis12C/KNL1 (*right*) binding interfaces is presented for the indicated systems.

**a**

| Construct | Sequence | Binding to KNL1 | Binding to NDC80 |
|---|---|---|---|
| Mis12C[WT] | MIS12, PMF1, NSL1, DSN1 | + | + |
| Mis12C[M6] | MIS12[NSL1(207-281)], PMF1[DSN1(331-356)], NSL1[Δ207-281], DSN1[Δ331-356] | + | - |
| Mis12C[M7] | MIS12[DSN1(331-356)], PMF1[NSL1(207-81)], NSL1[Δ207-281], DSN1[Δ331-356] | + | - |
| Mis12C[M8] | MIS12[NSL1(207-281)], PMF1, NSL1[Δ207-281], DSN1[Δ331-356] | + | - |
| Mis12C[M9] | MIS12, PMF1[NSL1(207-281)], NSL1[Δ207-281], DSN1[Δ331-356] | + | - |
| Mis12C[M10] | MIS12[NSL1(207-281)], PMF1[DSN1(318-356)], NSL1[Δ207-281], DSN1[Δ318-356] | + | - |
| Mis12C[M11] | MIS12[DSN1(318-356)], PMF1[NSL1(207-281)], NSL1[Δ207-281], DSN1[Δ318-356] | + | - |
| Mis12C[M12] | MIS12[NSL1(207-281)], PMF1, NSL1[Δ207-281], DSN1 | + | - |
| Mis12C[M13] | MIS12, PMF1[NSL1(207-281)], NSL1[Δ207-281], DSN1 | + | - |
| Mis12C[M14] | MIS12[NSL1(227-281)], PMF1[DSN1(318-356)], NSL1[Δ227-281], DSN1[Δ318-356] | + | - |
| Mis12C[M15] | MIS12[DSN1(318-356)], PMF1[NSL1(227-281)], NSL1[Δ227-281], DSN1[Δ318-356] | + | - |
| Mis12C[M16] | MIS12[NSL1(227-281)], PMF1[6GS-DSN1(318-356)], NSL1[Δ227-281], DSN1[Δ318-356] | + | - |
| Mis12C[M17] | MIS12[6GS-DSN1(318-356)], PMF1[NSL1(227-281)], NSL1[Δ227-281], DSN1[Δ318-356] | + | - |
| Mis12C[M18] | MIS12[NSL1(227-281)], PMF1, NSL1[Δ227-281], DSN1 | + | + |
| Mis12C[M19] | MIS12, PMF1[NSL1(227-281)], NSL1[Δ227-281], DSN1 | + | + |

**b**

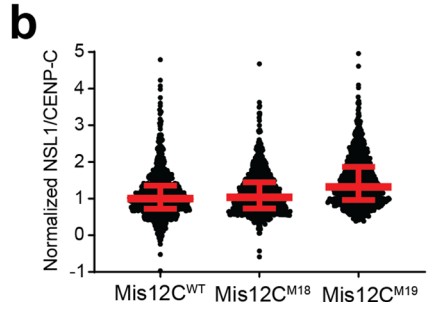
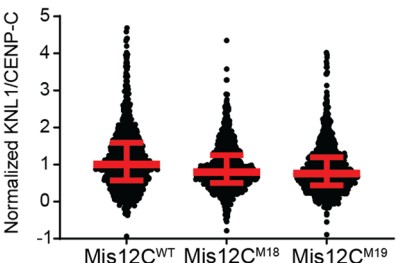
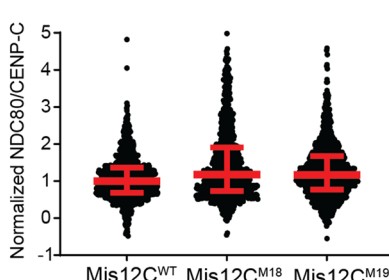

**Extended Data Fig. 8 | See next page for caption.**

**Extended Data Fig. 8 | Swap constructs and their kinetochore quantification.** (**a**) Table of swap constructs discussed in the main text. Binding proficiency (+) or lack thereof (-) was evaluated by size-exclusion chromatography as shown for the two constructs in Fig. 5b-c. (**b**) Further quantifications for the experiment in Fig. 5d from three independent experiments. Sample size for MIS12, KNL1

quantifications (n = 1034 kinetochores for Mis12$^{WT}$, n = 848 kinetochores for Mis12$^{MI8}$, and n = 884 kinetochores for Mis12$^{MI9}$). Sample size for NDC80 quantification (n = 942 kinetochores for Mis12$^{WT}$, n = 826 kinetochores for Mis12$^{MI8}$, and n = 875 kinetochores for Mis12$^{MI9}$). Median and interquartile range of the presented data are shown.

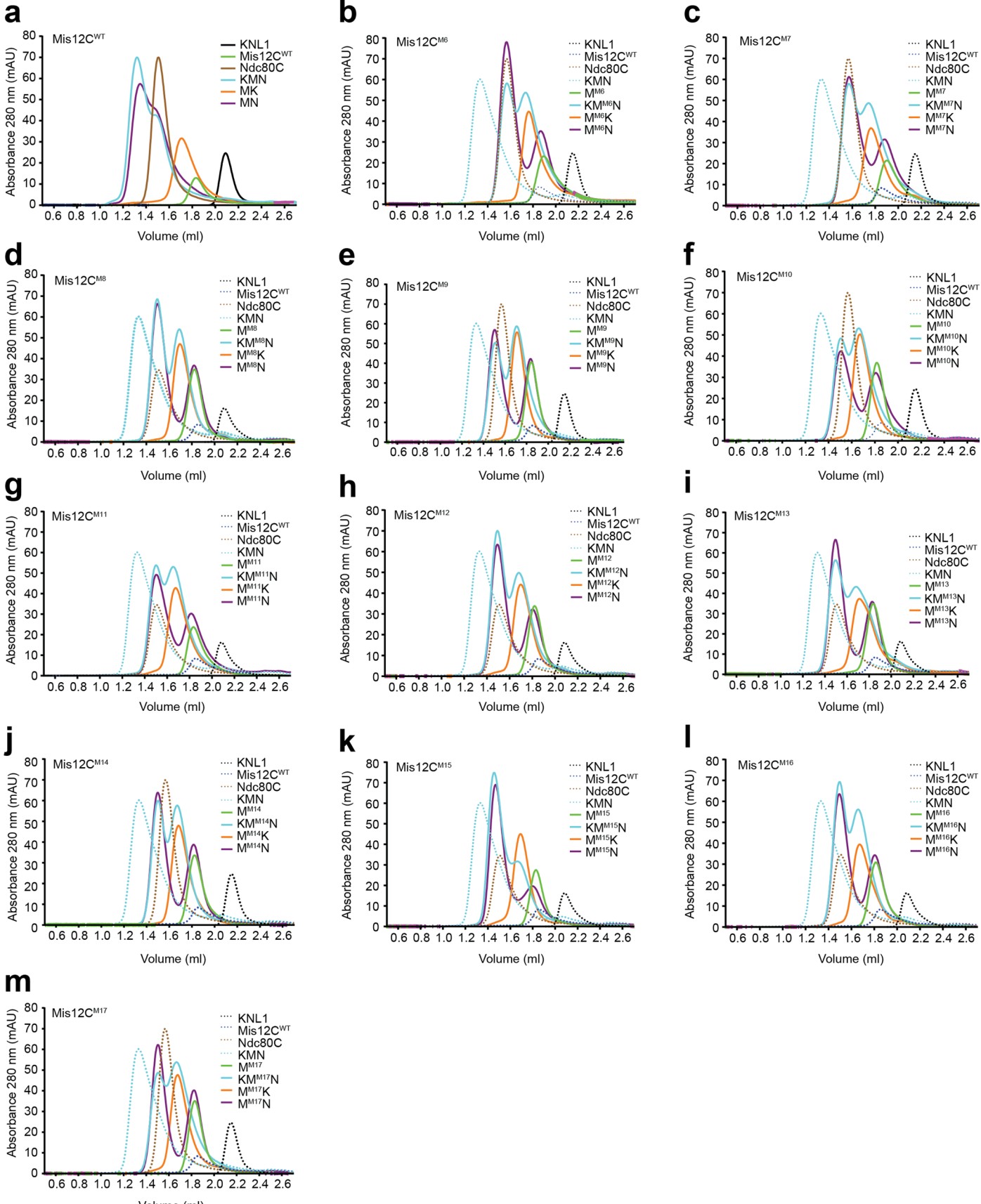

**Extended Data Fig. 9 | Hydrodynamic interaction analysis of swap mutants. (a-m)** Size-exclusion chromatography profiles of the indicated Mis12C constructs and their interactions with KNL1 and Ndc80C using a Superdex 200 5/150 column. All proteins were used at 3 μM.

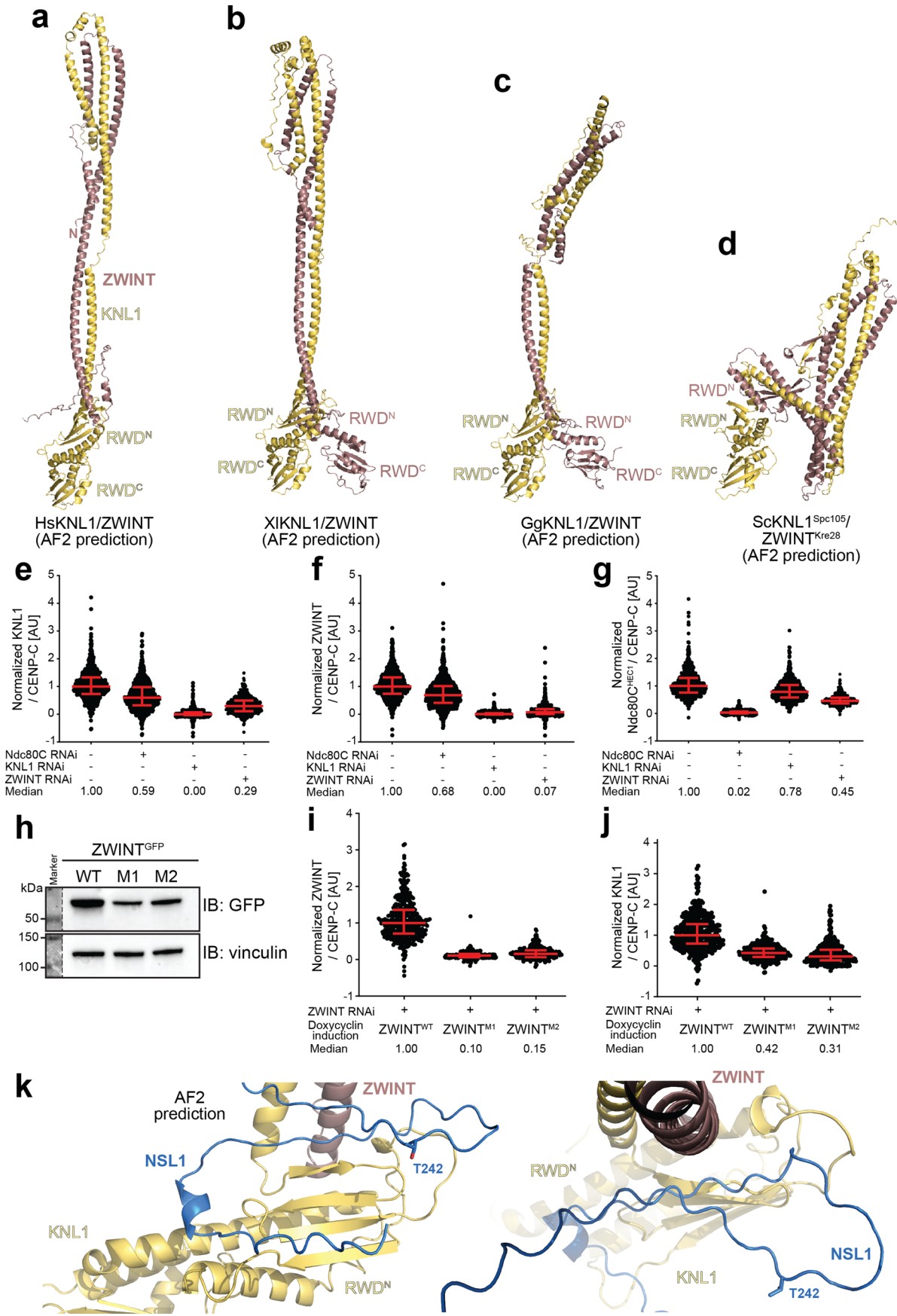

**Extended Data Fig. 10 | See next page for caption.**

**Extended Data Fig. 10 | Hydrodynamic interaction analysis of swap mutants.** (**a-d**) AF2 prediction of KNL1/ZWINT complexes in the indicated species, including a repetition of the human complex in panel **a**, (**a**) for HsKNL1/ZWINT, (**b**) for XlKNL1/ZWINT, (**c**) for GgKNL1/ZWINT, (**d**) for ScKNL1/ZWINT. (**e-g**) Residual levels of the indicated proteins in cells processed as indicated and from three independent experiments. Sample sizes for KNL1 and NDC80 quantification were n = 783 kinetochores for control, n = 907 kinetochores for Ndc80 RNAi, n = 848 kinetochores for KNL1 RNAi, and n = 569 kinetochores for ZWINT RNAi. Sample size for ZWINT quantifications were n = 853 kinetochores for control, n = 982 kinetochores for Ndc80 RNAi, n = 1032 kinetochores for KNL1 RNAi, and n = 885 kinetochores for ZWINT RNAi). (**h**) Immunofluorescence analysis of the indicated exogenous Zwint constructs used for the experiment in Fig. 6g. The shown marker lanes are from an intensity-enhanced version of the same blot. The vertical line is included solely to indicate this fact. The blots at normal and enhanced intensities are are shown in Supplementary Fig. 2. (**i-j**) Quantification of a localization experiment in Fig. 6g. Red bars represent median and interquartile range of normalized kinetochore intensity values from ZWINT (n = 391 kinetochores), Z1 (n = 291), Z2 (n = 308) from three independent experiments. (**k**) Cartoon model for an AF2 prediction of the NSL1 C-terminal tail bound to the KNL1/ZWINT complex. The position of a phosphorylated residue potentially involved in KNL1 regulation is indicated (see main text for details). Median and interquartile range of the quantified immunofluorescence data are shown.

# Reporting Summary

## Statistics

For all statistical analyses, confirm that the following items are present in the figure legend, table legend, main text, or Methods section.

| n/a | Confirmed | |
|---|---|---|
| ☐ | ☒ | The exact sample size (*n*) for each experimental group/condition, given as a discrete number and unit of measurement |
| ☐ | ☒ | A statement on whether measurements were taken from distinct samples or whether the same sample was measured repeatedly |
| ☐ | ☒ | The statistical test(s) used AND whether they are one- or two-sided *Only common tests should be described solely by name; describe more complex techniques in the Methods section.* |
| ☒ | ☐ | A description of all covariates tested |
| ☒ | ☐ | A description of any assumptions or corrections, such as tests of normality and adjustment for multiple comparisons |
| ☐ | ☒ | A full description of the statistical parameters including central tendency (e.g. means) or other basic estimates (e.g. regression coefficient) AND variation (e.g. standard deviation) or associated estimates of uncertainty (e.g. confidence intervals) |
| ☒ | ☐ | For null hypothesis testing, the test statistic (e.g. *F*, *t*, *r*) with confidence intervals, effect sizes, degrees of freedom and *P* value noted *Give P values as exact values whenever suitable.* |
| ☒ | ☐ | For Bayesian analysis, information on the choice of priors and Markov chain Monte Carlo settings |
| ☒ | ☐ | For hierarchical and complex designs, identification of the appropriate level for tests and full reporting of outcomes |
| ☒ | ☐ | Estimates of effect sizes (e.g. Cohen's *d*, Pearson's *r*), indicating how they were calculated |

*Our web collection on statistics for biologists contains articles on many of the points above.*

## Software and code

Policy information about availability of computer code

| | |
|---|---|
| Data collection | EPU v2.7 Thermo Fisher Scientific<br>Image Lab Bio-rad https://www.bio-rad.com/de-de/product/image-lab-software?ID=KRE6P5E8Z<br>Slidebook, 3i (Intelligent Imaging Innovations)<br>FortéBio Octet BLI Data acquisition software<br>Deltavision SoftWoRx software |
| Data analysis | cryoSPARC Version 4.2.1 (Punjani et al. 2017, https://cryosparc.com/)<br>TranSHIRE v1.4 (Stabrin et al., 2020 https://transphire.readthedocs.io/en/latest/<br>MOTIONCOR2 v1.3.0 Zheng et al., 2017 http://msg.ucsf.edu/em/software/motioncor2.html<br>CTFFIND4 (Rohou and Grigorieff, 2015 http://grigoriefflab.janelia.org/ctffind4)<br>MOTIONCOR2 (Zheng et al., 2017 http://msg.ucsf.edu/em/software/motioncor2.html)<br>SPHIRE v1.5 (Moriya et al., 2017 http://sphire.mpg.de)<br>crYOLO v1.7 (Wagner et al., 2019 https://cryolo.readthedocs.io/en/stable/)<br>RELION v3.1.2 (Scheres Lab https://www3.mrc-lmb.cam.ac.uk/relion/index.php?title=Main_Page)<br>Pymol v2.5.4 Schrödinger, LLC https://pymol.org/2/<br>COOT v0.9.8.2 (Emsley et al., 2010 https://www2.mrc-lmb.cam.ac.uk/personal/pemsley/coot/)<br>PHENIX v1.20.1 (Adams et al., 2010 https://www.phenix-online.org)<br>DeepEMhancer v0.14 (Sanchez-Garcia et al., 2021 https://github.com/rsanchezgarc/deepEMhancer)<br>AlphaFold-Multimer v3.2.1 Evans et al., 2021<br>DynDom6D (Veevers and Hayward, 2019 http://dyndom.cmp.uea.ac.uk/dyndom/dyndomDownload.jsp)<br>GraphPad Prism Version 9.0.2 (134) (GraphPad Software Inc http://www.graphpad.com) |

Fiji Version 2.0.0-rc-69/1.52n (Schindelin et al., 2012 http://imageJ.nih.gov/ij/)
CRaQ (Bodor et al., 2012 NA)
Image Lab (Bio-rad, https://www.bio-rad.com/de-de/product/image-lab-software?ID=KRE6P5E8Z)
Microcal PEAQ-ITC analysis software
FortéBio Octet BLI Data Analysis software

For manuscripts utilizing custom algorithms or software that are central to the research but not yet described in published literature, software must be made available to editors and reviewers. We strongly encourage code deposition in a community repository (e.g. GitHub). See the Nature Portfolio guidelines for submitting code & software for further information.

## Data

Policy information about availability of data

All manuscripts must include a data availability statement. This statement should provide the following information, where applicable:
- Accession codes, unique identifiers, or web links for publicly available datasets
- A description of any restrictions on data availability
- For clinical datasets or third party data, please ensure that the statement adheres to our policy

The cryo-EM map and atomic coordinates have been deposited in the EMDB under accession code EMD-18179 and the PDB under accession code 8Q5H, respectively. Validation report and processed files were provided with the manuscript for peer review.

## Research involving human participants, their data, or biological material

Policy information about studies with human participants or human data. See also policy information about sex, gender (identity/presentation), and sexual orientation and race, ethnicity and racism.

| | |
|---|---|
| Reporting on sex and gender | Not applicable |
| Reporting on race, ethnicity, or other socially relevant groupings | Not applicable |
| Population characteristics | Not applicable |
| Recruitment | Not applicable |
| Ethics oversight | Not applicable |

Note that full information on the approval of the study protocol must also be provided in the manuscript.

# Field-specific reporting

Please select the one below that is the best fit for your research. If you are not sure, read the appropriate sections before making your selection.

☒ Life sciences      ☐ Behavioural & social sciences      ☐ Ecological, evolutionary & environmental sciences

For a reference copy of the document with all sections, see nature.com/documents/nr-reporting-summary-flat.pdf

# Life sciences study design

All studies must disclose on these points even when the disclosure is negative.

| | |
|---|---|
| Sample size | The relevant figure legends report the number of kinetochores for each condition in which fluorescence values were collected. We did not predetermine sample size, nor we calculated sample size. The choice of sample size was based on previous examples in the field, and are consistent with extensive previous experimentation determining epistatic relationships within kinetochores. |
| Data exclusions | No data were excluded from the analysis |
| Replication | We indicate the number of technical replicates of each experiment in the relevant figure legends |
| Randomization | For each immuno-fluorescence analysis, cells (in the indicated number) were chosen randomly for each quantification. |
| Blinding | The investigators were not blinded during data collection. The same investigators carried out the data collection and data analysis processes. |

# Reporting for specific materials, systems and methods

We require information from authors about some types of materials, experimental systems and methods used in many studies. Here, indicate whether each material, system or method listed is relevant to your study. If you are not sure if a list item applies to your research, read the appropriate section before selecting a response.

## Materials & experimental systems

| n/a | Involved in the study |
|---|---|
| ☐ | ☒ Antibodies |
| ☐ | ☒ Eukaryotic cell lines |
| ☒ | ☐ Palaeontology and archaeology |
| ☒ | ☐ Animals and other organisms |
| ☒ | ☐ Clinical data |
| ☒ | ☐ Dual use research of concern |
| ☒ | ☐ Plants |

## Methods

| n/a | Involved in the study |
|---|---|
| ☒ | ☐ ChIP-seq |
| ☒ | ☐ Flow cytometry |
| ☒ | ☐ MRI-based neuroimaging |

## Antibodies

**Antibodies used**

anti-HsNSL1 [mouse monoclonal, clone QL24-1, generated in-house, 1:800]
anti-HsCENP-C (guinea pig polyclonal, MBL-PD030, MBL, 1:1000)
anti-HsNDC80 (mouse monoclonal, clone 9G3, ab3613, Abcam, 1:3000)
anti-CENP-T (rabbit polyclonal, SI0822, generated in-house, 1:1000)
anti-HsKNL1 (rabbit polyclonal, SI0788, generated in-house 1:500)
anti-GFP (rabbit polyclonal, generated in house, 1:1000)
anti-Vinculin (mouse monoclonal, clone hVIN-1, V9131, Sigma-Aldrich, 1:10,000)
anti-guinea pig secondary (goat polyclonal, Alexa Fluor 647, Invitrogen A21450, 1:200)
anti-mouse secondary (goat polyclonal, Alexa Fluor 488-conjugated, Invitrogen A11001, 1:200)
anti-mouse secondary (goat polyclonal, Rhodamine Red-conjugated, Jackson Immuno Research 115-295003, 1:200)
anti-rabbit secondary (donkey polyclonal, Alexa Fluor 488-conjugated, Invitrogen A21206, 1:200)
anti-rabbit secondary (donkey polyclonal, Rhodamine Red-conjugated, Jackson Immuno Research 711-295-152, 1:200)
anti-human secondary (goat polyclonal, Alexa Fluor 647-conjugated, Jackson Immuno Research 109-603-003, 1:200)
anti-rabbit secondary (donkey polyclonal, HRP-conjugated, Amersham NA934, 1:10,000)
anti-mouse secondary (sheep polyclonal, HRP-conjugated, Amersham NXA931, 1:10,000)

**Validation**

Primary antibodies used for western blotting detected proteins with expected sizes confirming the specific binding to the target proteins. Primary antibodies used for IF recognized a signal that disappeared upon RNAi depletion.

-https://www.abcam.com/products/primary-antibodies/hec1hec-antibody-9g3-ab3613.html
-https://www.mblbio.com/bio/g/dtl/A/?pcd=PD030
-https://www.thermofisher.com/antibody/product/A-21450.html?CID=AFLCA-A-21450)
-https://www.thermofisher.com/antibody/product/Goat-anti-Mouse-IgG-H-L-Cross-Adsorbed-Secondary-Antibody-Polyclonal/A-11001
-https://www.jacksonimmuno.com/catalog/products/115-295-003
-https://www.fishersci.com/shop/products/donkey-anti-rabbit-igg-h-l-highly-cross-adsorbed-secondary-antibody-alexa-fluor-488-invitrogen/A21206
-https://www.jacksonimmuno.com/catalog/products/711-295-152
-https://www.jacksonimmuno.com/catalog/products/109-036-003
-https://www.cytivalifesciences.com/en/us/shop/protein-analysis/blotting-and-detection/blotting-standards-and-reagents/amersham-ecl-hrp-conjugated-antibodies-p-06260
-https://www.cytivalifesciences.com/en/us/shop/protein-analysis/blotting-and-detection/blotting-standards-and-reagents/amersham-ecl-hrp-conjugated-antibodies-p-06260

## Eukaryotic cell lines

Policy information about cell lines and Sex and Gender in Research

**Cell line source(s)**

-Sf9 cells (GibcoTMInvitrogen Corporation, Cat. No. 11496-015)
-HeLa cells expressing mCherry-H2B were a gift of Sara Barozzi (Imaging Facility, IFOM-IEO Campus, Milan, Italy) and were not further authenticated.
-DLD1 Flp-In-T-REx also expressing osTIR1 used in this study were received from the laboratory of Don C. Cleveland and were not further authenticated

**Authentication**

None of the cell lines used were authenticated. The original commercial source of the HeLa cell line is unknown

**Mycoplasma contamination**

Cell lines were regularly tested for mycoplasma contamination and the test found to be negative

**Commonly misidentified lines**
(See ICLAC register)

We did not use any misidentified cell line

## Plants

Seed stocks

> Not applicable

Novel plant genotypes

> Not applicable

Authentication

> Not applicable

