## [Peer Review File · Nature Structural & Molecular Biology]

Peer Review Information

Manuscript Title: Structure of the human KMN complex and implications for regulation of its assembly

Corresponding author name(s): Andrea Musacchio

Reviewer Comments & Decisions:

Decision Letter, initial version:

Message: 13th Sep 2023

Dear Professor Musacchio,

Thank you again for submitting your manuscript "Insights into human outer kinetochore assembly and force transmission from a structure-function analysis of the KMN network". I apologise for the delay in responding, which resulted from the delay in obtaining suitable referee reports. Nevertheless, we now have comments (below) from the 3 reviewers who evaluated your paper. In light of these reports, we remain interested in your study and would like to see your response to the comments of the referees, in the form of a revised manuscript.

You will see that though the experts appreciate the structural and mechanistic findings, as well as their potential implications, they raise several important concerns that necessitate addressing in a revised manuscript. More specifically, both reviewer #2 and #3 note the absence of notable controls (e.g. for correct folding and stability of used mutants, potential side-effects of Mis12C reduction, etc). Furthermore, all experts agree that the manuscript at places requires rewriting and clarification (e.g. both reviewer #2 and #3 do not immediately understand the point of experiments in figure 5, whereas reviewer #1 provides extensive guidelines to make the manuscript more intuitive to the reader). Finally, reviewer #2 raises concerns with respect to the novelty of the findings (introduction and point 1) and reviewer #1 notes that the MD data are currently inconclusive; both these remarks require addressing.

Please be sure to address/respond to all concerns of the referees in full in a point-by-point response and highlight all changes in the revised manuscript text file. If you have comments that are intended for editors only, please include those in a separate cover letter.

We are committed to providing a fair and constructive peer-review process. Do not

hesitate to contact us if there are specific requests from the reviewers that you believe are technically impossible or unlikely to yield a meaningful outcome.

We expect to see your revised manuscript within 6 weeks. If you cannot send it within this time, please contact us to discuss an extension; we would still consider your revision, provided that no similar work has been accepted for publication at NSMB or published elsewhere.

Reporting Summary:

Data availability: this journal strongly supports public availability of data. All data used in accepted papers should be available via a public data repository, or alternatively, as Supplementary Information. If data can only be shared on request, please explain why in your Data Availability Statement, and also in the correspondence with your editor. Please note that for some data types, deposition in a public repository is mandatory - more information on our data deposition policies and available repositories can be found below: <https://www.nature.com/nature-research/editorial-policies/reporting-standards#availability-of-data>

[redacted]

Sincerely,

Dimitris Typas
Associate Editor
Nature Structural & Molecular Biology

ORCID: 0000-0002-8737-1319

Referee expertise:

Referee #1: Structural biology of kinetochores

Referee #2: Structural biology of kinetochores, Cellular biology of kinetochores

Referee #3: kinetochore function and biochemistry/cell biology

Reviewers' Comments:

Reviewer #1:

Remarks to the Author:

Comments on Polley et al

The paper reports a "comprehensive" structural and functional analysis of the "microtubule binding machinery of kinetochores". The authors also suggest that it "elucidates a path of MT-generated force transmission." The work integrates a lot of previous structural and functional studies but also provides enough new results to warrant the model in Fig. 7, which contains three noteworthy features not previously appreciated: (1) the relative rigidity of the Ndc80c: Mis13c interface; (2) the "upward" projection of the Knl1: Zwint module; (3) a specific picture for Mis12c autoinhibition (characterized biochemically and mutationally in yeast) and its relief by phosphorylation of two serines.

The new experimental data are as follows. (1) A 4.5 Å resolution structure of an eight-subunit complex that included the four chains of Mis12c, the two chains of the Spc end of Ndc80c, and ZWINT and the C-terminal region of Knl1. The structure shows that there is a substantial interface between Mis12c and Spc24/25, probably imparting a well defined relative directionality to the two rods. Although Knl1 does not interact with Spc24/25, it docks firmly to Mis12c, confirming an earlier, low-resolution EM picture from this group. It also captures a peptide from the C-terminus of Nsl1, as previously shown in a crystal structure, also from this group. The binding of the C-terminus of Dsn1 with Spc24/25 emulates, as expected, the similar interaction with CENP-T; that correspondence has also been shown for yeast (PDB 5T6J). The contact with residues from Nsl1 is new information, although the important Nsl1 residues at the interface were known (from their ref 28). Closure of the two heads of Mis12c shows the structural basis for Mis12c autoinhibition. (2) Mutational analysis in cells and binding experiments in vitro validate most of the conclusions from the structure. These experiments are valuable contributions. They distinguish this paper from the purely structural and biochemical paper co-submitted by Barford and co-workers. (3) Some molecular grafting experiments indicate that both the Dsn1 and the Nsl1 contacts must be present in their wild-type disposition. This result supports the inference mentioned above, of "limited plasticity of the Mis12c: Ndc80c connection". (4) The authors devote a lot of space to a rather inconclusive study of cooperativity in binding Mis12c between Ndc80 and Knl1. More on this below. (5) The ZWINT-Knl1 interaction is missing from the map, as is all of ZWINT. An AF2 model of ZWINT:Knl1 suggests that some residues left out of the ZWINT construct may have led to dissociation or some other reason for this disorder. An AF2 model predicts that the two interact as a parallel coiled-coil. Mutational studies (colocalization in cells) support the

prediction. The co-submitted paper shows that their inferences are solid. (6) AF2 modeling of ZWINT:Knl1 from other species, including yeast, suggests that the two proteins are paralogs, as predictions for ZWINT from many of the other species have one or two RWD domains at their C-termini.

The MS as presented is clearly of interest to the kinetochore field. This reviewer suggests that some aspects may be a bit overstated and others can usefully be truncated or deleted.

Possible overstatements:

(1) This complex is not really the full "microtubule binding machinery". In yeast, that machinery includes (quite crucially) the Dam1c/DASH complex, and in metazoans, it presumably includes the Ska complex. Although the functions of Ska are still not as well characterized as they are for the Dam1c -- the structure of its heterotrimeric assembly unit provides few clues -- their mutually exclusive presence in various taxa (Kops and co-workers) and some similar phosphorylation characteristics make it hard to dismiss its importance. Metazoans have to coordinate (or sense) multiple microtubules attachments to avoid syntely -- not an issue for point centromere yeast and apparently accomplished by other means in Dam1c-containing yeast with small regional centromeres and relatively few afferent microtubules. I suggest some modifications in the detailed list of suggestions, below.

(2) Not sure what "elucidates a path of MT-generated force transmission" means. Presumably the emphasis is on "a path", not "the path". Does the KMN really withstand particularly strong forces? If we assume that there are 20 microtubules attached to a human kinetochore (the conventional estimate) and that each one has six Mis12-based connections through Ndc80, then forces of up to 700 pN per kinetochore (Nicklas top estimate for grasshopper kinetochores) come down to ~5 pN per connection, even neglecting Cnn1-Ndc80 connections and possibly cross-connections from Ska. The key issue, as Biggins and Murray showed, is sensing tension (by Aurora B), not withstanding it.

Suggested deletions: This reviewer finds the entire MD simulation and attendant recruitment experiments too inconclusive to warrant inclusion. The authors are obviously and understandably interested in trying to reconcile the structure with the observations, from previous work and amplified a bit here, that indicate cooperativity, and they are therefore struck by the apparent absence of a direct interface between Knl1 and Spc24/25. But the whole thing turns into a distracting shaggy dog story, and I believe that the paper would be stronger if they deferred this issue until further work leads to a more definitive conclusion. MD (as a field) has really yet to prove that it gives us anything more than a (very) valuable intuitive notion of systems as complex as this one.

Detailed, line-by-line comments. Some of these are trivial comments (e.g., on one slightly bizarre word usage) and others are substantive. They are simply listed from top to bottom of the main text in the MS. Sentences or parts of sentences in quotes are suggested rewrites, except in one or two places, where they refer to an existing sentence that I suggest be deleted or radically shortened.

Title: Given my two points above, I would suggest changing the title (but probably the authors will disagree with me -- which is fine). For example: "Structure of the complete human KMN complex and implications for regulation of its assembly". Even if my

discussion of force transmission is flawed, this structure does not tell us what matters (except that it is quite stable -- hardly a surprise). And "Insights" is a good illustration of a weasel word. The correct reaction is "what insights?".

Abstract. Is the outer layer (as seen in the old thin-section micrographs) really the KMN complex? The spacing between layers is about 500 Å, presumably spanned by Mis12C and part of Ndc80c. Although we don't know all the outer-layer components, they presumably include Ska, perhaps other proteins that associate with the plus tips of microtubules, etc. So it is probably an incorrect assumption that this entire, branched rod is the "outer layer", even if some of it contributes to the dense staining there. (KMN is not, of course, a "network". The authors do a laudable job of avoiding "network" in much of the paper, although we're probably stuck with it in some contexts, as we are with "CCAN", which is hardly a "network" -- as this group and others have shown, it's a pretty solid assembly.)

Suggested edits in the Abstract: Initial definite article can be deleted. Suggest "through" rather than "using" in line 7 (does a molecule have the agency to "use" anything?). More important: modify last sentence to read: "Our work thus reports a comprehensive structural and functional analysis of this part of the kinetochore microtubule-binding machinery and elucidates the path of connections from the chromatin-bound components to the force-generating components." (Don't say "the first" about your own work!)

Introduction

First paragraph: (1) What do they mean by "self-perpetuating"? (2) The last two sentences are unnecessary. This paper is not about centromere position.

Second paragraph. (1) "constitutive" (typo) (2) For sentence beginning "The outer layer", I'd suggest: "Parts of the 10-subunit KMN super-assembly, which connect inward to the CCAN, contribute to the outer layer, probably together with other components." (See remarks about "outer layer", above.)

Third paragraph: (1) "Within the KMN, Mis12c coordinates association of Ndc80c and Knl1c." (Data in the literature and here show that it does -- "emerged" is an incidental historical note, not a truth about the real world. One can add references to the end of the sentence.) (2) "Each of the four subunits of Mis12c has long helical segments and runs". (3) Delete the last clause (it's an unnecessarily pedantic amplification of the meaning of "low resolution").

Fourth paragraph. (1) Do not use "showcasing". It means displaying in a particularly emphatic way, as in a showcase in a museum or store. It would be OK in the following: "The indictments showcase the thoroughgoing corruption and unrestrained criminality of the entire Trump coterie." It's inappropriate in any scientific context I can think of (and needs replacement at a couple of other spots in the MS). Here, my suggestion is "...even more elongated, ~65nm structure -- an overall shaft with globular domains at each end." (2) "... combined contour length of almost 90 nm." (The direct distance between the MT attachment point and the heads of Mis12c might be a bit less, because of the Ndc80c kink, etc. Relatively minor point, given the finding here that the Ndc80c: Mis12c connection is fairly rigid.)

Fifth paragraph: the paragraph needs to be trimmed of some verbal excess, as many readers look first at the last paragraph of the Introduction, to see what the bottom line will be. (1) "Despite the progress, several aspects" (Let the reader decide how crucial.)

(2) "The interfaces that allow Ndc80c and Knl1c to interact with Mis12c both engage RWD domains, but they appear to have no common structural theme." For the rest, the best way to trim is to make actual questions lead to the word "questions" in the penultimate sentence. "How does phosphorylation of Mis12c by Aurora B promote binding to CENP-C and CENP-T? What is the structure of ZWINT and what is the interaction with Knl1 that allows it to incorporate into the KMN complex? How does the Knl1:Mis12c interaction relate spatially to the Mdc12c contact with Spc24:Spc25? Do cooperative, allosteric interactions favor establishing complete KMN assemblies, as suggested by formation and stabilization of KMN complexes in assembly?" Here we answer these questions, by reporting By model fitting and AF2 modeling" (As I point out, I'm not sure that there's an answer to the allosteric question, so perhaps "answer many of these questions" might be more correct, allowing the authors to condense greatly the section I find distracting.)

Results and Discussion.

Third paragraph. Be direct: "Mis12c has a long axis of approximately 22 nm (Fig 1d)"(Could add refs if you'd like: Petrovic; Dimitrova, although the latter is yeast). "It has two globular heads, head 1 and head 2, each a four-helix bundle, which respectively encompass..., and a stalk, which is a compact, parallel helical bundle. The Spc24/Spc25 dimer, two tightly interacting RWD domains, caps the complex"

Fourth paragraph. The number of residues between the segment that binds head 1 and the N-terminal residue of head 2 is very modest in metazoans, so the heads need to come together, even if there's no "glue" -- they need to be "strapped" together, whether or not they are "glued" together. So no need to speculate about charge compensation. In the AF2 model, are the serines docked, so that phosphates would interrupt binding?

Mis12c/Knl1c interaction section. Second paragraph. "The extensive interface includes" (the sentence isn't very interesting and can be deleted -- almost any extensive interface will have interactions in all three categories).

Last sentence of same paragraph: "All mutants weakened the interaction, increasing the KD from" The interaction was just weaker -- 12fold is only about 6 kJ/mole.

Next paragraph, last sentence. Is what you mean the following: "The levels of Mis12c were unaffected, (Fig. 2e-f), indicating that the mutated residues on Knl1 are important for binding to kinetochore-localized Mis12c." The reason for confusion is that Fig. 2e needs a much more explicit caption -- I could not figure out how to interpret it.

Mis12c/Ndc80c interaction section.

First paragraph: "The interaction of Spc24/25 with Dsn1 closely resembles those it has with CENP-T (PDB 3VZA) and with yeast Dsn1 (PDB 5T6J)." (I assume it's fair to add the latter, as it superposes well on the model of CENP-T association from Malvezzi et al and on part of the more extended peptide interaction in 3VZA.)

Structure and role of ZWINT section:

Get rid of "showcase"! ("have" will do just fine)

Last sentences of the section: "The levels of endogenous Knl1 did not These results help validate the AF2 model"

Model section:

Delete "comprehensive" from the section title.

"Starting from the minimal core structure ..., we generated a model of the full KMN complex by incorporating ZWINT, the entire KNL1 subunit, and all four Ndc80c subunits (ref 45), with an overall molecular mass of 593 kDa (Fig. 7). In the absence of its known partners (refs to Musacchio's own work on MELT repeats, Mad1 docking, etc.), KNL1 is predicted to be disordered from its N-terminus up to the point of entry into the helical domain (at residues 1880)." [It isn't "intrinsically disordered", whatever that misleading phrase means; various segments are conditionally ordered, e.g., when phosphorylated and bound with their SAC partners.]

Conclusions:

"We report a comprehensive model" (Don't say "first" about your own work. I think "comprehensive" is OK here, if it is removed elsewhere to avoid overemphasis.)

"First, it suggests a structural mechanism for stabilization of an autoinhibited state of Mis12c by Dsn1 and for switching to an open conformation by phosphorylation, thereby facilitating CENP-C binding." (Petrovic et al and Dimitrova et al already provided "glimpses", so be explicit about what's new here.)

Depending on how the authors react to my suggestion to eliminate the MD and Knl1 cooperativity parts, the second half of the paragraph will need re-formulation. Indeed, that incomplete story dominates the paragraph, distracting the reader from the solid and quite interesting conclusions that make the paper valuable. Slightly greater emphasis on the "upward" (toward the head of Ndc80c) orientation of Knl1c would further restore balance. I'm sure it has not escaped the authors' attention that Mif2 attaches somewhere "upwards" as well.

Reviewer #2:

Remarks to the Author:

The kinetochore is an essential structure on the centromere for accurate chromosome segregation. While the kinetochore is associated with centromeric chromatin on a chromosome, it directly binds to microtubules, indicating that the kinetochore bridges between a chromosome and microtubules for chromosome segregation. The kinetochore is divided into two major groups: one is Constitutive Centromere Associated Network (CCAN), which associated with centromeric chromatin, and another one is Knl1, Mis12, and Ndc80 complexes (KMN network), which directly binds to microtubules. To know the kinetochore architecture, structural analyses are useful, and Musacchio and his colleagues largely contributed to understanding the structure for many subcomplexes of the kinetochore. In previous studies, Musacchio and his colleagues reported high-resolution structures of the Mis12 complex, an engineered Ndc80 complex (Bonsai), and a partial structure of the Knl1 complex, but not the entire structure of the KMN network. In this study, authors characterized Ndc80C and Knl1C interaction sites of Mis12C, based on the

cryo-EM analysis on the entire KMN complex and Alphafold2 predictions combined with mutational analyses. They proposed a structural model for the entire KMN network. Furthermore, using MD simulation, they proposed a model for mutual effect of Ndc80C and Knl1C to stabilize the entire KMN network.

Overall, the proposed structural model of the KMN network is consistent with another work by Barford and his colleagues, which was also submitted to same Journal. This reviewer agrees that this model provides some new interaction surfaces in the entire KMN complex, which would be useful in this research field. Findings based on the cryo-EM structure clarified previous biochemical and cell biology data related to interface between Mis12C and Ndc80C/Knl1C. However, while such a structural information is important, major findings could not add new aspects to published models. In addition, although authors proposed a model for KMN cooperation by MD simulation and cell biology experiments, evidence to support this model was not well presented. Therefore, novelty is limited, and the paper does not appear to give us a depth insight for the kinetochore architecture. To improve quality of this manuscript authors should address several concerns raised by this reviewer.

Major comments.

- 1) The KMN structure in this study reveals a closed packing of head1 and head2 of Mis12C. Alphafold2 predictions propose the Dsn1 loop region binds to head1 and head2, and therefore, it facilitates their closed packing. However, the experimental map lacks clear density for the loop, suggesting unstable binding. Could the authors explain why the Dsn1 loop region isn't well-defined and how head1 and head2 could pack together without stable binding of the Dsn1 loop region? Is there a direct head1-head2 interaction?
- 2) Concerning Fig 4 data, I do not fully agree with interpretation by authors. Rago et al. previously reported reduction of Mis12C levels at the artificial kinetochore upon Knl1 or Nuf2 depletion (Rago et al., *Curr Biol.*, 2015). In this paper, showing the reduction of Knl1 levels upon Ndc80 depletion or Ndc80C reduction upon Knl1 depletion (Fig 4a), authors proposed a cooperation of KMN network. However, considering results by Rago et al., Knl1 reduction by Ndc80 depletion or Ndc80C reduction upon Knl1 depletion could be due to the reduction of Mis12C levels. To reach authors' conclusion, they should quantify Mis12C levels in Knl1 or Ndc80C depleted cells.
- 3) Concerning the section entitled "limited plasticity of the Mis12C/Ndc80C connection", I could not completely follow the logic why they attempted to re-engineer the Mis12C C-terminus, which is schematized in Fig 5a. Please clarify the purpose why they performed these experiments using chimeric mutants, and what is a conclusion in these experiments. The statement "...an observation that suggests that communication between the binding sites for Ndc80C and Knl1 does not involve the Nsl1 C-terminal region..." might not be strictly correct, because these Mis12C mutants possess the Nsl1 C-terminal region either on Mis12 or PMF1 C-termini. In any case, I do not fully understand why authors concluded requirement of Nsl1 C-terminus for Knl1 and Ndc80C localization, based on Fig 5 experiments.
- 4) According to Fig 6g, the authors described "...the levels of endogenous Knl1 did not appear to be affected...ZWINT". However, in the quantitative data in Fig 6-Sup1e, Knl1 levels were 30-40 % in cells expressing ZwintM1 and ZwintM2, compared those in cells expressing ZwintWT. Please explain this inconsistency. The representative data of Knl1 kinetochore localization in Fig 6g need to be replaced with ones consistent with the quantitative data. I also found that remaining endogenous Zwint is still visible in Zwint KD cells (Fig 2-Sup1i). Is it possible that the localization of Knl1 in cells expressing ZwintM1 and ZwintM2 depends on remaining endogenous Zwint?

Minor comments

- 1) For the cryo-EM model shown in Fig 1d, please indicate the locations of the N- and C-termini for each protein.
- 2) Please reorganize the right-side panels in Fig 2a, as shown in Fig 3a. It is difficult to see how specified areas in the entire model (left-side panel) are magnified and rotated. For instance, the right-top and right-bottom panels can be exchanged.
- 3) Please use the same numbering of helices in Fig 2a and Fig 2-Sup1a.
- 4) In the top model of Fig 2-Sup1b, authors extended the cryo-EM structure of Nsl1 using AlphaFold2 prediction as described in the figure legend, but it appears to be misleading. The predicted parts should be clearly indicated. A different color or dotted line can be used.
- 5) Please mention why different Knl1 mutants were used for the in vitro experiments in Fig 2c (KNL1M1, KNL1M2) and for the localization experiments using cultured cells in Fig 2e (KNL1M3, KNL1M4, KNL1M5)?
- 6) KNL1Y2245, a mutation site in mutant KNL1M5 indicated in Fig 2b, is not shown in Fig 2a.
- 7) The label in the second column of Fig 2e is "NSL1", while the label in the fourth column is "MIS12". Which one is correct?
- 8) Please examine protein levels of the EGFP fused Knl1 fragment in Fig 2e and EGFP fused Zwint in Fig 6g.
- 9) Please provide more detailed information about the interaction mechanism between the Nsl1 PVIHL motif and Spc24/Spc25. Explanation on it was quite poor.
- 10) In Fig 3e, the signal intensities of NDC80 in MIS12CM3 appear to be different between the first (black and white image) and third columns (green in the merged color image). These should be same.
- 11) DSN1L336, a mutation site in mutant Mis12CM4 in Fig 3c, is not shown in Fig 3a.
- 12) The SEC elution chromatograms corresponding to the results shown in Fig5-sup1a should be provided. +/- presentation is not possible without data presentation.
- 13) In Fig 5-sup1a, "NSL1Δ207-81" in MIS12CM10-M19 should be corrected to "NSL1Δ207-281", and "DSN1Δ318-56" in MIS12CM10,M11,M14-M17 should be corrected to "DSN1Δ318-356".
- 14) In Fig 5-sup1b, "Mis12C18" and "Mis12C19" should be corrected to "Mis12CM18" and "Mis12CM19", respectively.
- 15) In Fig 7-sup1a-d, please indicate which protein is corresponding to the aligned residues in Y-axis.

Reviewer #3:

Remarks to the Author:

In this manuscript, Polley et al. use single particle cryo-EM to characterize the organization and architecture of the KMN complex. The authors generate a 4.5 Å resolution map of this complex into which they build a model of a subcomplex consisting of the RWD domains of SPC24/SPC25, the RWD domains of KNL1, and the entire Mis12 complex (aided in several cases by Alpha Fold predictions). The authors use a combination of biochemistry (ITC and BLI) and cell biology to validate the protein-protein interfaces detailed in their KMN complex model. The authors then perform a series of molecular dynamic simulations and biochemical experiments to show that the Ndc80 complex stabilizes the interaction of the Mis12 complex with KNL1 and that KNL1 stabilizes the interactions between the Mis12 complex and the Ndc80 complex. The authors also use a

combination of structural data from previous studies, their KMN subcomplex model, and structural predictions to create a model of the entire 10 subunit KMN complex. The work detailed here appears robust and provides a valuable illustration and structural insights into the KMN complex. However, the paper is quite detailed and primarily confirms prior observations instead of providing new paradigms for KMN organization or function, with the feeling that the independent study mentioned in the conclusions section may have accelerated the submission of this manuscript. Overall, the work described here is of high quality, but there are several points that should be addressed.

Major Points:

1. Because the particles suffer from preferential views, the authors should include the directional 3DFSC plot/histogram. The authors should also include a figure with the reconstructed volume colored by local resolution estimates.
2. The authors use several mutant constructs to probe their model biochemically without providing evidence that these introduced mutations do not grossly affect the structure and thus the binding activity of the protein. The authors should comment on or provide data (i.e. SEC chromatogram or CD spectra) that illustrate that the mutant proteins (KNL1 and ZWINT) are folded or mutant complexes (i.e. Mis12CM1) are intact.
3. The authors describe a model where the basic region of DSN1 helps glue together the two heads of Mis12C together to stabilize the closed state and prevent CENP-C binding. Previous work by the Mussachio lab (Petrovic et al. 2016) and others provides support for this mechanism. However, fluorescence polarization experiments performed in Petrovic et al 2016 show that a CENP-C peptide can bind to WT Mis12C with a reasonably high affinity (126 nM). Can the authors comment on how this data fits into their model? Does CENP-C have a higher affinity for Mis12C containing DSN1 with phosphomimetic mutations in S100 and S109? Does DSN1 outcompete CENP-C for binding to MIS12C and can this be relieved with phosphomimetic mutants of DSN1 in vitro?
4. The rationale for performing the experiments in Figure 5 should be clarified. It is unclear why these experiments are being performed and how they contribute to the model the authors propose.

Minor Points:

1. The manuscript could benefit from revision of the text and figure legends. Specifically, there are multiple sentences that are difficult to follow because of their structure. For example, "The Mis12C then continues with a compact parallel helical bundle of the four subunits, the stalk." The use of the passive voice also sounds awkward in many cases – for example "Responsible for these phosphorylation events that explain the stabilization of outer kinetochore assembly during mitosis is the prominent mitotic kinase Aurora B." is instead of "The mitotic kinase Aurora B is responsible"
2. In Figure 3F, it is unclear how the authors obtained the intensity values for Mis12C.
3. In the sentence "The wild type KNL1 localized normally ..." the authors cite Figure 2C, but I think they intended to cite Figure 2B?
4. The authors should clarify in the figure that the construct of KNL1 described in Figure 2E spans residues 2026-2342.

5. In Figure 6G it looks like there is variable expression of each of the ZWINT constructs. Can the authors show uniform expression by western blot?

6. In figure 7 the authors should detail in the caption or in the figure which parts of the model are derived from data and which are derived from structural predictions.

Author Rebuttal to Initial comments

Reviewer #1:

The paper reports a "comprehensive" structural and functional analysis of the "microtubule binding machinery of kinetochores". The authors also suggest that it "elucidates a path of MT-generated force transmission." The work integrates a lot of previous structural and functional studies but also provides enough new results to warrant the model in Fig. 7, which contains three noteworthy features not previously appreciated: (1) the relative rigidity of the Ndc80c:Misc12c interface; (2) the "upward" projection of the Knl1:Zwint module; (3) a specific picture for Mis12c autoinhibition (characterized biochemically and mutationally in yeast) and its relief by phosphorylation of two serines.

The new experimental data are as follows. (1) A 4.5 Å resolution structure of an eight-subunit complex that included the four chains of Mis12c, the two chains of the Spc end of Ndc80c, and ZWINT and the C-terminal region of Knl1. The structure shows that there is a substantial interface between Mis12c and Spc24/25, probably imparting a well defined relative directionality to the two rods. Although Knl1 does not interact with Spc24/25, it docks firmly to Misc12c, confirming an earlier, low-resolution EM picture from this group. It also captures a peptide from the C-terminus of Nsl1, as previously shown in a crystal structure, also from this group. The binding of the C-terminus of Dsn1 with Spc24/25 emulates, as expected, the similar interaction with CENP-T; that correspondence has also been shown for yeast (PDB 5T6J). The contact with residues from Nsl1 is new information, although the important Nsl1 residues at the interface were known (from their ref 28). Closure of the two heads of Mis12c shows the structural basis for Mis12c autoinhibition. (2) Mutational analysis in cells and binding experiments in vitro validate most of the conclusions from the structure. These experiments are valuable contributions. They distinguish this paper from the purely structural and biochemical paper co-submitted by Barford and co-workers. (3) Some molecular grafting experiments indicate that both the Dsn1 and the Nsl1 contacts must be present in their wild-type disposition. This result supports the inference mentioned above, of "limited plasticity of the Mis12c:Ndc80c connection". (4) The authors devote a lot of space to a rather inconclusive study of cooperativity in binding Mis12c between Ndc80 and Knl1. More on this below. (5) The ZWINT-Knl1 interaction is missing from the map, as is all of ZWINT. An AF2 model of ZWINT:Knl1 suggests that some residues left out of the ZWINT construct may have led to dissociation or some other reason for this disorder. An AF2 model predicts that the two interact as a parallel coiled-coil. Mutational studies (colocalization in cells) support the prediction. The co-submitted paper shows that their inferences are solid. (6) AF2 modeling of ZWINT:Knl1 from other species, including yeast, suggests that the two proteins are paralogs, as predictions for ZWINT from many of the other species have one or two RWD domains at their C-termini.

The MS as presented is clearly of interest to the kinetochore field. This reviewer suggests that some aspects may be a bit overstated and others can usefully be truncated or deleted.

We are grateful for this supportive and helpful review

Possible overstatements:

(1) This complex is not really the full "microtubule binding machinery". In yeast, that machinery includes (quite crucially) the Dam1c/DASH complex, and in metazoans, it presumably includes the Ska complex. Although the functions of Ska are still not as well characterized as they are for the Dam1c -- the structure of its heterotrimeric assembly unit provides few clues -- their mutually exclusive presence in various taxa (Kops and co-workers) and some similar phosphorylation characteristics make it hard to dismiss its importance. Metazoans have to coordinate (or sense) multiple microtubule attachments to avoid syntely -- not an issue for point centromere yeast and apparently accomplished by other means in Dam1c-containing yeast with small regional

centromeres and relatively few afferent microtubules. I suggest some modifications in the detailed list of suggestions, below.

Thank you for these remarks. We answer them specifically later in our response

(2) Not sure what "elucidates a path of MT-generated force transmission" means. Presumably the emphasis is on "a path", not "the path". Does the KMN really withstand particularly strong forces? If we assume that there are 20 microtubules attached to a human kinetochore (the conventional estimate) and that each one has six Mis12-based connections through Ndc80, then forces of up to 700 pN per kinetochore (Nicklas top estimate for grasshopper kinetochores) come down to ~5 pN per connection, even neglecting Cnn1-Ndc80 connections and possibly cross-connections from Ska. The key issue, as Biggins and Murray showed, is sensing tension (by Aurora B), not withstanding it.

We agree with the reviewer that forces may be in a range of 5-10 pN per linkage. We completely agree with the reviewer that Aurora B is crucial for force sensing. We also argue that we are not yet in a position to conclude that force sensing in the kinetochore is exclusively through Aurora B. For instance, force may modify the conformation of the Ndc80 complex so that it turns into a poorer substrate of Aurora B. What levels of force are required is an open question in our view.

Suggested deletions: This reviewer finds the entire MD simulation and attendant recruitment experiments too inconclusive to warrant inclusion. The authors are obviously and understandably interested in trying to reconcile the structure with the observations, from previous work and amplified a bit here, that indicate cooperativity, and they are therefore struck by the apparent absence of a direct interface between Knl1 and Spc24/25. But the whole thing turns into a distracting shaggy dog story, and I believe that the paper would be stronger if they deferred this issue until further work leads to a more definitive conclusion. MD (as a field) has really yet to prove that it gives us anything more than a (very) valuable intuitive notion of systems as complex as this one.

We have taken this suggestion on board and have now greatly reduced the weight of the MD simulation in our interpretation of the data.

Detailed, line-by-line comments. Some of these are trivial comments (e.g., on one slightly bizarre word usage) and others are substantive. They are simply listed from top to bottom of the main text in the MS. Sentences or parts of sentences in quotes are suggested rewrites, except in one or two places, where they refer to an existing sentence that I suggest be deleted or radically shortened.

Title: Given my two points above, I would suggest changing the title (but probably the authors will disagree with me -- which is fine). For example: "Structure of the complete human KMN complex and implications for regulation of its assembly". Even if my discussion of force transmission is flawed, this structure does not tell us what matters (except that it is quite stable -- hardly a surprise). And "Insights" is a good illustration of a weasel word. The correct reaction is "what insights?".

We thank the reviewer for this suggestion of a new title, that we have incorporated in the revised manuscript

Abstract. Is the outer layer (as seen in the old thin-section micrographs) really the KMN complex? The spacing between layers is about 500 Å, presumably spanned by Mis12C and part of Ndc80c.

Although we don't know all the outer-layer components, they presumably include Ska, perhaps other proteins that associate with the plus tips of microtubules, etc. So it is probably an incorrect assumption that this entire, branched rod is the "outer layer", even if some of it contributes to the dense staining there. (KMN is not, of course, a "network". The authors do a laudable job of avoiding "network" in much of the paper, although we're probably stuck with it in some contexts, as we are with "CCAN", which is hardly a "network" -- as this group and others have shown, it's a pretty solid assembly.)

We have now modified the abstract as proposed by the reviewer. Specifically, we clarify that the KMN is one of the main components of the outer layer and we tried to avoid using "network" and replaced it with "assembly". All changes are highlighted in the text collated at the end of this rebuttal.

Suggested edits in the Abstract: Initial definite article can be deleted.

Deleted

Suggest "through" rather than "using" in line 7 (does a molecule have the agency to "use" anything?).

Done

More important: modify last sentence to read: "Our work thus reports a comprehensive structural and functional analysis of this part of the kinetochore microtubule-binding machinery and elucidates the path of connections from the chromatin-bound components to the force-generating components." (Don't say "the first" about your own work!)

Thank you for this suggestion, which we have adopted.

Introduction

First paragraph: (1) What do they mean by "self-perpetuating"? (2) The last two sentences are unnecessary. This paper is not about centromere position.

We removed "self-perpetuating" (it was meant to imply that the kinetochore "encodes" its own duplication, but we agree that it was cryptic. We also removed the last two sentences of the paragraph as suggested.

Second paragraph. (1) "constitutive" (typo) (2) For sentence beginning "The outer layer", I'd suggest: "Parts of the 10-subunit KMN super-assembly, which connect inward to the CCAN, contribute to the outer layer, probably together with other components." (See remarks about "outer layer", above.)

Thanks, we have corrected the typo and replaced the sentence as suggested.

Third paragraph: (1) "Within the KMN, Mis12c coordinates association of Ndc80c and Knl1c." (Data in the literature and here show that it does -- "emerged" is an incidental historical note, not a truth about the real world. One can add references to the end of the sentence.) (2) "Each of the four subunits of Mis12c has long helical segments and runs". (3) Delete the last clause (it's an unnecessarily pedantic amplification of the meaning of "low resolution").

We followed all three helpful suggestions

Fourth paragraph. (1) Do not use "showcasing". It means displaying in a particularly emphatic way, as in a showcase in a museum or store. It would be OK in the following: "The indictments showcase the thoroughgoing corruption and unrestrained criminality of the entire Trump coterie." It's inappropriate in any scientific context I can think of (and needs replacement at a couple of other spots in the MS). Here, my suggestion is "...even more elongated, ~65nm structure -- an overall shaft with globular domains at each end." (2) "... combined contour length of almost 90 nm." (The direct distance between the MT attachment point and the heads of Mis12c might be a bit less, because of the Ndc80c kink, etc. Relatively minor point, given the finding here that the Ndc80c: Mis12c connection is fairly rigid.)

We removed all instances of "showcasing" and incorporated point 2.

Fifth paragraph: the paragraph needs to be trimmed of some verbal excess, as many readers look first at the last paragraph of the Introduction, to see what the bottom line will be. (1) "Despite the progress, several aspects" (Let the reader decide how crucial.) (2) "The interfaces that allow Ndc80c and Knl1c to interact with Mis12c both engage RWD domains, but they appear to have no common structural theme." For the rest, the best way to trim is to make actual questions lead to the word "questions" in the penultimate sentence. "How does phosphorylation of Mis12c by Aurora B promote binding to CENP-C and CENP-T? What is the structure of ZWINT and what is the interaction with Knl1 that allows it to incorporate into the KMN complex? How does the Knl1: Msc12c interaction relate spatially to the Mdc12c contact with Spc24: Spc25? Do cooperative, allosteric interactions favor establishing complete KMN assemblies, as suggested by formation and stabilization of KMN complexes in assembly?" Here we answer these questions, by reporting By model fitting and AF2 modeling" (As I point out, I'm not sure that there's an answer to the allosteric question, so perhaps "answer many of these questions" might be more correct, allowing the authors to condense greatly the section I find distracting.)

Thank you for these helpful suggestions, which we have incorporated in the revised manuscript.

Results and Discussion.

Third paragraph. Be direct: "Mis12c has a long axis of approximately 22 nm (Fig 1d)" (Could add refs if you'd like: Petrovic; Dimitrova, although the latter is yeast). "It has two globular heads, head 1 and head 2, each a four-helix bundle, which respectively encompass..., and a stalk, which is a compact, parallel helical bundle. The Spc24/ Spc25 dimer, two tightly interacting RWD domains, caps the complex"

Thank you, we incorporated this suggestion.

Fourth paragraph. The number of residues between the segment that binds head 1 and the N-terminal residue of head 2 is very modest in metazoans, so the heads need to come together, even if there's no "glue" -- they need to be "strapped" together, whether or not they are "glued" together. So no need to speculate about charge compensation. In the AF2 model, are the serines docked, so that phosphates would interrupt binding?

Deletion of the Dsn1 phosphorylation loop increases the binding affinity of Mis12C for CENP-C (and CENP-T), implying that the non-phosphorylated form of the loop stabilizes the closed conformation of the head. Indeed, AF2 predicts engagement of all serines in the stabilization of

the phosphorylation loop, and extensive interactions of the entire phosphorylation loop with Head1 that would stabilize its otherwise modest interaction with Head2. For these reasons, we think that the interpretation that the phosphorylation loop contributes to keep the two heads in the closed conformation may be correct.

Mis12c/Knl1c interaction section. Second paragraph. "The extensive interface includes" (the sentence isn't very interesting and can be deleted -- almost any extensive interface will have interactions in all three categories).

Done

Last sentence of same paragraph: "All mutants weakened the interaction, increasing the KD from" The interaction was just weaker -- 12fold is only about 6 kJ/mole.

Done

Next paragraph, last sentence. Is what you mean the following: "The levels of Mis12c were unaffected, (Fig. 2e-f), indicating that the mutated residues on Knl1 are important for binding to kinetochore-localized Mis12c." The reason for confusion is that Fig. 2e needs a much more explicit caption -- I could not figure out how to interpret it.

Yes, we meant precisely what the reviewer understood and have changed the text to make it more easily understandable. We have also modified the caption.

Mis12c/Ndc80c interaction section.

First paragraph: "The interaction of Spc24/25 with Dsn1 closely resembles those it has with CENP-T (PDB 3VZA) and with yeast Dsn1 (PDB 5T6J)." (I assume it's fair to add the latter, as it superposes well on the model of CENP-T association from Malvezzi et al and on part of the more extended peptide interaction in 3VZA.)

Thank you for this suggestion, which we have followed

Structure and role of ZWINT' section:

Get rid of "showcase"! ("have" will do just fine)

Done!

Last sentences of the section: "The levels of endogenous Knl1 did not These results help validate the AF2 model"

As suggested, we removed the unnecessary adverbs at the beginning of each sentence.

Model section:

Delete "comprehensive" from the section title.

Deleted

"Starting from the minimal core structure ..., we generated a model of the full KMN complex by incorporating ZWINT, the entire KNL1 subunit, and all four Ndc80c subunits (ref 45), with an overall molecular mass of 593 kDa (Fig. 7). In the absence of its known partners (refs to Musacchio's own work on MELT repeats, Mad1 docking, etc.), KNL1 is predicted to be disordered from its N-terminus up to the point of entry into the helical domain (at residues 1880)." [It isn't "intrinsically disordered", whatever that misleading phrase means; various segments are conditionally ordered, e.g., when phosphorylated and bound with their SAC partners.]

Thank you, we have modified the text accordingly

Conclusions:

"We report a comprehensive model" (Don't say "first" about your own work. I think "comprehensive" is OK here, if it is removed elsewhere to avoid overemphasis.)
Done

"First, it suggests a structural mechanism for stabilization of an autoinhibited state of Mis12c by Dsn1 and for switching to an open conformation by phosphorylation, thereby facilitating CENP-C binding." (Petrovic et al and Dimitrova et al already provided "glimpses", so be explicit about what's new here.)

Done

Depending on how the authors react to my suggestion to eliminate the MD and Knl1 cooperativity parts, the second half of the paragraph will need re-formulation. Indeed, that incomplete story dominates the paragraph, distracting the reader from the solid and quite interesting conclusions that make the paper valuable. Slightly greater emphasis on the "upward" (toward the head of Ndc80c) orientation of Knl1c would further restore balance. I'm sure it has not escaped the authors' attention that Mif2 attaches somewhere "upwards" as well.

Thank you for this suggestion. In this revised manuscript, we have now considerably reduced the weight of the MD simulations and given slightly more emphasis to the upward orientation of Knl1C in our Conclusions.

Reviewer #2:

The kinetochore is an essential structure on the centromere for accurate chromosome segregation. While the kinetochore is associated with centromeric chromatin on a chromosome, it directly binds to microtubules, indicating that the kinetochore bridges between a chromosome and microtubules for chromosome segregation. The kinetochore is divided into two major groups: one is Constitutive Centromere Associated Network (CCAN), which associated with centromeric chromatin, and another one is Knl1, Mis12, and Ndc80 complexes (KMN network), which directly binds to microtubules. To know the kinetochore architecture, structural analyses are useful, and Musacchio and his colleagues largely contributed to understanding the structure for many subcomplexes of the kinetochore. In previous studies, Musacchio and his colleagues reported high-resolution structures of the Mis12 complex, an engineered Ndc80 complex (Bonsai), and a partial structure of the Knl1 complex, but not the entire structure of the KMN network. In this study, authors characterized Ndc80C and Knl1C interaction sites of Mis12C, based on the cryo-EM analysis on the entire KMN complex and Alphafold2 predictions combined with mutational analyses. They proposed a structural model for the entire KMN network. Furthermore, using MD

simulation, they proposed a model for mutual effect of Ndc80C and Knl1C to stabilize the entire KMN network.

Overall, the proposed structural model of the KMN network is consistent with another work by Barford and his colleagues, which was also submitted to same Journal. This reviewer agrees that this model provides some new interaction surfaces in the entire KMN complex, which would be useful in this research field. Findings based on the cryo-EM structure clarified previous biochemical and cell biology data related to interface between Mis12C and Ndc80C/Knl1C. However, while such a structural information is important, major findings could not add new aspects to published models. In addition, although authors proposed a model for KMN cooperation by MD simulation and cell biology experiments, evidence to support this model was not well presented. Therefore, novelty is limited, and the paper does not appear to give us a depth insight for the kinetochore architecture. To improve quality of this manuscript authors should address several concerns raised by this reviewer.

We thank for the reviewer for his/her careful reading and insightful comments on this manuscript. While the reviewer praises the work, he/she also identified some substantive concerns. We have now carefully considered the reviewers concerns and believe the revised manuscript addresses most or all of them, as detailed below.

Major comments.

1) The KMN structure in this study reveals a closed packing of head1 and head2 of Mis12C. AlphaFold2 predictions propose the Dsn1 loop region binds to head1 and head2, and therefore, it facilitates their closed packing. However, the experimental map lacks clear density for the loop, suggesting unstable binding. Could the authors explain why the Dsn1 loop region isn't well-defined and how head1 and head2 could pack together without stable binding of the Dsn1 loop region? Is there a direct head1-head2 interaction?

This echoes a request for clarification by Reviewer 1. Deletion of the Dsn1 phosphorylation loop increases the binding affinity of Mis12C for CENP-C (and CENP-T), implying that the non-phosphorylated form of the DSN1 phosphorylation loop stabilizes the closed conformation of the head. We have re-written this section of the manuscript to improve its readability.

2) Concerning Fig 4 data, I do not fully agree with interpretation by authors. Rago et al. previously reported reduction of Mis12C levels at the artificial kinetochore upon Knl1 or Nuf2 depletion (Rago et al., *Curr Biol.*, 2015). In this paper, showing the reduction of Knl1 levels upon Ndc80 depletion or Ndc80C reduction upon Knl1 depletion (Fig 4a), authors proposed a cooperation of KMN network. However, considering results by Rago et al., Knl1 reduction by Ndc80 depletion or Ndc80C reduction upon Knl1 depletion could be due to the reduction of Mis12C levels. To reach authors' conclusion, they should quantify Mis12C levels in Knl1 or Ndc80C depleted cells.

The reviewer makes an important point that we had not considered while working on the original submission. We have now repeated all the co-dependency experiments and measured in addition the levels of Mis12C. As predicted by the results of Rago et al. and in agreement with the reviewer's warning, both KNL1 and Ndc80C lead to a reduction of the kinetochore levels of Mis12C (measured through NSL1). These experiments, with their quantifications, are now included in Figure 4 and replace our previous data. We refer to the Rago *et al.* experiments in this context (the paper was already cited in the original manuscript).

These new experiments are also somehow connected with a request by reviewer 1 to reduce the "weight" of our MD simulations. We have now had time to discuss this, and decided to shorten considerably the description of our MD results, moving the majority to the Methods and

Supplemental material for interested readers. In essence, the MD section in the main text is now limited to one panel in Figure 4 that reports a prediction from MD that the interactions with Ndc80C and Knl1C stabilize the Mis12C. However, we do not argue any longer that the reciprocal stabilization of the two binding sites may explain the changes in localization levels. In other words, we report an interesting prediction of the MD dynamics simulation, but we do not attribute any specific biological significance to the observations.

3) Concerning the section entitled “limited plasticity of the Mis12C/Ndc80C connection”, I could not completely follow the logic why they attempted to re-engineer the Mis12C C-terminus, which is schematized in Fig 5a. Please clarify the purpose why they performed these experiments using chimeric mutants, and what is a conclusion in these experiments. The statement “...an observation that suggests that communication between the binding sites for Ndc80C and Knl1 does not involve the Nsl1 C-terminal region...” might not be strictly correct, because these Mis12C mutants possess the Nsl1 C-terminal region either on Mis12 or PMF1 C-termini. In any case, I do not fully understand why authors concluded requirement of Nsl1 C-terminus for Knl1 and Ndc80C localization, based on Fig 5 experiments.

This concern echoes a similar concern of reviewer 3. We have now extensively rewritten this section to clarify our thinking. We hope that the reviewer will find this section improved.

4) According to Fig 6g, the authors described “...the levels of endogenous Knl1 did not appear to be affected...ZWINT”. However, in the quantitative data in Fig 6-Sup1e, Knl1 levels were 30-40 % in cells expressing ZwintM1 and ZwintM2, compared those in cells expressing ZwintWT. Please explain this inconsistency. The representative data of Knl1 kinetochore localization in Fig 6g need to be replaced with ones consistent with the quantitative data. I also found that remaining endogenous Zwint is still visible in Zwint KD cells (Fig 2-Sup1i). Is it possible that the localization of Knl1 in cells expressing ZwintM1 and ZwintM2 depends on remaining endogenous Zwint?

The reviewer is correct and we apologize for the inconsistency in the original presentation of our data. Depletion of ZWINT does indeed destabilize kinetochore KNL1 (and Ndc80C) to a very significant extent and we now make this clear with quantifications (Extended Data Figure 10e-g) and in the text. These data also demonstrate that the depletion of ZWINT is very penetrant. We have also included more representative KNL1 and ZWINT images (Figure 6g). We also monitored the expression of the ZWINT transgenes (Extended Data Figure 10h) and demonstrate that the mutants do not rescue the levels of KNL1 beyond the residual levels in ZWINT RNAi-treated cells.

Minor comments

1) For the cryo-EM model shown in Fig 1d, please indicate the locations of the N- and C-termini for each protein.

We have modified the figure as suggested and indicate the N- and C-termini for all displayed subunits.

2) Please reorganize the right-side panels in Fig 2a, as shown in Fig 3a. It is difficult to see how specified areas in the entire model (left-side panel) are magnified and rotated. For instance, the right-top and right-bottom panels can be exchanged.

Thank you for this suggestion. We have rearranged the presentation as suggested by the reviewer.

3) Please use the same numbering of helices in Fig 2a and Fig 2-Sup1a.

We apologize for the confusion. We have now made numbering homogenous.

4) In the top model of Fig 2-Sup1b, authors extended the cryo-EM structure of Nsl1 using AlphaFold2 prediction as described in the figure legend, but it appears to be misleading. The predicted parts should be clearly indicated. A different color or dotted line can be used.

We were unclear about this comment. We only used the AF2 model in panel B and we report this fact in the legend. The reciprocal position of the heads in the experimental map, illustrated in the upper panels, is sufficiently clear.

5) Please mention why different Knl1 mutants were used for the *in vitro* experiments in Fig 2c (KNL1M1, KNL1M2) and for the localization experiments using cultured cells in Fig 2e (KNL1M3, KNL1M4, KNL1M5)?

We have now re-written the paragraph describing these results. We clarify that the three mutants studied *in vivo* were not studied *in vitro*, and vice versa. The reason is that we wanted to use two orthogonal approaches for measuring the effects of mutations, but subjecting every mutant to the whole battery of approaches we used here would have been very cumbersome.

6) KNL1Y2245, a mutation site in mutant KNL1M5 indicated in Fig 2b, is not shown in Fig 2a.

Good eye! We now show the residue.

7) The label in the second column of Fig 2e is "NSL1", while the label in the fourth column is "MIS12". Which one is correct?

Thank you for noticing. NSL1 is the correct label and we have corrected the figure accordingly.

8) Please examine protein levels of the EGFP fused Knl1 fragment in Fig 2e and EGFP fused Zwint in Fig 6g.

We have now included the expression levels of these constructs as panels Extended Data Figure 4g and Extended Data Figure 10h.

9) Please provide more detailed information about the interaction mechanism between the Nsl1 PVIHL motif and Spc24/Spc25. Explanation on it was quite poor.

We have included an explanation of the contribution of this motif to the binding interface.

10) In Fig 3e, the signal intensities of NDC80 in MIS12CM3 appear to be different between the first (black and white image) and third columns (green in the merged color image). These should be same.

Thank you, this has been corrected.

11) DSN1L336, a mutation site in mutant Mis12CM4 in Fig 3c, is not shown in Fig 3a.

Thank you for pointing this out. The residue is “barely shown” because in both panels it ends up behind the helix and is not clearly visible. We have nevertheless added a label to indicate its position.

12) The SEC elution chromatograms corresponding to the results shown in Fig5-sup1a should be provided. +/- presentation is not possible without data presentation.

We now provide the SEC elution chromatograms for all constructs in Extended Data Figure 9

13) In Fig 5-sup1a, “NSL1Δ207-81” in MIS12CM10-M19 should be corrected to “NSL1Δ207-281”, and “DSN1Δ318-56” in MIS12CM10,M11,M14-M17 should be corrected to “DSN1Δ318-356”.

Done. This is now Extended Data Figure 8

14) In Fig 5-sup1b, “Mis12C18” and “Mis12C19” should be corrected to “Mis12CM18” and “Mis12CM19”, respectively.

Corrected, thank you for noticing (again, this is now Extended Data Figure 8)

15) In Fig 7-sup1a-d, please indicate which protein is corresponding to the aligned residues in Y-axis.

We apologize for the confusion. The “title” line reports the succession of polypeptides that were included in the prediction and their boundaries. We have now extended this for the Mis12C (where we did not report the succession of subunits) and have added a statement to the legend that clarified this point.

Reviewer #3:

In this manuscript, Polley et al. use single particle cryo-EM to characterize the organization and architecture of the KMN complex. The authors generate a 4.5 Å resolution map of this complex into which they build a model of a subcomplex consisting of the RWD domains of SPC24/SPC25, the RWD domains of KNL1, and the entire Mis12 complex (aided in several cases by Alpha Fold predictions). The authors use a combination of biochemistry (ITC and BLI) and cell biology to validate the protein-protein interfaces detailed in their KMN complex model. The authors then perform a series of molecular dynamic simulations and biochemical experiments to show that the Ndc80 complex stabilizes the interaction of the Mis12 complex with KNL1 and that KNL1 stabilizes the interactions between the Mis12 complex and the Ndc80 complex. The authors also use a combination of structural data from previous studies, their KMN subcomplex model, and structural predictions to create a model of the entire 10 subunit KMN complex. The work detailed here appears robust and provides a valuable illustration and structural insights into the KMN complex. However, the paper is quite detailed and primarily confirms prior observations instead of providing new paradigms for KMN organization or function, with the feeling that the independent study mentioned in the conclusions section may have accelerated the submission of this manuscript. Overall, the work described here is of high quality, but there are several points that should be addressed.

Major Points:

1. Because the particles suffer from preferential views, the authors should include the directional 3DFSC plot/histogram. The authors should also include a figure with the reconstructed volume colored by local resolution estimates.

We thank the reviewer for this suggestion. As suggested, we have now included a 3D FSC plot that shows a broad resolution range over different angles. We have also included a figure showing a local-resolution-colored (and -filtered) reconstructed volume which shows that the highest resolution around the interaction sites of the KNL1 and SPC24/SPC25 modules with the MIS12 portion.

2. The authors use several mutant constructs to probe their model biochemically without providing evidence that these introduced mutations do not grossly affect the structure and thus the binding activity of the protein. The authors should comment on or provide data (i.e. SEC chromatogram or CD spectra) that illustrate that the mutant proteins (KNL1 and ZWINT) are folded or mutant complexes (i.e Mis12CM1) are intact.

We have now included the SEC chromatograms for the mutants used in our study. They are collected in Extended Data Figures 3 and 9

3. The authors describe a model where the basic region of DSN1 helps glue together the two heads of Mis12C together to stabilize the closed state and prevent CENP-C binding. Previous work by the Mussachio lab (Petrovic et al. 2016) and others provides support for this mechanism. However, fluorescence polarization experiments performed in Petrovic et al 2016 show that a CENP-C peptide can bind to WT Mis12C with a reasonably high affinity (126 nM). Can the authors comment on how this data fits into their model? Does CENP-C have a higher affinity for Mis12C containing DSN1 with phosphomimetic mutations in S100 and S109? Does DSN1 outcompete CENP-C for binding to MIS12C and can this be relieved with phosphomimetic mutants of DSN1 in vitro?

We investigated this issue in our previous work, and in particular in Petrovic *et al.* 2016 and Walstein *et al.* 2021 (for CENP-C and CENP-T, respectively). CENP-C has ~100-fold higher affinity for a Mis12C where the phosphorylation loop has been deleted. In neither study we had created a phosphor-mimetic mutant, but we had rather mutated three positively charged residues neighboring one of the phosphorylation targets and found an increase in binding activity, indicating that residues in the phosphorylation loop contribute to stabilization of the closed state. For a regulated interaction, a 100-fold change in binding affinity is plenty, we think. We have rewritten this section of the manuscript to make it more comprehensible.

4. The rationale for performing the experiments in Figure 5 should be clarified. It is unclear why these experiments are being performed and how they contribute to the model the authors propose.

This concern echoes a similar concern from reviewer 2. We have not extensively rewritten this section to clarify the thinking behind these experiments. We hope that the reviewer will find the description clearer.

Minor Points:

1. The manuscript could benefit from revision of the text and figure legends. Specifically, there are multiple sentences that are difficult to follow because of their structure. For example, “The Mis12C then continues with a compact parallel helical bundle of the four subunits, the stalk.” The use of the passive voice also sounds awkward in many cases – for example “Responsible for these phosphorylation events that explain the stabilization of outer kinetochore assembly during mitosis is the prominent mitotic kinase Aurora B.” is instead of “The mitotic kinase Aurora B is responsible”

Thank you for these suggestions. We have simplified both sentences and more generally, also with directions from Reviewer 1, we have significantly streamlined the narrative.

2. In Figure 3F, it is unclear how the authors obtained the intensity values for Mis12C.

We now clarify in the legend that the intensity values for Mis12C were obtained from samples that were collected in parallel. The reason for this is that the anti-Ndc80C and anti NSL1 antibodies are both mice monoclonals.

3. In the sentence “The wild type KNL1 localized normally ...” the authors cite Figure 2C, but I think they intended to cite Figure 2B?

Corrected, thank you

4. The authors should clarify in the figure that the construct of KNL1 described in Figure 2E spans residues 2026-2342.

Thank you for pointing this out. We have now included this missing information to the figure

5. In Figure 6G it looks like there is variable expression of each of the ZWINT constructs. Can the authors show uniform expression by western blot?

We have now included the WB control to Extended Data Figure 10h. The two mutants are expressed at similar levels, but at somewhat lower levels than wild type ZWINT. This may be due to limited stability of the mutants when their interaction with KNL1 is affected.

6. In figure 7 the authors should detail in the caption or in the figure which parts of the model are derived from data and which are derived from structural predictions.

We have now added to Figure 7 the map shown in Figure 1C for comparison and indicate in the legend that KMN segments outside of this map are predicted.

Decision Letter, first revision:

Message: Our ref: NSMB-A48075A

1st Dec 2023

Dear Professor Musacchio,

Thank you for submitting your revised manuscript "Structure of the human KMN complex and implications for regulation of its assembly" (NSMB-A48075A). It has now been seen by the original referees and their comments are below. The reviewers find that the paper has improved in revision, and therefore we'll be happy to accept it in principle in Nature Structural & Molecular Biology, pending minor revisions to satisfy the referees' final requests (please see points raised by reviewer #2 which will need to be textually fixed) and to comply with our editorial and formatting guidelines.

We are now performing detailed checks on your paper and will send you a checklist detailing our editorial and formatting requirements in about two weeks. Please do not upload the final materials and make any revisions until you receive this additional information from us.

To facilitate our work at this stage, it is important that we have a copy of the main text as a word file. If you could please send along a word version of this file as soon as possible, we would greatly appreciate it; please make sure to copy the NSMB account (cc'ed above).

Sincerely,

Dimitris Typas
Associate Editor
Nature Structural & Molecular Biology
ORCID: 0000-0002-8737-1319

Reviewer #1 (Remarks to the Author):

The authors have thoughtfully incorporated nearly all of my suggestions -- I'm pleased that my effort to make detailed editorial suggestions was worthwhile, as the MS now reads more smoothly. The reduced emphasis on MD simulations helps substantially to avoid what I called a "shaggy dog story" in my original review. The Conclusion now focuses, succinctly, on the structural features established by the new data in this paper and on the functional consequences of those findings.

Reviewer #2 (Remarks to the Author):

This is a revised MS for the structure of KMN and this reviewer found that authors did additional efforts for revision. Authors have faced each of concerns raised by this reviewer and responded each of them constructively. Therefore, the MS has been improved well. However, there are still two remaining concerns, which authors should address:

1. In the section entitled "Structure and role of ZWINT", authors now presented quantitative data (Extended Data Figure 10e-g, i-j). The data indicated that KNL1 levels were reduced upon ZWINT knockdown or expression of ZWINT mutants, suggesting that the KNL1-ZWINT interaction is crucial for KNL1 localization to the kinetochore. However, in lines 338-339, the authors still described, "The levels of endogenous KNL1 did not appear to be affected, suggesting that KNL1 can reach kinetochores independently of ZWINT". This description is not consistent with their own quantitative data. Authors should describe their conclusion, based on the data. In addition, while I appreciate structure presentation of KNL1-ZWINT, it remains unclear for the role of ZWINT in the current MS. It might be better to revise section title. Alternatively, authors can describe the role of ZWINT with the clear presentation.

2. On my previous comment as Minor Comment 4, authors responded "We only used the AF2 model in panel B...". However, I guess that the top model of Extended Data Figure 3b (previously Fig2-Sup1b) was created by a combination of the cryo-EM structure of NSL1 with AlphaFold2 predictions, because authors described in the figure legend: "The model of the NSL1 chain in this representation includes extensions, built by AlphaFold2, to the helical segment shown in Figure 2a". Is this model created by only AlphaFold2 or combination of the Cryo-EM structure with AlphaFold2? Please clarify this. If this model was created by the combination, it would be better to indicate which parts were extended with AlphaFold2, using a different color or a dotted line.

Reviewer #3 (Remarks to the Author):

In this revised manuscript, the authors have addressed all of the major and minor points that were initially raised with additional data, updated figures, and clarified text. Overall, this revised manuscript has improved on the previous submission and I support publication of this work.

Author Rebuttal, first revision:

Reviewer #2:

This is a revised MS for the structure of KMN and this reviewer found that authors did additional efforts for revision. Authors have faced each of concerns raised by this reviewer and responded each of them constructively. Therefore, the MS has been improved well. However, there are still two remaining concerns, which authors should address:

We thank the reviewer for his/her support.

1. In the section entitled “Structure and role of ZWINT”, authors now presented quantitative data (Extended Data Figure 10e-g, i-j). The data indicated that KNL1 levels were reduced upon ZWINT knockdown or expression of ZWINT mutants, suggesting that the KNL1-ZWINT interaction is crucial for KNL1 localization to the kinetochore. However, in lines 338-339, the authors still described, “The levels of endogenous KNL1 did not appear to be affected, suggesting that KNL1 can reach kinetochores independently of ZWINT”. This description is not consistent with their own quantitative data. Authors should describe their conclusion, based on the data. In addition, while I appreciate structure presentation of KNL1-ZWINT, it remains unclear for the role of ZWINT in the current MS. It might be better to revise section title. Alternatively, authors can describe the role of ZWINT with the clear presentation.

The reviewer is completely correct. We apologise for overlooking this required revision of our conclusions while preparing the manuscript for revision and thank the reviewer for their careful reading. We now write:

"These reduced levels of ZWINT also correlated with lower levels of endogenous KNL1, suggesting that KNL1 and ZWINT are interdependent for kinetochore localization."

2. On my previous comment as Minor Comment 4, authors responded “We only used the AF2 model in panel B...”. However, I guess that the top model of Extended Data Figure 3b (previously Fig2-Sup1b) was created by a combination of the cryo-EM structure of NSL1 with AlphaFold2 predictions, because authors described in the figure legend: “The model of the NSL1 chain in this representation includes extensions, built by AlphaFold2, to the helical segment shown in Figure 2a”. Is this model created by only AlphaFold2 or combination of the Cryo-EM structure with AlphaFold2? Please clarify this. If this model was created by the combination, it would be better to indicate which parts were extended with AlphaFold2, using a different color or a dotted line.

We thank the reviewer for raising this point. We now report in the legend to Figure 2: "Dotted lines represent invisible segments of the model of the NSL1 C-terminal tail and are merely illustrative of the missing connections. In Extended Data Figure 3b we include an AF2 model of the missing regions."

Final Decision Letter:

Message 19th Jan 2024

:
Dear Professor Musacchio,

We are now happy to accept your revised paper "Structure of the human KMN complex and implications for regulation of its assembly" for publication as an Article in Nature Structural & Molecular Biology.

Acceptance is conditional on the manuscript's not being published elsewhere and on there

being no announcement of this work to the newspapers, magazines, radio or television until the publication date in Nature Structural & Molecular Biology.

As soon as your article is published, you can generate your shareable link by entering the DOI of your article here: http://authors.springernature.com/share. Corresponding authors will also receive an automated email with the shareable link

Your paper will be published online soon after we receive proof corrections and will appear in print in the next available issue. You can find out your date of online publication by contacting the production team shortly after sending your proof corrections.

You can now use a single sign-on for all your accounts, view the status of all your manuscript submissions and reviews, access usage statistics for your published articles and

download a record of your refereeing activity for the Nature journals.

Please note that *Nature Structural & Molecular Biology* is a Transformative Journal (TJ). Authors may publish their research with us through the traditional subscription access route or make their paper immediately open access through payment of an article-processing charge (APC). Authors will not be required to make a final decision about access to their article until it has been accepted. [Find out more about Transformative Journals](https://www.springernature.com/gp/open-research/transformative-journals)

Sincerely,

Dimitris Typas
Associate Editor
Nature Structural & Molecular Biology
ORCID: 0000-0002-8737-1319